# Acetylcholine prioritises direct synaptic inputs from entorhinal cortex to CA1 by differential modulation of feedforward inhibitory circuits

Jon Palacios-Filardo[1], Matt Udakis[1], Giles A. Brown[2,4], Benjamin G. Tehan[2,4], Miles S. Congreve[2], Pradeep J. Nathan[3], Alastair J. H. Brown[2] & Jack R. Mellor [1✉]

Acetylcholine release in the hippocampus plays a central role in the formation of new memory representations. An influential but largely untested theory proposes that memory formation requires acetylcholine to enhance responses in CA1 to new sensory information from entorhinal cortex whilst depressing inputs from previously encoded representations in CA3. Here, we show that excitatory inputs from entorhinal cortex and CA3 are depressed equally by synaptic release of acetylcholine in CA1. However, feedforward inhibition from entorhinal cortex exhibits greater depression than CA3 resulting in a selective enhancement of excitatory-inhibitory balance and CA1 activation by entorhinal inputs. Entorhinal and CA3 pathways engage different feedforward interneuron subpopulations and cholinergic modulation of presynaptic function is mediated differentially by muscarinic $M_3$ and $M_4$ receptors, respectively. Thus, our data support a role and mechanisms for acetylcholine to prioritise novel information inputs to CA1 during memory formation.

[1] Center for Synaptic Plasticity, School of Physiology, Pharmacology and Neuroscience, University of Bristol, University Walk, Bristol, UK. [2] Sosei Heptares, Steinmetz Building, Granta Park, Great Abingdon, Cambridge, UK. [3] Department of Psychiatry, University of Cambridge, Cambridge, UK. [4] Present address: OMass Therapeutics Ltd, The Schrödinger Building, Oxford, UK. ✉email: Jack.Mellor@Bristol.ac.uk

Cognitive processing in the brain must continuously adapt to changing environmental situations. However, the stability of physical connectivity within neuronal networks, at least over relatively short timescales (<min), means that the brain requires systems that can enact rapid functional network reconfigurations. Release of neuromodulator transmitters via long-range projections fulfils the requirements for functional reconfiguration[1] and occurs in response to situations that demand behavioural or cognitive adaptation[2]. But the mechanisms by which neuromodulators such as acetylcholine reconfigure neuronal networks remain largely unknown.

The widespread release of acetylcholine within the brain is historically associated with arousal and attention[3–6]. More recently it has also been found to be associated with unexpected rewards or punishments[7,8] signalling the need to update existing representations with new salient information. To achieve this acetylcholine must reconfigure neural networks in two key ways: (i) open a window for encoding new memories or updating existing ones, and (ii) prioritise new sensory information for incorporation into memory ensembles[9,10]. Acetylcholine facilitates the induction of synaptic plasticity thereby opening a window for the creation of memory ensembles[11–16] and it increases the output gain from primary sensory cortices enhancing signal-to-noise for new sensory information[17–19]. It is also proposed to prioritise sensory inputs from the neocortex into memory ensembles within the hippocampus[10,11,20,21] but this critical component of the mechanism by which acetylcholine gates the updating of memory representations has yet to be tested in detail.

The hippocampus is a hub for the encoding, updating and retrieval of episodic memories, enabling events to be placed into a context. Individual items of information from the neocortex are thought to be sparsely encoded and separated by strong lateral inhibition in the dentate gyrus before being assembled into larger memory representations within the recurrent CA3 network[10,22]. These memory representations are then transferred via the Schaffer collateral (SC) pathway to CA1 which also receives new sensory information directly from the entorhinal cortex layer III pyramidal neurons via the temporoammonic (TA) pathway enabling CA1 to compare and integrate the new information[23–26]. It is therefore predicted that acetylcholine enhances the relative weights of TA inputs to CA1 over SC inputs during memory formation[10].

Perhaps counter-intuitively, acetylcholine inhibits both TA and SC glutamatergic excitatory transmission in CA1. In the SC pathway, this occurs via presynaptic muscarinic $M_4$ receptors but the identity of the receptors mediating depression at the TA pathway is unclear[27–29]. The anatomical segregation of TA and SC inputs to distal and more proximal dendritic locations on CA1 pyramidal neurons, respectively[25], together with muscarinic receptor specificity provide potential mechanisms for differential sensitivity to acetylcholine and therefore altering the relative weights of synaptic input. However, the evidence for this is equivocal with exogenously applied cholinergic agonists indicating that SC transmission is more sensitive to cholinergic modulation than TA transmission[21] but the reverse reported for endogenous synaptically released acetylcholine[28].

An alternative mechanism by which acetylcholine might rebalance the relative weights of SC and TA inputs is the modulation of the intrinsic and synaptic properties of hippocampal GABAergic interneurons[30–33] which have a profound impact on CA1 pyramidal neuron input integration rules and subsequent output[30,34]. Feedforward interneurons in the SC pathway are primarily perisomatic targeting basket cells expressing parvalbumin (PV) or cholecystokinin (CCK)[34–38] whose inhibition is strongly regulated by acetylcholine[31,33] whereas the mediators of feedforward inhibition in the TA pathway are primarily CCK or neuropeptide Y (NPY) expressing interneurons[34,37,38] that are also potentially regulated by acetylcholine[32,39]. Moreover, feedback inhibition via oriens lacunosum moleculare (OLM) interneurons, which specifically target the same distal dendritic regions as the TA pathway, are directly excited by acetylcholine[30,40]. This indicates that cholinergic modulation of inhibition within the hippocampal circuit regulates excitatory input integration and CA1 output, but the integrated effect of acetylcholine on the hippocampal network and its input-output function has not been investigated.

In this study, we tested the hypothesis that acetylcholine release in the hippocampus prioritises new sensory input to CA1 via the TA pathway over internal representations via the SC pathway. We find that endogenous synaptically released acetylcholine depresses SC and TA excitatory inputs equally but that feedforward inhibition in the TA pathway is more sensitive to cholinergic modulation. This produces an increase in excitatory–inhibitory ratio selectively for the TA pathway driven by differential regulation of interneuron subpopulations and distinct muscarinic receptor subtypes. We, therefore, provide a mechanism by which acetylcholine dynamically prioritises sensory information direct from the entorhinal cortex over internal representations held in CA3.

## Results

**Endogenous acetylcholine release modulates synaptic inputs to CA1.** To enable selective activation of endogenous acetylcholine release we expressed the light-activated cation channel channelrhodopsin-2 (ChR2) in a cre-dependent manner using mice that express cre recombinase under the control of the promoter for Choline AcetylTransferase (ChAT-cre) crossed with mice expressing cre-dependent ChR2 (ChAT-ChR2 mice; see 'Methods' section). Immunohistochemistry confirmed that ChR2 was expressed in cholinergic cells within the medial septum (Fig. 1A, B) whose axon fibres densely innervated the dorsal hippocampus (Fig. 1C) in agreement with the previously described anatomy[6]. Whole-cell patch-clamp recordings from medial septal neurons expressing ChR2 confirmed they fired action potentials in response to 5 ms of 470 nm light up to a maximum frequency of ~25 Hz (Fig. 1B). We also confirmed that light stimulation in hippocampal slices resulted in acetylcholine release. Recordings from interneurons located in Stratum Oriens revealed fast synaptic responses to light stimulation mediated by nicotinic receptors (Fig. 1D) consistent with activation of cholinergic axons and endogenous release of acetylcholine[30]. In these recordings and further recordings from CA1 pyramidal cells, we saw no inhibitory post-synaptic currents that might be caused by light-evoked co-release of GABA or excitatory glutamatergic currents from either local or long-range ChAT-expressing neurons (Fig. 1E)[41,42].

To selectively activate the Schaffer collateral and temporoammonic pathways into CA1 stimulating electrodes were placed within the two axon pathways in dorsal hippocampal slices. This enabled independent stimulation of each pathway and the engagement of both the direct excitatory inputs and disynaptic feedforward inhibitory inputs without activating direct inhibitory inputs, demonstrated by the blockade of inhibitory inputs by NBQX (20 µM) (Supplementary Fig. S1A, B). We also pharmacologically confirmed the identity of the TA input by application of the mGluR group II/III agonist DCG-IV (3 µM) that selectively inhibits glutamate release from temporoammonic pathway terminals[43] (Supplementary Fig. S1C).

To test the effect of endogenous acetylcholine release on synaptic inputs to CA1, hippocampal slices were stimulated with

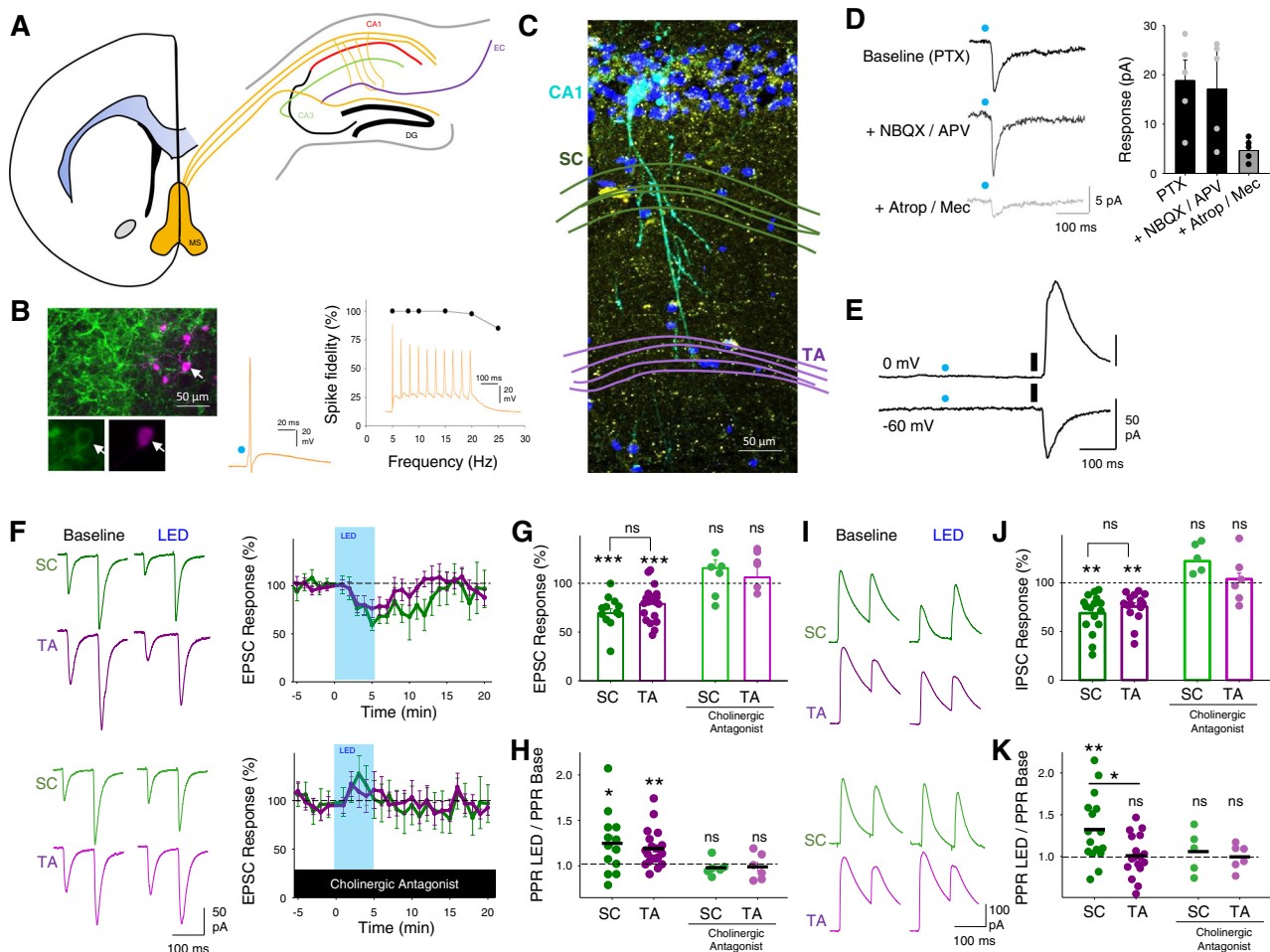

**Fig. 1 Endogenous release of acetylcholine reduces excitatory and inhibitory synaptic inputs to CA1 pyramidal neurons. A** Coronal section illustration of medial septum (MS, yellow) and its projections to dorsal hippocampus including Schaffer collateral (green) and temporoammonic (purple) inputs to CA1 from CA3 and entorhinal cortex (EC), respectively. **B** Example recording of ChR2-YFP expressing cholinergic neuron in medial septum. Left, immunofluorescence images showing cholinergic neuron (arrow) in medial septum filled with neurobiotin (purple) and expression of ChR2-YFP protein (green). Higher magnification images below show ChR2-YFP on the membrane. Right, light-evoked stimulation of cholinergic neuron (blue dot, 10 ms) reliably elicits action potentials at frequencies <25 Hz. Representative example of four independent recordings. **C** Immunofluorescence of CA1 area of the hippocampus highlighting a CA1 pyramidal neuron filled with neurobiotin (light blue) and surrounding cholinergic axons (yellow). Nuclei stained with DAPI (dark blue) and location of Schaffer collateral (SC) and temporoammonic (TA) axons illustrated in green and purple, respectively. Representative example of 5 separate slice images. **D** Light stimulation of cholinergic fibres (blue dot) elicits fast synaptic responses in Stratum Oriens interneurons recorded at −60 mV that are sensitive to atropine (25 μM) and mecamylamine (50 μM) but not picrotoxin (PTX, 50 μM), NBQX (20 μM) or D-APV (25 μM). **E** No response to light stimulation of cholinergic fibres (blue dot) was seen in CA1 pyramidal neurons recorded at 0 mV or −60 mV in contrast to electrical stimulation of SC axons (black line). **F** SC (green) and TA (purple) evoked EPSCs in CA1 pyramidal neurons are reversibly depressed by 2 Hz light stimulation (LED) of cholinergic fibres for 5 minutes (top). The depression of EPSCs is blocked by the application of cholinergic antagonists atropine (25 μM) and mecamylamine (50 μM) (Bottom). **G, H** Acetylcholine release depressed SC and TA pathway-evoked EPSCs (G; SC pathway, $n = 12$ from 6 mice, $p = 0.0002$; TA pathway, $n = 20$ from 9 mice, $p = 0.0002$; SC – TA comparison, $p = 0.46$) and increased paired-pulse ratio (H, PPR; SC pathway, $p = 0.021$; TA pathway, $p = 0.008$; SC – TA comparison, $p = 0.52$). Nicotinic and muscarinic receptor antagonists atropine (25 μM) and mecamylamine (50 μM) blocked the effects of acetylcholine release on EPSCs (SC pathway, $n = 6$ from 3 mice, $p = 0.73$; TA pathway, $n = 6$ from 3 mice, $p = 0.75$) and PPR (SC pathway, $p = 0.86$; TA pathway, $p = 0.61$). **I** Feedforward disynaptic IPSCs evoked by stimulation of SC and TA pathways are depressed by light stimulation of cholinergic fibres (top). The depression of IPSCs is blocked by the application of cholinergic antagonists atropine and mecamylamine (bottom). **J, K** Effects of acetylcholine release on SC and TA pathway-evoked IPSC response (**J** SC pathway, $n = 16$ from 10 mice, $p = 0.001$; TA pathway, $n = 16$ from 11 mice, $p = 0.009$; SC – TA comparison, $p = 0.28$) and PPR (**K** SC pathway, $p = 0.006$; TA pathway, $p = 0.75$; SC – TA comparison, $p = 0.011$). Cholinergic receptor antagonists blocked the effects of acetylcholine release on IPSCs (SC pathway, $n = 5$ from 4 mice, $p = 0.17$; TA pathway, $n = 6$ from 4 mice, $p = 0.26$) and PPR (SC pathway, $p = 0.64$; TA pathway, $p = 0.95$). Data are mean ± SEM; inter-group comparisons one-way ANOVA with post hoc Bonferroni correction. Within-group comparisons two-tailed paired $t$-test ***$p < 0.001$ **$p < 0.01$ *$p < 0.05$. Source data are provided as a Source Data file.

light at a frequency of 2 Hz for 5 min to evoke physiologically maximal acetylcholine release[44]. In the presence of the GABA$_A$ receptor antagonist picrotoxin, isolated SC and TA pathway excitatory post-synaptic current (EPSC) amplitudes were depressed by very similar amounts (Fig. 1F, G; SC pathway $69 \pm 5\%$, $n = 12$ from 6 mice, $p < 0.001$; TA pathway $79 \pm 4\%$, $n = 20$ from 9 mice, $p < 0.001$) with a concomitant increase in the paired-pulse ratio (PPR) (Fig. 1H; SC pathway, $125 \pm 9\%$, $p < 0.05$; TA pathway, $119 \pm 5\%$, $p < 0.01$), indicating a presynaptic locus of action. Application of nicotinic and muscarinic receptor antagonists atropine (25 µM) and mecamylamine (50 µM) blocked the effects of endogenous acetylcholine release (Fig. 1F–H; SC pathway $116 \pm 8\%$, $n = 6$ from 3 mice, $p > 0.05$; TA pathway $106 \pm 15\%$, $n = 6$ from 3 mice, $p > 0.05$). Therefore, contrary to our initial hypothesis[10,20,21,28], acetylcholine did not inhibit one pathway more than the other but instead depressed both equally.

We next tested disynaptic feedforward inhibitory post-synaptic currents (IPSCs) in response to stimulation of SC or TA pathways. The amplitude of evoked IPSCs was also reduced by endogenous acetylcholine release (Fig. 1I, J; SC pathway, $69 \pm 5\%$, $n = 16$ from 10 mice, $p < 0.01$; TA pathway, $76 \pm 4\%$, $n = 16$ from 11 mice, $p < 0.01$) but surprisingly IPSC PPR was only increased in the SC pathway (Fig. 1I, K; SC pathway, $132 \pm 10\%$, $p < 0.05$; TA pathway, $101 \pm 6\%$, $p > 0.05$). Similar to EPSCs, the reduction in IPSCs was completely blocked by muscarinic and nicotinic receptor antagonists (Fig. 1I–K; SC pathway IPSC $122 \pm 6\%$ and PPR $106 \pm 10\%$, $n = 5$ from 4 mice, $p > 0.05$; TA pathway IPSC $98 \pm 6\%$ and PPR $99 \pm 6\%$, $n = 6$ from 4 mice, $p > 0.05$). The observation that IPSCs were depressed equally in each pathway but PPR was increased in the SC pathway suggests that during repetitive stimulation inhibitory drive will increase in the SC pathway relative to the TA pathway. This predicts that although acetylcholine depresses excitatory synaptic transmission in the TA and SC pathways equally, its overall effect on excitatory–inhibitory ratio favours TA inputs during repetitive stimulation when the effects of acetylcholine on feedforward inhibition are taken into account.

**Cholinergic modulation of excitatory–inhibitory ratio for inputs to CA1.** To test whether excitatory–inhibitory balance was differentially altered between SC and TA input pathways we recorded monosynaptic EPSCs and disynaptic feedforward IPSCs for SC and TA pathways in the same CA1 pyramidal neuron (see 'Methods' section; Fig. 2A). 5 consecutive stimuli at 10 Hz were given alternately to SC then TA pathway to determine the evolution of synaptic modulation by acetylcholine during a repetitive train of stimuli. In these experiments, we again used light stimulation of cholinergic fibres (2 Hz for 5 min). The depression of EPSCs and IPSCs with the endogenous release of acetylcholine occurred for all responses in both SC and TA pathways (Fig. 2B and Supplementary Fig. S2D, E; SC pathway EPSC, $63 \pm 5\%$, $n = 20$ from 13 mice; TA pathway EPSC, $81 \pm 7\%$, $n = 19$ from 12 mice; SC pathway IPSC, $70 \pm 6\%$, $n = 20$ from 13 mice; TA pathway IPSC, $75 \pm 4\%$, $n = 19$ from 12 mice) but the degree of depression was not consistent between pathways over the course of repetitive stimulation. Cholinergic receptor activation enhanced synaptic facilitation and increased PPR for excitatory and feedforward inhibitory connections in the SC pathway, while the TA pathway only displayed a marked increase in PPR in excitatory but not feedforward inhibitory inputs (Fig. 2B and Supplementary Fig. S2D, E; 5th stimuli PPR change for SC EPSC, $148 \pm 10\%$, $p < 0.01$; SC IPSC, $198 \pm 32\%$, $p < 0.001$; TA EPSC, $134 \pm 10\%$, $p < 0.001$; TA IPSC, $116 \pm 8\%$, $p > 0.05$), supporting the initial results from Fig. 1. Indeed, the close similarity in PPR increase for both excitatory and feedforward inhibitory transmission in the SC pathway ensured that the excitatory–inhibitory

(E–I) ratio in the SC pathway did not change after cholinergic receptor activation for any stimuli within the train (Fig. 2C; 5th stimuli on SC E–I ratio, $0.57 \pm 0.09$ and $0.45 \pm 0.07$, for baseline and light stimulation, respectively, $p > 0.05$). Conversely, excitation-inhibition ratio in the TA pathway showed a marked increase after CCh application that evolved over the course of the train of stimuli (Fig. 2C; 5th stimuli on TA E–I ratio, $0.75 \pm 0.16$ and $0.88 \pm 0.19$, for baseline and light stimulation, respectively, $p < 0.01$). This meant that over the course of the train the TA input exerted relatively greater influence over the post-synaptic neuron compared to the SC input when cholinergic receptors were activated, as demonstrated by the comparison of excitation-inhibition ratio between the SC and TA pathways (Fig. 2D; 5th stimuli on TA/SC E–I ratio, $1.61 \pm 0.32$ and $2.45 \pm 0.45$, for baseline and CCh, respectively, $p < 0.01$).

We also tested whether we could mimic the release of endogenous acetylcholine with the application of the cholinergic receptor agonist carbachol (CCh), a non-hydrolysable analogue of acetylcholine that is not selective between cholinergic receptor subtypes. This is important for mechanistic investigations where optogenetic stimulation of cholinergic fibres are not compatible with pharmacological manipulations or optogenetic stimulation of interneurons. Application of increasing concentrations of CCh revealed that 10 µM CCh was required to induce depression for both EPSCs and IPSCs in both SC and TA pathways similar to endogenous acetylcholine release (Supplementary Fig. S2F; SC pathway EPSC, $35 \pm 6\%$, $n = 20$ from 11 mice; TA pathway EPSC, $50 \pm 5\%$, $n = 20$ from 11 mice; SC pathway IPSC, $29 \pm 3\%$, $n = 20$ from 11 mice; TA pathway IPSC, $40 \pm 4\%$, $n = 20$ from 11 mice), but at lower concentrations of CCh SC excitatory synaptic transmission showed higher sensitivity to CCh than the TA pathway[21] suggestive of different receptor affinities or signalling pathways regulating presynaptic release (Supplementary Fig. S2A; CCh 1 µM at SC pathway, $52 \pm 6\%$, $n = 9$ from 4 mice; TA pathway, $91 \pm 18\%$, $n = 9$ from 4 mice). CCh and endogenous acetylcholine also produced remarkably similar differentiation of inhibitory short-term plasticity dynamics between SC and TA pathways resulting in a robust increase in E–I ratio for the TA pathway relative to the SC pathway (Fig. 2E–G and Supplementary Fig. S2B, C; 5th stimuli PPR change for SC EPSC, $197 \pm 23\%$, $p < 0.01$; SC IPSC, $188 \pm 13\%$, $p < 0.001$; TA EPSC, $170 \pm 13\%$, $p < 0.001$; TA IPSC, $120 \pm 13\%$, $p > 0.05$, $n = 20$ from 11 mice; 5th stimuli on SC E–I ratio, $0.29 \pm 0.05$ and $0.41 \pm 0.10$, for baseline and CCh, respectively, $p > 0.05$; 5th stimuli on TA E–I ratio, $0.34 \pm 0.06$ and $0.6 \pm 0.10$, for baseline and CCh, respectively, $p < 0.001$; 5th stimuli on TA/SC E–I ratio, $1.28 \pm 0.25$ and $1.7 \pm 0.25$, for baseline and CCh, respectively, $p < 0.01$).

Analysis of the cumulative inhibitory drive across the 5 stimuli revealed that CCh or endogenous acetylcholine reduced inhibition to a much greater extent in the SC pathway compared to the TA pathway (Supplementary Fig. S2G, H) highlighting that differential modulation of feedforward inhibition between SC and TA pathways by cholinergic receptor activation produces an increase in the relative strength of the TA input to CA1 pyramidal neurons. Furthermore, the data suggest that SC and TA pathways engage distinct local inhibitory interneuron populations with different overall short-term dynamic responses to acetylcholine.

**Cholinergic modulation of feedforward inhibitory synaptic transmission.** Feedforward interneurons in the SC pathway are primarily perisomatic targeting basket cells expressing parvalbumin (PV[+]) or cholecystokinin (CCK[+]) whereas the mediators of feedforward inhibition in the TA pathway are likely dendritically targeting CCK[+] or neuropeptide Y (NPY[+]) expressing interneurons[34–38]. Analysis of our recordings revealed that feedforward SC IPSCs had

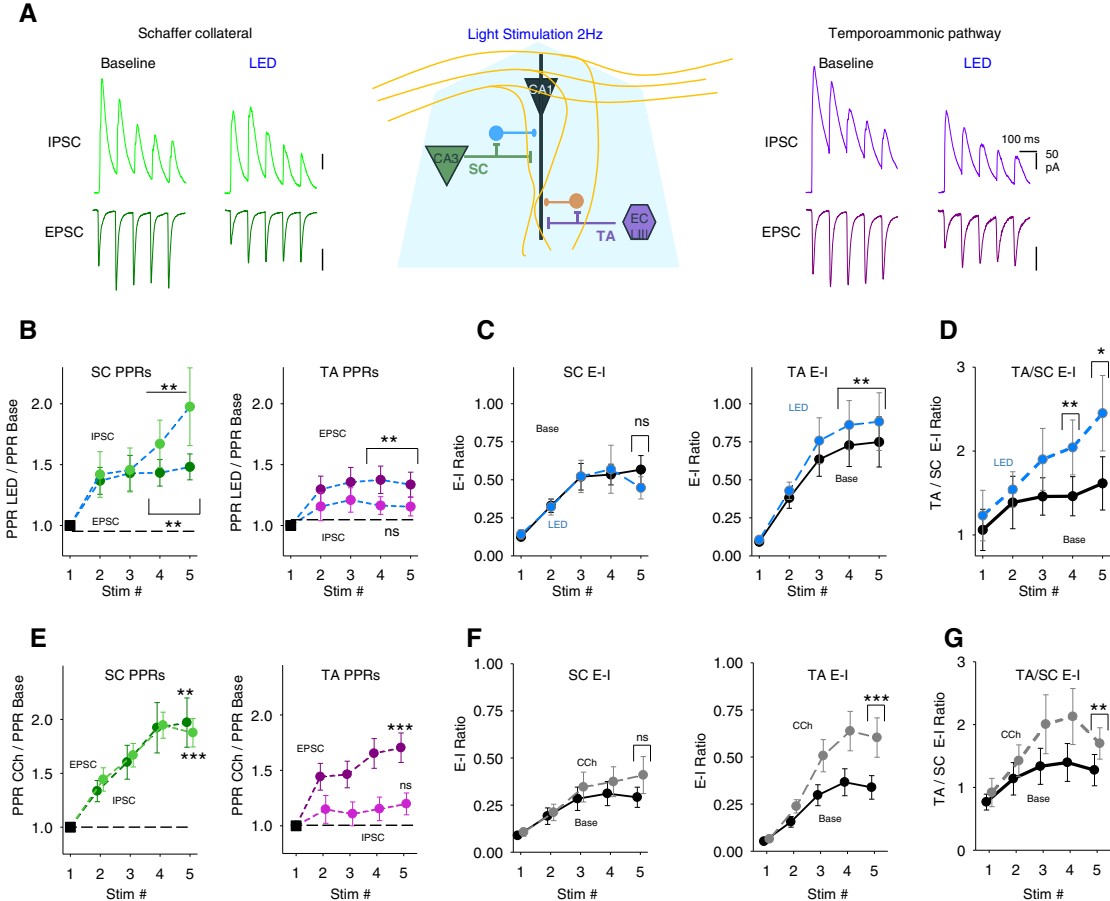

**Fig. 2 Acetylcholine release enhances excitatory–inhibitory balance for temporoammonic synaptic inputs relative to Schaffer collateral inputs.**
**A** Middle, schematic representation of the experimental approach incorporating simultaneous recording of excitatory ($V_h = -60$ mV) and feedforward inhibitory ($V_h = 0$ mV) synaptic inputs from Schaffer collateral (SC) and temporoammonic (TA) input pathways to CA1 pyramidal neuron (bottom). Example traces for EPSCs and IPSCs in response to trains of 5 stimuli at 10 Hz to SC (green, left) and TA (purple, right) pathways before and after light stimulation (LED) of cholinergic fibres at 2 Hz for 5 min. **B** Change in paired-pulse ratio (PPR) after light stimulation of cholinergic fibres for excitatory and inhibitory responses to SC (left) and TA (right) pathway stimulation (5th stimuli PPR change for SC EPSC, $n = 20$ from 13 mice, $p = 0.010$; SC IPSC, $p = 0.007$; TA EPSC, $p = 0.003$; TA IPSC, $p = 0.073$). PPR is measured compared to the first response for each response in the train. **C** Comparison of synaptic Excitatory–Inhibitory (E–I) ratio before and after light stimulation measured by charge transfer at $V_h = -60$ mV and 0 mV for SC (left) and TA (right) input pathways (5th stimuli on SC E–I ratio, $p = 0.059$; TA E–I ratio, $p = 0.007$). **D** Comparison of synaptic E–I ratio between TA and SC input pathways before and after light stimulation. Acetylcholine release enhanced the overall relative synaptic charge transfer from the TA pathway (5th stimuli on TA-SC E–I ratio, $p = 0.016$). **E–G** similar quantification to **B–D** for experiments using bath applied exogenous carbachol (CCh, 10 μM) (**E** 5th stimuli PPR change for SC EPSC, $n = 20$ from 11 mice, $p = 0.002$; SC IPSC, $p = 0.00002$; TA EPSC, $p = 0.0003$; TA IPSC, $p = 0.220$. **F** 5th stimuli on SC E–I ratio, $p = 0.285$; TA E–I ratio, $p = 0.0002$. **G** 5th stimuli on TA-SC E–I ratio, $p = 0.002$). Data are mean ± SEM; comparisons are two-tailed paired $t$-test ***$p < 0.001$, **$p < 0.01$, *$p < 0.05$. Source data are provided as a Source Data file.

faster decay kinetics than TA IPSCs (Fig. 3A–C; SC IPSC decay tau, 43.0 ± 2.7 ms, $n = 45$ from 24 mice vs TA IPSC decay tau, 60.1 ± 3.4 ms, $n = 92$ from 36 mice, $p < 0.005$). This confirms the stimulation of distinct interneuron populations and signifies either distinct GABAergic subunits at those synapses or that the more distal synaptic location of inhibitory inputs from TA feedforward interneurons and therefore increased dendritic filtering means that these IPSCs have slower kinetics[34]. GABAergic synapses from PV+ and NPY+, but not CCK+, interneurons onto CA1 pyramidal cells are depressed by μ-opioid receptors[45–47]. SC IPSCs were more sensitive to μ-opioid receptor agonist DAMGO (1 μM) than TA IPSCs (Fig. 3D, E; IPSC 1st response peak after DAMGO, 51 ± 4% and 69 ± 5%, for SC and TA, respectively, $n = 11$ from 4 mice, $p < 0.05$) indicating that in our experiments PV+ interneurons form a major component of feedforward inhibition in the SC pathway whereas CCK+ interneurons form the major component of feedforward inhibition in the TA pathway. There are also minor components from other interneuron

subtypes, most likely CCK+ basket cells in the SC pathway and PV+ or NPY+ interneurons in the TA pathway[34–38] that suggests some overlap in interneuron populations between the SC and TA pathways.

The engagement of different interneuron subtypes in feedforward inhibition in the SC and TA pathways might explain the differential modulation of feedforward inhibition by acetylcholine. Therefore, we investigated whether the output from these interneurons onto CA1 pyramidal cells is modulated by acetylcholine and, if so, whether modulation evolves differentially for the 2 inputs during a burst of responses. To test this we used mice expressing ChR2 in PV+ or CCK+ interneurons (see 'Methods' section) and gave a train of 5 light stimuli at 10 Hz to the slices whilst recording IPSCs from pyramidal neurons at 0 mV in the presence of NBQX and D-APV to avoid recording glutamatergic, disynaptic inhibitory inputs or ChR2 currents (Supplementary Fig. S3). We and others have demonstrated that neurons expressing ChR2 fire action potentials reliably at 10 Hz

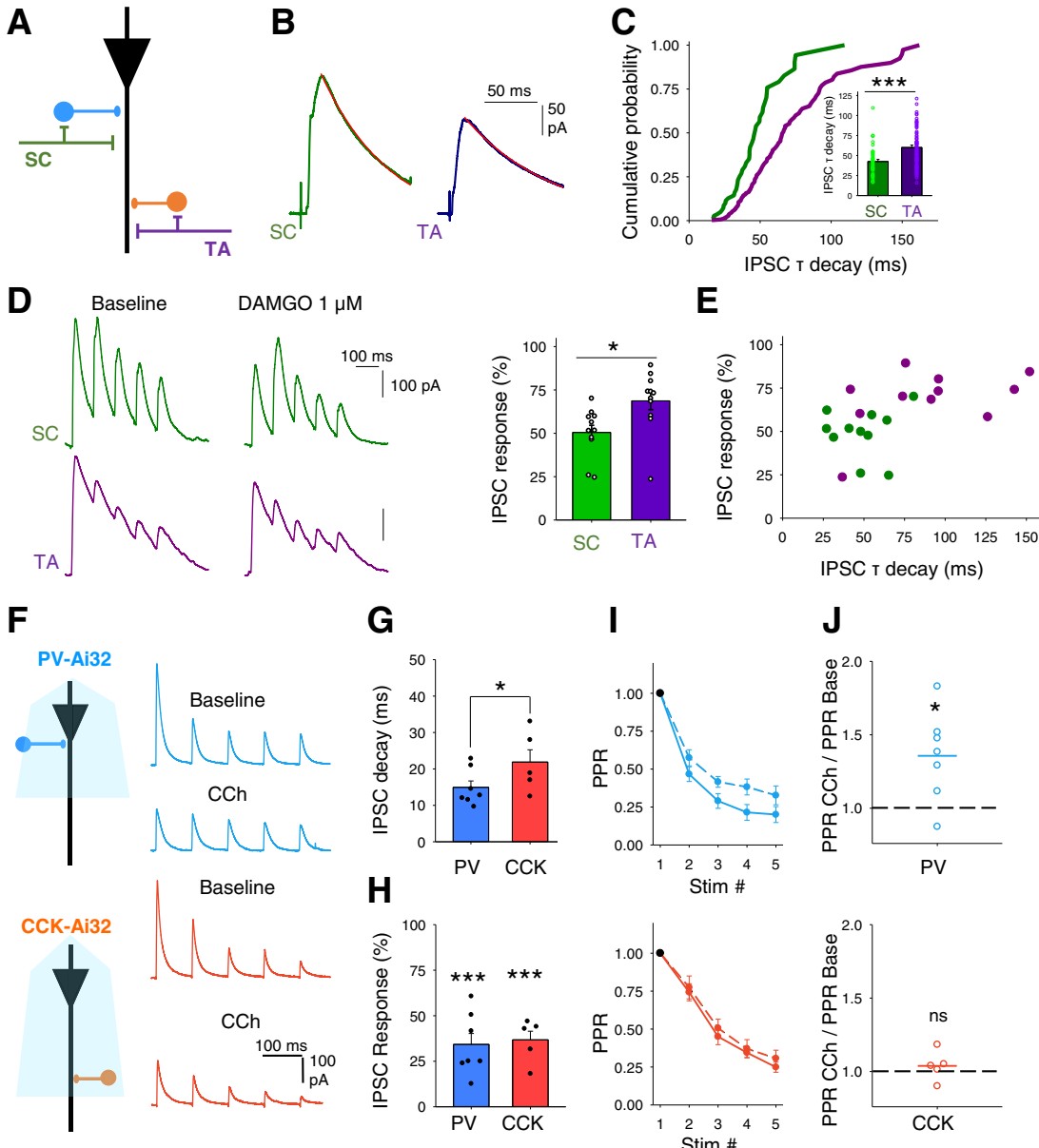

**Fig. 3 Cholinergic modulation of inhibitory inputs from distinct feedforward interneuron populations. A** Schematic representation of different feedforward interneuron populations engaged by Schaffer collateral (SC) and temporoammonic (TA) pathways within CA1. **B**, **C** Disynaptic feedforward IPSCs (**B**) and distribution of decay kinetics (**C**) for Schaffer collateral (SC, green) and temporoammonic (TA, purple) input pathways demonstrating distinct populations of feedforward interneurons. Quantification of the IPSC tau decay (insert **C**) (SC, $n = 45$ from 24 mice; TA, $n = 92$ from 36 mice; $p = 0.0002$). **D** μ-opioid receptor agonist DAMGO (1 μM) depression of disynaptic feedforward IPSCs from SC and TA pathways (SC vs TA pathways, $n = 11$ from 4 mice, $p = 0.015$). **E** IPSC decay kinetics and sensitivity to DAMGO correlate and distinguish SC from TA evoked IPSCs. **F** Optogenetic activation of either PV (top) or CCK (bottom) interneurons at 10 Hz evoked a train of IPSCs in CA1 pyramidal neurons. IPSCs from both interneurons are depressed by CCh (10 μM). **G**, **H** IPSCs from PV interneurons display faster decay kinetics than IPSCs from CCK interneurons (**G** PV, $n = 7$; CCK, $n = 5$; $p = 0.02$) but CCh depressed the IPSC amplitudes of the first responses in the train by a similar amount (**H** PV, $n = 7$, $p < 0.0001$; CCK, $n = 5$, $p < 0.0001$). **I**, **J** IPSCs from both PV and CCK interneurons demonstrated frequency-dependent depression. Frequency-dependent depression was reduced after CCh application for PV (top; $n = 7$, $p = 0.010$) but not CCK (bottom; $n = 5$; $p = 0.172$) evoked IPSCs. Data are mean ± SEM; Comparisons are two-tailed paired $t$-test ***$p < 0.001$, **$p < 0.01$, *$p < 0.05$. Source data are provided as a Source Data file.

with this stimulation protocol (Fig. 1)[38,48,49]. To test the sensitivity of $PV^+$ and $CCK^+$ synapses to cholinergic modulation, CCh was bath applied to the slice whilst selectively evoking either $PV^+$ or $CCK^+$ derived IPSCs (Fig. 3F). $PV^+$ evoked IPSCs displayed faster decay kinetics to $CCK^+$ evoked IPSCs confirming the stimulation of separate interneuron populations (Fig. 3G; $PV^+$ decay kinetics, $14.9 \pm 1.8$ ms, $n = 7$ vs $CCK^+$ decay kinetics, $21.9 \pm 3.4$ ms, $n = 5$, $p < 0.05$). The decay kinetics correlate with

the inferred contribution of $PV^+$ and $CCK^+$ interneurons to SC and TA feedforward pathways and again suggests their perisomatic and dendritic synaptic locations, respectively. Decay kinetics of optogenetically evoked IPSCs were faster than disynaptically evoked feedforward IPSCs as predicted for inputs with greater synchrony. CCh depressed IPSCs from both $PV^+$ and $CCK^+$ synapses indicating a direct cholinergic modulation of these interneurons (Fig. 3F, H; $PV^+$ responses, $34.3 \pm 6.0\%$, $n = 7$,

$p < 0.005$; $CCK^+$ responses, $37.8 \pm 4.8\%$, $n = 5$, $p < 0.005$). Both synapses exhibited frequency-dependent depression but CCh selectively increased PPR of $PV^+$ but not $CCK^+$ synapses (Fig. 3I, J; $PV^+$ IPSC PPR, $136 \pm 11\%$, $n = 7$, $p < 0.05$; $CCK^+$ IPSC PPR, $104 \pm 4\%$, $n = 5$, $p > 0.05$). The lack of effect of cholinergic receptor activation on PPR at $CCK^+$ synapses mirrors the lack of effect on PPR for feedforward inhibition in the TA pathway indicating that $CCK^+$ interneurons are the major component of feedforward inhibition in the TA pathway whereas $PV^+$ interneurons and synapses that increase PPR form feedforward inhibition in the SC pathway. The differential effect of acetylcholine at $PV^+$ and $CCK^+$ synapses provides a mechanism for the enhancement of TA pathway excitatory–inhibitory ratio in comparison to the SC pathway.

**Presynaptic modulation of the temporoammonic pathway by muscarinic $M_3$ receptors.** The synaptic depression of Schaffer collateral inputs to CA1 by acetylcholine is characterised genetically and pharmacologically to be mediated by muscarinic $M_4$ receptors[27,29]. This was confirmed by application of the dual muscarinic $M_4$ and $M_1$ receptor agonist Compound 1 (1 μM; Supplementary Fig. S4)[50], which selectively depressed SC but not TA pathway excitatory inputs (Fig. 4A–D; SC EPSC response, $63 \pm 5\%$, $n = 17$, from 8 mice, $p < 0.001$; TA EPSC response, $94 \pm 5\%$, $n = 17$, from 8 mice, $p > 0.05$). However, the identity of cholinergic receptors mediating the depression of TA inputs is unclear. Therefore, we aimed to determine which cholinergic receptors modulate the TA pathway feedforward excitatory and inhibitory synaptic transmission onto CA1 pyramidal neurons. TA pathway excitatory synaptic transmission was isolated by recording in the presence of PTX and holding the membrane voltage at −65 mV (see 'Methods' section; Fig. 4E). Similar to previous results (Figs. 1 and 2 and Supplementary Fig. S2), TA EPSCs were depressed by application of 10 μM CCh and PPR was increased (Fig. 4F; EPSC response, $43 \pm 5\%$, $n = 13$ from 7 mice, $p < 0.01$; PPR, $129 \pm 8\%$, $p < 0.05$). These data suggest a presynaptic locus of action of cholinergic receptors. We next pharmacologically dissected which cholinergic receptor subtypes were involved. Application of the non-selective nicotinic receptor antagonist mecamylamine (25 μM) had no effect on CCh depression of EPSCs (Fig. 4G; $40.6 \pm 9.5\%$, $n = 6$ from 3 mice, $p < 0.01$) and PPR (Fig. 4H; $124 \pm 8\%$, $n = 6$ from 3 mice, $p < 0.05$), while the non-selective muscarinic receptor antagonist atropine (10 μM) blocked the decrease of EPSCs (Fig. 4G; $91 \pm 4\%$, $n = 6$ from 3 mice, $p > 0.05$) and prevented the increase in PPR (Fig. 4H; $105 \pm 4\%$, $n = 6$ from 3 mice, $p > 0.05$), suggesting a direct involvement of muscarinic receptors. Muscarinic $M_1$ receptor agonist GSK-5 (500 nM)[16] did not replicate CCh depression of EPSCs and increase in PPR (Fig. 4G, H; EPSCs, $91 \pm 4\%$, PPR $101 \pm 5\%$, $n = 7$ from 4 mice, $p > 0.05$) nor did the selective $M_1$ receptor antagonist, nitrocaramiphen (1 μM) prevent CCh-induced depression and increase in PPR (Fig. 4F–H; EPSC $51 \pm 4\%$, $n = 6$ from 4 mice, $p < 0.01$; PPR $124 \pm 6\%$, $n = 6$ from 4 mice, $p < 0.05$). The high density of muscarinic $M_3$ receptors localised to Stratum Lacunosum Moleculare where TA inputs synapse in CA1[51] suggests a role for $M_3$ receptors modulating the TA pathway. Supporting a role for $M_3$ receptors, the selective $M_3$ receptor antagonist DAU5884 (1 μM)[52] prevented the EPSC depression and increase in PPR caused by CCh (Fig. 4G, H; EPSC $105 \pm 11\%$, $n = 6$ from 4 mice, $p > 0.05$; PPR $101 \pm 6\%$, $n = 6$ from 4 mice, $p > 0.05$) or endogenous acetylcholine (Fig. 4G, H; DAU5884 amplitude response $101 \pm 3\%$, $n = 7$ from 3 mice $p > 0.05$; PPR $0.97 \pm 4\%$, $p > 0.05$) suggesting that TA pathway synaptic transmission onto CA1 pyramidal neurons is modulated by muscarinic $M_3$ receptors with a presynaptic locus of action.

To confirm the involvement of muscarinic $M_3$ receptors, we tested the effects of CCh in $M_3$ receptor knockout mice (CHRM3 KO)[53]. Although TA evoked EPSCs recorded from CHRM3 KO slices were reduced by CCh with an associated increase in PPR (Fig. 4I–K; EPSC, $68 \pm 4\%$, $n = 20$ from 11 mice, $p < 0.001$; PPR, $115 \pm 5\%$, $p < 0.05$), this CCh-induced depression was less than that recorded in CHRM3 WT slices (Fig. 4J; WT EPSC vs $M_3$ KO EPSC, $p < 0.05$). This confirms the pharmacological data for $M_3$ receptor involvement in the TA pathway but also suggests some compensation for $M_3$ receptor deletion within M3 KO mice. The most likely subunit to compensate for $M_3$ deletion are $M_1$ receptors that are also coupled to Gq signalling pathways. Therefore, to further explore possible compensatory mechanisms, we tested the selective muscarinic $M_1$ receptor agonist GSK-5 in the $M_3$ KO mice (Supplementary Fig. S5). $M_1$ receptor activation depolarises and increases spike rates in pyramidal neurons[12] thereby increasing spontaneous EPSCs. Application of GSK-5 increased spontaneous EPSC frequency in slices from both WT and $M_3$ KO mice (Supplementary Fig. S5A) but caused a selective decrease in TA EPSC and the corresponding increase in PPR in the $M_3$ KO but not the WT (Supplementary Fig. S5B, C). Furthermore, the $M_1$ receptor antagonist nitrocaramiphen blocked the actions of carbachol in the $M_3$ KO mice (Supplementary Fig. S5B, C). This indicates that $M_1$ receptors partially compensate for deleted $M_3$ receptors to modulate presynaptic TA terminals in $M_3$ KO mice.

Feedforward synaptic inhibitory transmission in the TA pathway was isolated by holding the membrane voltage at 0 mV (see 'Methods' section; Fig. 4L, M). As previously described (Figs. 1, 2 and Supplementary Fig. S2), CCh depressed IPSCs without an effect on PPR (Fig. 4L–N; IPSC, $48 \pm 6\%$, $n = 9$ from 4 mice, $p < 0.01$; PPR, $108 \pm 3\%$, $n = 9$ from 4 mice, $p > 0.05$). The pharmacological data again supported a role for $M_3$ receptors. Nicotinic receptor antagonist mecamylamine (25 μM) did not prevent CCh-induced depression (Fig. 4N; $31 \pm 6\%$, $n = 5$ from 3 mice, $p < 0.01$) but the muscarinic receptor antagonist atropine (10 μM) did (Fig. 4N; $87 \pm 5\%$, $n = 6$ from 3 mice, $p > 0.05$), demonstrating that, as for excitatory synaptic transmission, inhibitory inputs to CA1 pyramidal neurons are depressed by muscarinic receptor activation. Muscarinic $M_1$ receptors did not alter TA IPSC as the agonist GSK-5 was unable to modulate inhibitory synaptic transmission (Fig. 4N; GSK-5 500 nM; $83 \pm 6\%$; $n = 4$ from 2 mice, $p > 0.05$) and the $M_1$ receptor antagonist nitrocaramiphen was unable to block the CCh effect (Fig. 4N; nitrocaramiphen 1 μM, $49 \pm 2\%$, $n = 4$ from 2 mice, $p < 0.01$). Similar to excitatory transmission, muscarinic $M_3$ receptor antagonist (DAU5884 1 μM) blocked TA pathway IPSC modulation by CCh or endogenous release of acetylcholine (Fig. 4N; CCh $84 \pm 4\%$, $n = 8$ from 4 mice, $p > 0.05$ and LED activation $93 \pm 5\%$, $n = 5$ from 3 mice, $p > 0.05$). These results indicate that $M_3$ muscarinic receptors modulate presynaptic TA terminals where they depress the release of glutamate onto CA1 pyramidal neurons and feedforward inhibition within the TA pathway.

$M_3$ receptors are expressed in entorhinal cortex pyramidal cells that project in the TA pathway and immunohistochemistry localises them preferentially to the stratum lacunosum moleculare in CA1 where TA fibres terminate without distinguishing between pre- or post-synaptic locations[51]. This evidence coupled with our results suggests that $M_3$ receptors are presynaptically located at TA synapses, but it is unusual for Gq-coupled receptors to regulate presynaptic function. Alternative mechanisms include activation of post-synaptic $M_3$ receptors causing excitation of pyramidal cells or interneurons. These could lead to reduced TA glutamate release by a retrograde messenger (such as an endocannabinoid) or GABA spillover activating presynaptic

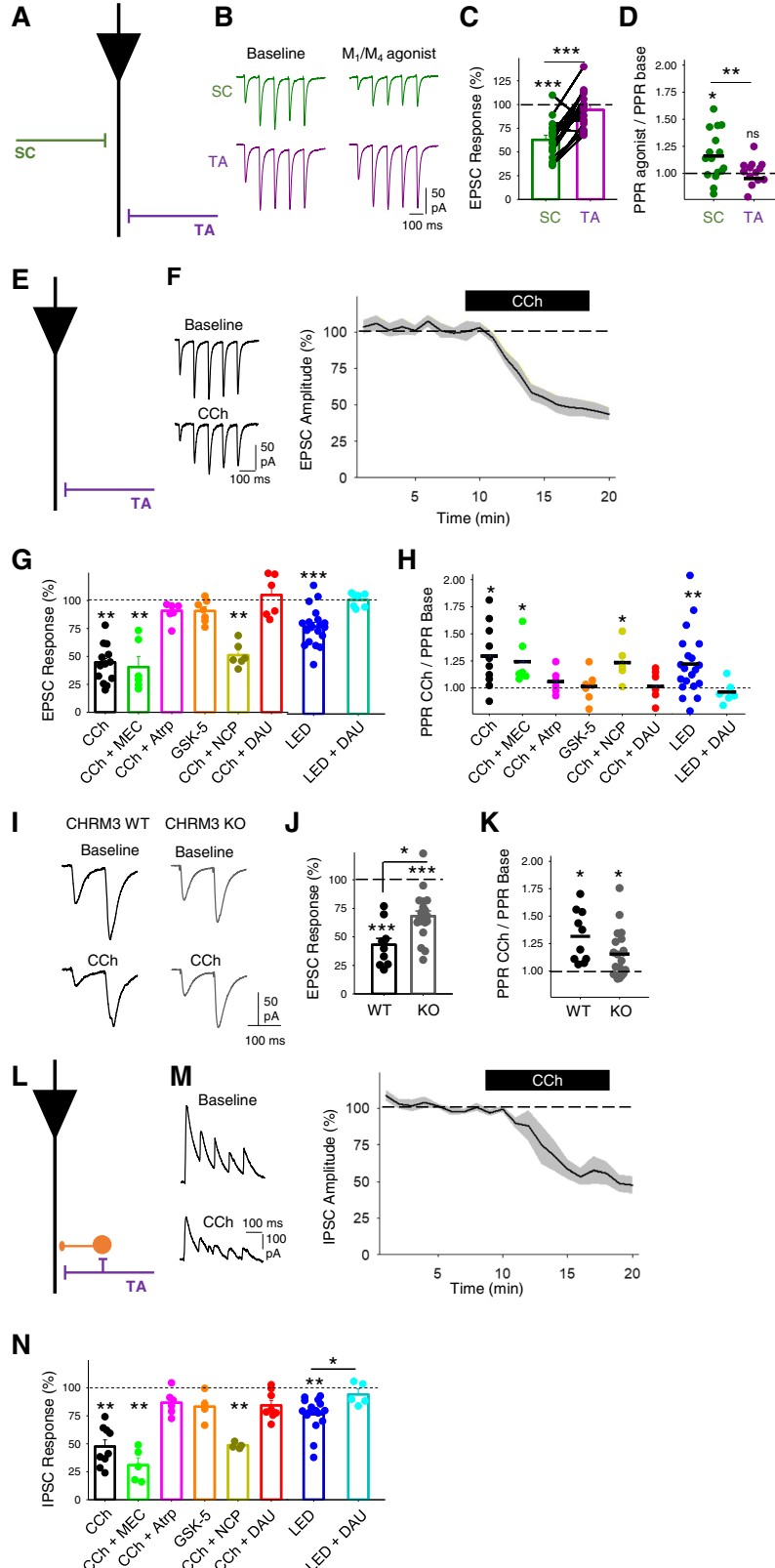

GABA$_B$ receptors. To explore the location of M$_3$ receptors we performed three additional sets of experiments to test whether M$_3$ receptors located on pyramidal cells or interneurons in CA1[31,32] regulate presynaptic neurotransmitter release. The first experiment blocked post-synaptic M$_3$ receptor function by the inclusion of GDP-β-S in the patch pipette. This prevented the

depolarisation and increase in input resistance mediated by Gq-coupled muscarinic receptors in response to the application of 10 μM CCh (Fig. 5C; membrane potential change, control $4.1 \pm 0.9$ mV, $n = 21$ from 13 mice, $p < 0.001$ and GDP-β-S $0.4 \pm 1.4$ mV, $n = 7$ from 3 mice, $p > 0.05$; input resistance change, control $25.1 \pm 5.9$ MΩ, $n = 13$ from 6 mice, $p < 0.001$ and GDP-β-

**Fig. 4 Muscarinic $M_3$ receptors modulate temporoammonic pathway EPSCs and disynaptic IPSCs in CA1 pyramidal neurons. A** Schematic illustrating recording of pharmacologically isolated EPSCs from temporoammonic (TA) and Schaffer collateral (SC) pathways. **B–D** The dual $M_1$ and $M_4$ and muscarinic receptor agonist Compound 1 (1 μM) depressed evoked EPSCs (**C**) and increased paired-pulse ratio (PPR) (**D**) for SC (green) but not TA (purple) pathway (SC EPSC, $n = 17$ from 8 mice, $p < 0.001$; TA EPSC, $n = 17$, from 8 mice, $p = 0.29$; SC vs TA EPSC comparison, $p < 0.001$; SC PPR, $p = 0.012$; TA PPR, $p = 0.265$; SC vs TA PPR comparison, $p = 0.005$). **E** Schematic illustrating recording of pharmacologically isolated EPSCs from temporoammonic (TA) pathway. **F** CCh (10 μM) reliably reduced evoked EPSC amplitudes. **G, H** Pharmacology of cholinergic depression of EPSCs. EPSC depression induced by CCh or light stimulation of cholinergic fibres (**G**) is prevented by the application of muscarinic receptor antagonist atropine (Atrp, 10 μM) or $M_3$ receptor antagonist DAU5884 (DAU, 1 μM) but not $M_1$ receptor antagonist Nitrocaramiphen (NCP, 1 μM) or nicotinic receptor antagonist mecamylamine (MEC, 25 μM) and is not replicated by $M_1$ receptor agonist GSK-5 (500 nM). PPR changes reflect conditions of cholinergic-induced EPSC depression (**H**) (CCh, $n = 6$ from 3 mice, EPSC $p = 0.001$, PPR $p = 0.021$; CCh + MEC, $n = 6$ from 3 mice, EPSC $p = 0.001$, PPR $p = 0.037$; CCh + Atrp, $n = 6$ from 3 mice, EPSC $p = 0.37$, PPR $p = 0.29$; GSK-5, $n = 7$ from 4 mice, EPSC $p = 0.32$, PPR $p = 0.52$; CCh + NCP, $n = 6$ from 4 mice, EPSC $p = 0.001$, PPR $p = 0.020$; CCh + DAU, $n = 6$ from 4 mice, EPSC $p = 0.54$, PPR $p = 0.85$; LED, $n = 7$ from 3 mice, EPSC $p < 0.0001$, PPR $p = 0.007$; LED + DAU, $n = 6$ from 3 mice, EPSC $p = 0.80$, PPR $p = 0.41$). **I–K** Comparison of the effects of CCh on TA pathway-evoked EPSCs in wild type (WT) and $M_3$ receptor knockout mice ($M_3$ KO). EPSC depression (**J**) and PPR increase (**K**) by CCh were reduced in slices from $M_3$ KOs in comparison to WT (WT EPSC, $n = 9$ from mice, $p < 0.0001$; KO EPSC, $n = 20$ from 11 mice, $p = 0.0001$; WT vs KO EPSC, $p = 0.03$; WT PPR, $p = 0.022$; KO PPR, $p = 0.020$). **L** Schematic illustrating recording of disynaptic feedforward IPSCs from pyramidal neurons at 0 mV in TA pathway. **M** CCh (10 μM) reliably reduced evoked IPSC amplitudes. **N** Pharmacology of cholinergic depression of IPSCs. IPSC depression induced by CCh or light stimulation of cholinergic fibres is prevented by the application of muscarinic receptor antagonist atropine or $M_3$ receptor antagonist DAU5884 but not $M_1$ receptor antagonist Nitrocaramiphen or nicotinic receptor antagonist mecamylamine and is not replicated by $M_1$ receptor agonist GSK-5 (CCh, $n = 9$ from 4 mice, EPSC $p = 0.004$; CCh + MEC, $n = 5$ from three mice, EPSC $p = 0.001$; CCh + Atrp, $n = 6$ from 3 mice, EPSC $p = 0.64$; GSK-5, $n = 4$ from 2 mice, EPSC $p = 0.34$; CCh + NCP, $n = 4$ from 2 mice, EPSC $p = 0.003$; CCh + DAU, $n = 8$ from 4 mice, EPSC $p = 0.17$.; LED, $n = 5$ from 3 mice, EPSC $p = 0.003$; LED + DAU, $n = 5$ from 3 mice, EPSC $p = 0.18$). Data are mean ± SEM; inter-group comparisons one-way ANOVA with post hoc Bonferroni correction. Within-group comparisons two-tailed unpaired t-test ***$p < 0.001$, **$p < 0.01$, *$p < 0.05$. Source data are provided as a Source Data file.

S $0.9 \pm 5.7$ MΩ, $n = 7$ from 3 mice, $p > 0.05$) but had no effect on the depression of excitatory TA pathway transmission when left to diffuse into the dendrites for >15 min (Fig. 5D, G, H; control EPSC $43 \pm 5\%$ and PPR $129 \pm 8\%$, $n = 13$ from 8 mice; GDP-β-S EPSC $51 \pm 5\%$ and PPR $114 \pm 8\%$, $n = 10$ from 5 mice, $p > 0.05$). The second experiment blocked $GABA_B$ receptors with CGP55845 (5 μM) to prevent activation of presynaptic $GABA_B$ receptors on TA pathway terminals. CGP55845 had no effect on CCh-induced depression of excitatory TA pathway transmission (Fig. 5E, G, H; CGP EPSC $52 \pm 6\%$ and PPR $118 \pm 5\%$, $n = 9$ from 4 mice, $p > 0.05$ compared to CCh alone). The third experiment blocked CB1 receptors with AM251 (1 μM). In the presence of AM251 CCh still induced a substantial depression of excitatory TA pathway transmission and caused an increase in PPR but the depression was slightly less than CCh alone (Fig. 5F–H; AM251 EPSC $65 \pm 4\%$ and PPR $126 \pm 9\%$, $n = 12$ from 4 mice, $p < 0.01$ for EPSC AM251 vs control). This suggests a minor component of the TA pathway depression may be mediated by CB1 receptors. However, taken together these experiments do not support a major role for $M_3$ receptors expressed on CA1 pyramidal cells or interneurons in regulating presynaptic TA pathway function and therefore support the conclusion that $M_3$ receptors are presynaptically located on TA pathway terminals.

**Pathway specific cholinergic disinhibition of CA1 output.** The modulation of hippocampal synaptic transmission and in particular the differential regulation of excitatory–inhibitory balance of SC and TA synaptic pathways predicts that acetylcholine prioritises CA1 response to inputs from entorhinal cortex via the TA pathway. To test this prediction, we monitored spike generation in CA1 pyramidal neurons in response to SC and TA pathway stimulation using trains of 10 stimuli at 10 Hz given to SC or TA pathways. The stimulus intensities were set so that post-synaptic potentials (PSPs) were suprathreshold for action potential initiation on some but not all stimuli ($P_{spike}$; see 'Methods' section). Light stimulation of cholinergic fibres (2 Hz for 5 min) decreased the probability of spiking in response to SC pathway stimulation (Fig. 6A–C; SC $P_{spike}$ baseline $0.61 \pm 0.04$ vs light stimulation $0.44 \pm 0.06$, normalised SC decrease $0.77 \pm 0.1$, $n = 13$ from 8 mice, $p < 0.05$), whereas the probability of spiking

in response to TA pathway stimulation increased (Fig. 6A–C; TA $P_{spike}$ baseline $0.43 \pm 0.05$ vs light stimulation $0.62 \pm 0.05$, normalised TA increase $1.71 \pm 0.29$, $n = 14$ from 7 mice, $p < 0.005$). This opposite modulation of SC and TA pathways was striking in a subset of recordings made from both pathways in the same neuron ($P_{spike}$ SC vs TA, $n = 11$, $p < 0.05$). These changes were matched by an increase to the time to first spike in the SC pathway and a decrease for the TA pathway (Fig. 6D; SC normalised time to first spike $1.77 \pm 0.28$, $p < 0.05$ and TA $0.72 \pm 0.05$, $p < 0.001$). Analysis of the spike probability changes for each experiment showed that the amount of spike probability change was correlated with initial spike probability, but the direction of change was not (Supplementary Fig. S7). This indicates that although TA inputs target more distal dendrites and therefore have a generally lower initial efficacy for spike generation this does not dictate the differential effects of acetylcholine on the SC and TA pathways. Importantly, the effects of light stimulation of cholinergic fibres on CA1 spike probability and latency were completely blocked by the inclusion of muscarinic and nicotinic antagonists (Fig. 6A, C, D; SC normalised $P_{spike}$ $1.04 \pm 0.1$, $n = 9$ from 4 mice, $p > 0.05$ and TA $0.82 \pm 0.16$, $n = 6$ from 4 mice, $p > 0.05$; SC normalised time to first spike $0.86 \pm 0.06$, $p > 0.05$ and TA $1.33 \pm 0.17$, $p > 0.05$). Therefore, endogenous acetylcholine release downregulates CA1 pyramidal neuron responses to the SC pathway and upregulates responses to the TA pathway.

Similar results were obtained using an exogenous application of 10 μM CCh, the major difference being that CCh depolarised CA1 pyramidal neurons (average depolarisation $5.3 \pm 0.7$ mV). To dissociate the effects of CCh on membrane potential and synaptic inputs, current was initially injected to maintain membrane potential at baseline ($i \neq 0$) and assessed changes in spike probability. Subsequently, the injected current was removed ($i = 0$) to examine how cholinergic depolarisation affected spike probability. With membrane potential maintained at baseline levels, CCh dramatically reduced the probability of spikes generated by SC pathway stimulation (Supplementary Fig. S6A₁–C₁; $P_{spike}$ baseline $0.59 \pm 0.07$ vs CCh $i \neq 0$ $0.14 \pm 0.05$, $n = 12$ from 5 mice, $p < 0.001$) and required more stimuli within a train and therefore a longer delay to generate the

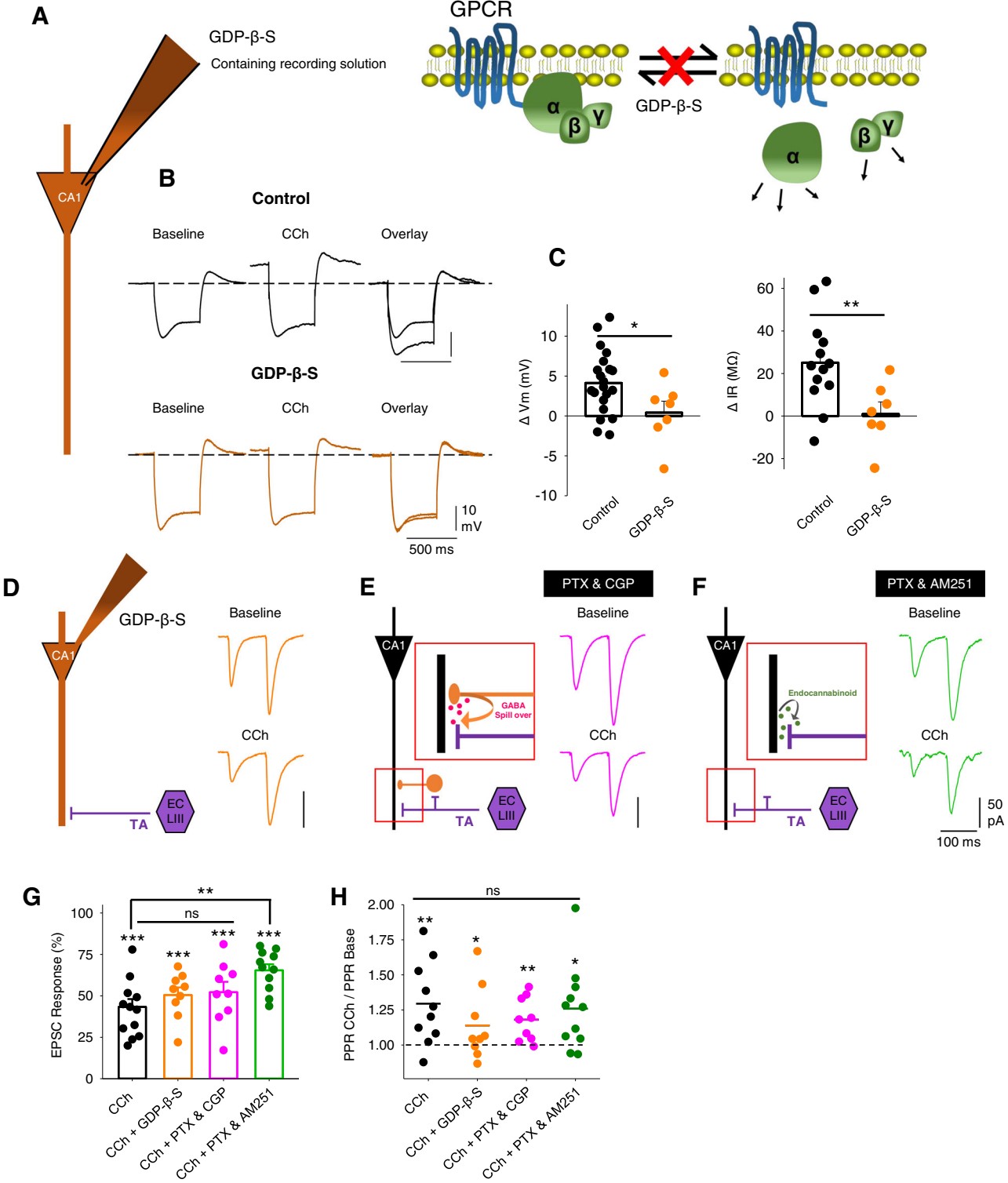

first spike (Supplementary Fig. S6A$_1$–C$_1$; baseline, 298 ± 58 ms vs CCh $i \neq 0$, 775 ± 83 ms, $p < 0.001$). With current injection removed and membrane potential allowed to depolarise, spike probability increased slightly but failed to return to baseline levels (Supplementary Fig. S6A$_1$–C$_1$; P$_{spike}$ 0.33 ± 0.05, $p < 0.05$ baseline vs CCh $i = 0$). In contrast, CCh application had little effect on TA pathway driven spike probability and delay to the first spike when the membrane potential was maintained at baseline levels (Supplementary Fig. S6A$_2$–C$_2$; P$_{spike}$ baseline, 0.33 ± 0.06 vs CCh $i \neq 0$, 0.43 ± 0.08, $n = 15$ from 10 mice, $p > 0.05$; delay to

spike baseline, 397 ± 56 ms vs CCh $i \neq 0$, 483 ± 66 ms, $p > 0.05$). However, with current injection removed and CA1 neurons allowed to depolarise spike probability increased and the delay to the first spike shortened (Supplementary Fig. S6A$_2$–C$_2$; P$_{spike}$ 0.61 ± 0.06, $p < 0.01$ vs baseline; delay to spike 280 ms ± 27 ms, $p < 0.05$ vs baseline).

Since endogenous acetylcholine reduces excitatory synaptic inputs from the SC and TA pathways equally (Figs. 1 and 2), our data suggest the acetylcholine-induced increase in spike probability in response to TA pathway input is caused by a frequency-

**Fig. 5 Presynaptically located cholinergic receptors mediate TA EPSC reduction at CA1 pyramidal neurons. A** Schematic illustrating GDP-β-S blockade of GPCR function by inclusion in the recording pipette. **B**, **C** GDP-β-S blocks Carbachol (CCh, 10 µM)-induced depolarisation of membrane potential ($V_m$, left) and reduction of input resistance ($R_{in}$, right) (control, $n = 21$ from 13 mice, GDP-β-S, $n = 7$ from 3 mice, two-tailed unpaired $t$-tests, $V_m$ $p = 0.025$, $R_{in}$ $p = 0.009$). **D** Schematic illustrating pharmacological isolation of EPSCs from TA pathway and example traces showing depression by CCh (10 µM) in the presence of GDP-β-S. **E** Schematic illustrating the putative role of GABA spillover from neighbouring synapses on presynaptic TA pathway terminals and example traces showing depression of EPSCs by CCh in the presence of GABA receptor antagonists picrotoxin (PTX, 50 µM) and CGP55845 (5 µM). **F** Schematic illustrating putative action of retrograde endocannabinoids on presynaptic TA pathway terminals and example traces showing depression of EPSCs by CCh in the presence of CB1 receptor antagonist AM251 (1 µM) and GABA receptor antagonist picrotoxin (50 µM). **G**, **H** Quantification of EPSC depression by CCh (**G** CCh, $n = 21$ from 13 mice, $p < 0.0001$; CCh + GDP-β-S, $n = 7$ from 3 mice, $p < 0.0001$; CCh + PTX + CGP, $n = 9$ from 4 mice, $p < 0.0001$; CCh + PTX + AM251, $n = 12$ from 4 mice, $p < 0.0001$; CCh vs CCh + GDP-β-S, $p = 0.31$; CCh vs CCh + PTX + GDP, $p = 0.26$; CCh vs CCh + PTX + AM251, $p = 0.002$) and change in PPR (**H** CCh, $n = 21$ from 13 mice, $p = 0.006$; CCh + GDP-β-S, $n = 7$ from 3 mice, $p = 0.047$; CCh + PTX + CGP, $n = 9$ from 4 mice, $p = 0.007$; CCh + PTX + AM251, $n = 12$ from 4 mice, $p = 0.017$; CCh vs CCh + GDP-β-S, $p = 0.18$; CCh vs CCh + PTX + GDP, $p = 0.31$; CCh vs CCh + PTX + AM251, $p = 0.79$; inter-group comparisons one-way ANOVA with post hoc Bonferroni correction. Within-group comparisons two-tailed paired $t$-test). Data are mean ± SEM; **$p < 0.01$, *$p < 0.05$. Source data are provided as a Source Data file.

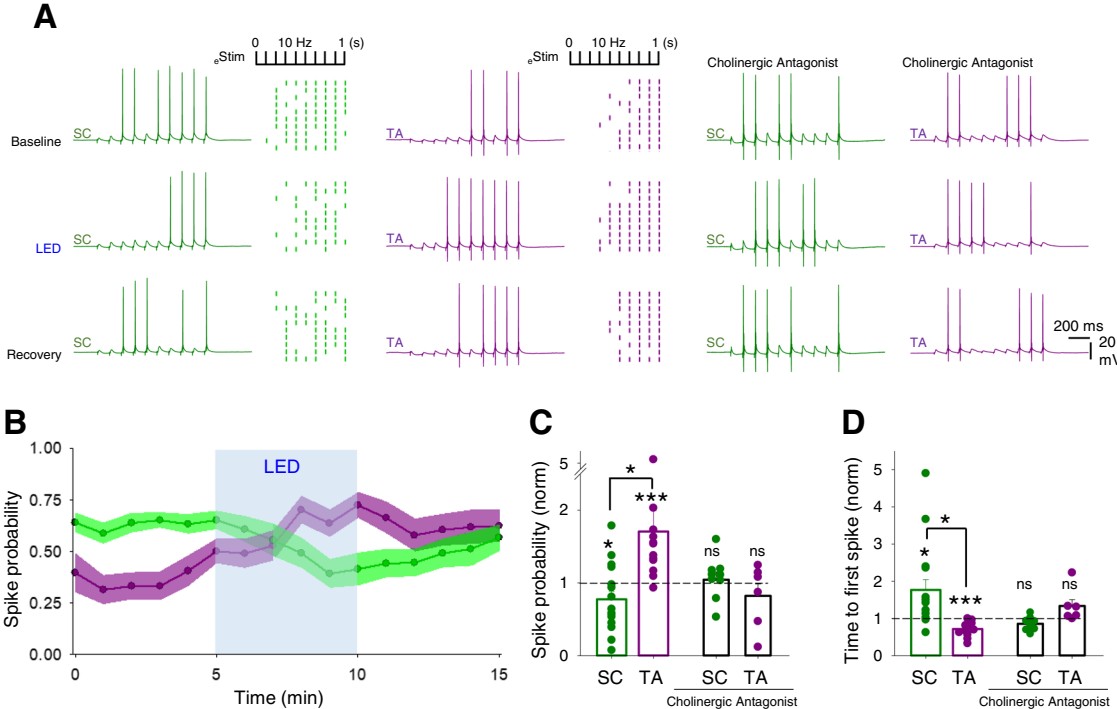

**Fig. 6 Endogenous acetylcholine release enhances CA1 response to temporoammonic over Schaffer collateral input. A** Responses in CA1 pyramidal neurons to 10 stimuli at 10 Hz given to Schaffer collateral (SC, left columns) or temporoammonic (TA, middle columns) input pathways. Raster plots show representative experiments where SC and TA pathways are stimulated alternately with trains of 10 stimuli. Light stimulation of cholinergic fibres (LED, 2 Hz for 5 min) modulates SC and TA pathways spike generation. Cholinergic receptor antagonists, atropine (25 µM) and mecamylamine (50 µM) prevented modulation of spike generation in SC and TA pathways by endogenous acetylcholine release (right columns). **B** Time course of spike probability modulation on SC (green) and TA (purple) pathway by the endogenous release of acetylcholine. **C**, **D** Quantification of spike probability (**C**) and time to first spike (**D**) modulation on SC and TA pathways in the absence or presence of cholinergic antagonists (SC spike probability, $n = 13$ from 8 mice, $p = 0.017$; TA spike probability, $n = 14$ from 7 mice, $p = 0.0002$; SC vs TA spike probability, $p = 0.046$; cholinergic antagonists, SC spike probability, $p = 0.70$; TA spike probability, $p = 0.40$; SC time to first spike, $n = 13$ from 8 mice, $p = 0.010$; TA time to first spike, $n = 14$ from 7 mice, $p = 0.0004$; SC vs TA time to first spike, $p = 0.031$; cholinergic antagonists, SC spike probability, $p = 0.07$; TA spike probability, $p = 0.12$). Data are mean ± SEM; inter-group comparisons one-way ANOVA with post hoc Tukey's correction. Within-group comparisons two-tailed paired $t$-test ***$p < 0.001$ *$p < 0.05$. Source data are provided as a Source Data file.

dependent depression of feedforward inhibition, and therefore increase in excitatory–inhibitory balance, selectively in the TA pathway (Fig. 2). Indeed, a substantial hyperpolarising envelope driven by inhibitory synaptic inputs was seen in spike probability recordings from both SC and TA pathways and could be removed by application of a GABA$_A$ receptor antagonist (picrotoxin, 50 µM) (Fig. 7A, B; SC hyperpolarising envelope $-2.45 \pm 0.63$ mVs, $n = 15$ from 5 mice vs SC GABA$_A$ antagonist $-0.10 \pm 0.66$ mVs, $n = 7$ from 2 mice, $p < 0.05$; TA hyperpolarising envelope $-3.01 \pm 0.46$ mVs, $n = 23$ from 9 mice vs TA GABA$_A$ antagonist

$-1.19 \pm 0.52$ mVs, $n = 7$ from 2 mice, $p < 0.05$). To test the importance of inhibition for prioritisation of TA inputs by acetylcholine we next repeated spike probability experiments in the presence of the GABA$_A$ receptor antagonist. Under these experimental conditions, SC pathway behaved similarly to the absence of GABA$_A$ receptor antagonist, decreasing spike generation probability upon light stimulation of cholinergic fibres (Fig. 7C–E; normalised SC decrease $0.76 \pm 0.04$ $n = 13$ from 7 mice, $p < 0.01$). In contrast, the TA pathway, which increased $P_{spike}$ after light stimulation when PSP included both excitatory

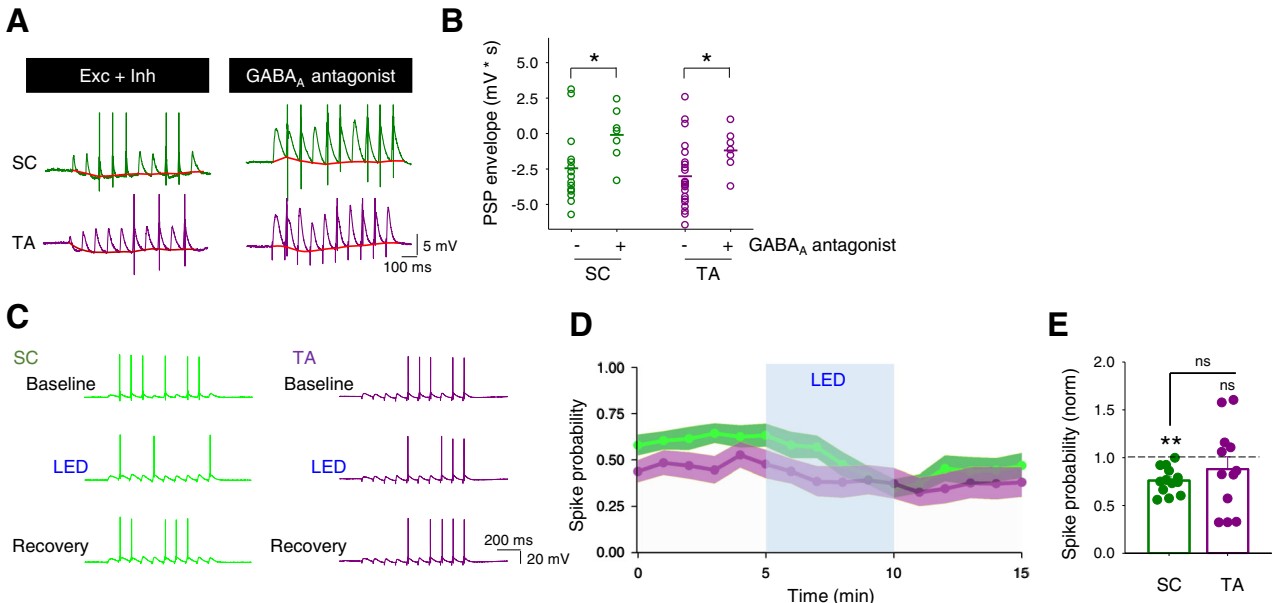

**Fig. 7 Cholinergic enhancement of CA1 responses to temporoammonic inputs is mediated by inhibition. A, B** Inhibition of GABA$_A$ receptors with picrotoxin (50 μM) reduced the underlying hyperpolarising envelope (red) in response to 10 stimuli at 10 Hz for both SC (green; control, $n = 15$ from 5 mice; GABA$_A$ antagonist, $n = 7$ from 2 mice; $p = 0.028$, two-tailed unpaired $t$-test) and TA (purple; control, $n = 23$ from 9 mice; GABA$_A$ antagonist, $n = 7$ from 2 mice; $p = 0.021$, two-tailed unpaired $t$-test) input pathways to CA1 pyramidal neurons. **C–E** In the presence of a GABA$_A$ receptor antagonist, light stimulation of cholinergic fibres (LED) reduced spike probability for SC pathway but produced no change in the TA pathway (SC spike probability, $n = 13$ from 7 mice, $p = 0.005$; TA spike probability, $n = 12$ from 6 mice, $p = 0.38$; SC vs TA spike probability, $p = 0.37$; inter-group comparisons one-way ANOVA with post hoc Bonferroni correction. Within-group comparisons two-tailed paired $t$-test). Data are mean ± SEM; **$p < 0.01$, *$p < 0.05$. Source data are provided as a Source Data file.

and inhibitory drive, yielded a similar spike probability outcome to baseline when inhibition was blocked, slightly decreasing spike probability (Fig. 7C–E; normalised TA decrease $0.88 \pm 0.23$, $n = 12$ from 6 mice, $p > 0.05$).

Similar results were obtained using an exogenous application of 10 μM CCh with inhibition blocked. For the SC pathway, CCh decreased spike generation probability when membrane potential was kept unaltered (Supplementary Fig. S5D$_1$; baseline, $0.7 \pm 0.06$ and CCh $i \neq 0$, $0.21 \pm 0.06$, $n = 9$ from 3 mice, $p < 0.01$) and showed an increase during depolarisation without reaching baseline levels ($0.44 \pm 0.07$, $p < 0.05$ vs baseline), which was correlated with delay to first spike (baseline, 318 ms ± 67 ms, CCh $i \neq 0$, 673 ms ± 122 ms, CCh $i = 0$, 397 ms ± 92 ms, $p < 0.05$ baseline vs CCh $i \neq 0$). In contrast, the TA pathway, which increased P$_{spike}$ after CCh when PSP included both excitatory and inhibitory drive, yielded a similar spike probability outcome to SC pathway when inhibition was blocked, decreasing spike probability whether membrane potential was depolarised or not (Supplementary Fig. S6D$_2$; baseline, $0.58 \pm 0.06$; CCh $i \neq 0$, $0.17 \pm 0.05$, CCh $i = 0$, $0.37 \pm 0.08$; $n = 8$ from 4 mice; $p < 0.01$ baseline vs CCh $i \neq 0$ and $p < 0.05$ baseline vs CCh $i = 0$). This was associated with increases in the delay to first spike (Supplementary Fig. S6C$_2$, D$_2$; baseline 246 ms ± 23 ms; CCh $i \neq 0$, 631 ms ± 119 ms; CCh $i = 0$, 464 ms ± 119 ms; $p < 0.05$ baseline vs CCh $i \neq 0$).

Altogether, our data indicate that cholinergic receptor activation produces a decrease of spike output in response to SC activity while enhancing output in response to TA activity via a differential effect on feedforward inhibition to CA1.

## Discussion

A long-standing and influential theory proposes that acetylcholine release in the hippocampus prioritises novel sensory information input to enable incorporation into memory ensembles[10].

This theory is based on computational modelling and the observation that SC synaptic inputs are more sensitive than TA inputs to depression caused by exogenous cholinergic agonists[10,11,20,21]. In contrast, we show that excitatory synaptic transmission at SC and TA inputs to CA1 are equally depressed by endogenous acetylcholine released in response to optogenetic stimulation (Fig. 1). Furthermore, in the absence of inhibition, we show that this results in a dramatic reduction of spike output from CA1 in response to either SC or TA input (Fig. 7). However, when we considered the effects of acetylcholine on local inhibitory networks as well as excitatory inputs, we find that acetylcholine depresses feedforward inhibition in the TA pathway more than the SC pathway over the course of a burst of stimuli (Figs. 2 and 3). This results in an overall enhancement of spike output from CA1 in response to TA input but not the SC input (Fig. 6) supporting the hypothesis that acetylcholine enhances responses to novel sensory information arriving via the TA pathway.

The regulation of local inhibitory networks by acetylcholine is therefore central to prioritisation of TA inputs by acetylcholine and differences in the regulation of synaptic output from interneuron subtypes are a critical factor. Although the interneuron subtypes engaged by the SC and TA pathways are a mixed population highlighted by their dendritic arbours extending into both stratum radiatum and stratum lacunosum moleculare (Supplementary Fig. S8), the difference in IPSC kinetics and sensitivity to the μ-opioid receptor agonist DAMGO in our recordings support previous findings that PV$^+$ cells form the majority of feedforward inhibition in the SC pathway whereas CCK$^+$ cells are the major contributors to feedforward inhibition in the TA pathway[34–38]. There is potential for slice orientation to affect interneuron connectivity networks, but these findings appear to be independent of slicing angle. Care must also be taken since there is some overlap in expression of CCK and PV in

interneuron populations[54], but our evidence and that of other groups suggests the overlap is small[38,55–57]. Crucially, the synaptic output from $PV^+$ and $CCK^+$ interneurons is differentially regulated by acetylcholine. Whilst both outputs are depressed by acetylcholine, the depression of $CCK^+$ output is greater over the course of a burst of stimuli showing enhanced depression for later responses in the burst, mirroring the effect of acetylcholine on feedforward inhibition in the TA pathway. Acetylcholine does not cause a greater depression for later responses in the burst for $PV^+$ synaptic output and therefore feedforward inhibition in the SC pathway is relatively greater over a burst of stimuli reducing the impact of SC stimulation when acetylcholine is present. Interestingly, the excitability of different interneuron subtypes is regulated by different cholinergic receptors with $M_3$ receptors in $CCK^+$ interneurons, $M_1$ receptors in $PV^+$ and $NPY^+$ interneurons and nicotinic α2 receptors in OLM feedback interneurons[30–32,39,58]. This further supports the major contribution of $CCK^+$ interneurons in the TA pathway since the $M_3$ receptor antagonist DAU5884 completely blocked the CCh-induced depression of feedforward inhibition in the TA pathway.

An alternative mechanism for cholinergic regulation of CA1 pyramidal cell output is direct modulation of dendritic excitability. Cholinergic receptors modulate multiple ion channels including HCN, Kv4.2, Kv7, SK and R-type $Ca^{2+}$ channels that broadly enhance dendritic excitability, for example by increasing HCN or $Ca^{2+}$ channel function or inhibiting $K^+$ channels[12,59–62]. These actions might be expected to increase spike output in response to synaptic input and it is possible that specific distributions of channels along the proximal-distal dendritic axis could result in enhancement of TA vs SC inputs. However, we note that in our voltage-clamp experiments the internal solution blocked most of these conductances and that blockade of inhibitory synaptic transmission prevented the change in SC and TA input influence leaving little role for direct enhancement of dendritic excitability. These mechanisms may play a greater role in the facilitation of long-term synaptic plasticity[12,63].

The differential regulation of SC and TA pathways is mediated by selective expression of $M_4$ and $M_3$ receptors. The targeting of $M_3$ and $M_4$ receptors to presynaptic terminals of TA and SC axons, respectively, fits with a broader picture of highly specific localisation of muscarinic receptor subtypes to cellular and sub-cellular domains within the hippocampus that includes the localisation of $M_2$ receptors to inhibitory presynaptic terminals of $PV^+$ basket cells. This agrees with the observed highly laminar localisation of $M_3$ receptors in the Stratum Lacunosum Moleculare, $M_4$ receptors in Stratum Radiatum and $M_2$ receptors in the Stratum Pyramidale[51] (but see ref. [28]). At each terminal, muscarinic receptors depress neurotransmitter release probability[27,29,33,51] and we show that this includes $M_3$ receptors in the TA pathway. We tested whether these $M_3$ receptors directly modulate presynaptic terminals of TA axons to depress the release of glutamate or indirectly modulate by causing the release of a retrograde messenger (Fig. 5). $M_3$ receptors are also expressed in $CCK^+$ interneurons where they increase excitability[31,32] and our data suggest that $M_3$ receptors expressed in these cells can also regulate the release of GABA at synapses onto pyramidal cells (Figs. 3 and 4). However, an increase in GABA release and spillover onto presynaptic $GABA_B$ receptors does not explain the modulation of presynaptic function (Fig. 5). Another potential mechanism for presynaptic regulation by acetylcholine is via endocannabinoid release and presynaptic CB1 receptors[64]. This is best characterised at inhibitory CCK synapses onto pyramidal cells where endocannabinoid release depresses transmission and increases PPR[65]. In our experiments, PPR does not increase (Fig. 3) and excitatory TA pathway transmission was still depressed by CCh in the presence of the CB1

receptor antagonist with only a minor reduction in the amount of depression (Fig. 5). Therefore, our data do not support a major role for endocannabinoids in cholinergic modulation of synaptic transmission. By ruling out these indirect mechanisms we conclude that $M_3$ receptors most likely act directly at presynaptic terminals of TA axons to depress the release of glutamate.

Given the importance of the TA input for synaptic plasticity in the hippocampus[26,66] it is expected that $M_3$ receptors play an important role in hippocampal-dependent learning. However, the evidence from studies using mice with genetic deletion of $M_3$ receptors is somewhat equivocal[53,67]. A potential explanation lies in the compensation for deletion of $M_3$ with the expression of $M_1$ receptors (Supplementary Fig. S4) that couple to similar Gq-mediated signalling pathways and it is interesting that knockin mutations of phosphorylation-deficient $M_3$ receptors with potentially less compensation show greater effects on learning and memory[67]. The compensation for $M_3$ deletion by $M_1$ receptors is somewhat surprising since $M_1$ receptors are generally expressed widely in somatic and dendritic cellular domains in pyramidal cells and interneurons where they regulate intrinsic excitability leading to effects on synaptic plasticity and network oscillations[12,14–16,51,68–71] but are not generally found in presynaptic terminals[72].

Cholinergic neurons in vivo fire at frequencies ranging from 0.3 to 5 Hz with higher frequencies recorded during waking activity[73,74]. Responses to salient events such as positive or negative reinforcement have been demonstrated[7,74,75] and there is evidence to suggest that acetylcholine release can be functionally targeted to specific regional or cellular domains at behaviourally relevant timepoints[42,74,76,77]. Equally, other evidence indicates substantial synchronicity of acetylcholine release between and within brain regions[7,19,74,75,78]. Even at behaviourally relevant timepoints, cholinergic firing rates do not increase markedly but rather activity across cholinergic neurons is synchronised or occurs in short bursts[74,77]. Interestingly, the release of acetylcholine plateaus at firing rates around 2 Hz[44] indicating that the dynamic range of acetylcholine release occurs at frequencies below 2 Hz. Therefore, optogenetic stimulation that synchronises release at 2 Hz over extended time periods is likely to be physiologically maximal and relevant to the functional reorganisation of hippocampal networks regardless of the specificity of cholinergic targeting. Cholinergic neurons are also reported to co-release glutamate and more prominently GABA both from long-range projections and also local cholinergic interneurons[41,42,79]. However, we found no evidence for glutamate or GABA release after optogenetic stimulation of cholinergic fibres (Fig. 1D). Therefore, under our experimental conditions, optogenetic stimulation of cholinergic fibres at 2 Hz likely provides a maximally effective release of acetylcholine without co-release of glutamate or GABA that can be mimicked by exogenous application of 10 μM CCh.

Acetylcholine increases the output gain from primary sensory cortices enhancing signal-to-noise for new sensory information and desynchronising the local cortical network by reorganising inhibition to disinhibit pyramidal neurons[17–19]. A contrary situation is reported in the hippocampus where cholinergic activation of dendritically targeting interneurons inhibits pyramidal neurons and potentially gates CA1 output[30,75,80]. Both of these mechanisms may be important for learning new representations, however, neither of these situations addresses whether acetylcholine prioritises one set of inputs over another. This is important if one considers the role of CA1 as a comparator between learned representations in CA3 and a structural framework for sensory inputs from the entorhinal cortex[81]. Here, we reveal a mechanism whereby acetylcholine alters the short-term dynamics of information processing in CA1 by acting on two

distinct muscarinic receptor subtypes located in the SC and TA pathways. This potentially reduces the role of learned representations encoded in the CA3 network in driving the output of CA1 and it will be interesting to discern in future how these various mechanisms interact across different behavioural epochs.

Multiple compounds have been developed to selectively target $M_1$ and $M_4$ muscarinic receptors for potential cognitive enhancement whereas $M_2$ and $M_3$ receptors have received much less attention due to complications with peripheral effects on cardiac and enteric function. The $M_1/M_4$ receptor dual agonist Xanomeline has cognitive-enhancing and antipsychotic efficacy in clinical trials[82,83] and whilst it is not clear whether $M_1$ or $M_4$ receptors are the key target, in separate studies, selective $M_1$ agonists and $M_4$ agonists have been shown to have memory-enhancing and/or antipsychotic efficacy[84,85] whereas deletion of $M_1$ receptors in mice causes memory deficits[86]. Our data provide a mechanism for the actions of $M_1/M_4$ receptor dual agonists such as Xanomeline and Compound 1 where activation of $M_1$ receptors facilitates synaptic plasticity[12] and activation of $M_4$ receptors prioritises new information to incorporate into memory. Our data also predict that selective activation of $M_3$ receptors could potentially facilitate the consolidation of memory by reducing interference from new information. Interestingly, the link that we demonstrate between selective muscarinic receptor activation and distinct interneuron subtypes suggests a mechanism to selectively target and regulate these interneuron populations. This could have therapeutic value in disorders with disruption to specific interneuron populations such as $PV^+$ neurons in schizophrenia[87]. Overall, acetylcholine release in the hippocampus supports cognition and the identification of specific roles for each muscarinic receptor subtype provides mechanisms to selectively modulate individual aspects of acetylcholine's actions. The identification of $M_3$ receptors as regulators of TA inputs in contrast to $M_4$ receptors acting on SC inputs provides a mechanism by which specific targeting of these muscarinic receptors could represent a therapeutic strategy to bias hippocampal processing and enhance cognitive flexibility.

## Methods

**Animal strains**. All experiments were performed using male mice. C57BL/6J (Charles River) mice were used as the background strain. The generation of the $M_3$ receptor KO mice (CHRM3 KO) has been described[53]. The $M_3$ KO mice used for this study had been backcrossed for ten times onto the C57BL/6NTac background. Cre reporter allele mice (The Jackson Laboratory) were used to tag specific neuronal populations: Cholinergic neurons (Chat-IRES-Cre; Stock No. 006410), parvalbumin interneurons (B6 PV$^{CRE}$; Stock No. 017320) and cholecystokinin interneurons (CCK-IRES-Cre; Stock No. 012706). Homozygous cre reporter mice were crossed with homozygous Ai32 mice (B6.Cg-Gt(ROSA)26Sortm32(CAG-COP4*H134R/EYFP)Hze/J; Stock No. 024109) to generate litters of heterozygous offspring expressing ChR2.

**Slice preparations**. All animal procedures were performed in accordance with Home Office guidelines as stated in the United Kingdom Animals (Scientific Procedures) Act 1986 and EU Directive 2010/63/EU 2010 and experimental protocols were approved by the British National Committee for Ethics in Animal Research.

Brain slices were prepared from P30-40 male mice. Following cervical dislocation and decapitation, brains were removed and sliced in ice-cold sucrose solution containing (in mM): 252 sucrose, 10 glucose, 26.2 NaHCO$_3$, 1.25 NaH$_2$PO$_4$, 2.5 KCl, 5 MgCl$_2$ and 1 CaCl$_2$ saturated with 95% O$_2$ and 5% CO$_2$. Parasagittal slices 350 μm thick were cut using a VT1200 (Leica) vibratome. Slices were transferred to warm (32 °C) aCSF for 30 min containing (in mM): 119 NaCl, 10 glucose, 26.2 NaHCO$_3$, 1.25 NaH$_2$PO$_4$, 2.5 KCl, 1.3 MgSO$_4$ and 2.5 CaCl$_2$ saturated with 95% O$_2$ and 5% CO$_2$ and then kept at room temperature until use.

**Electrophysiology**. Whole-cell patch-clamp recordings were made from hippocampal CA1 pyramidal neurons visualised under infra-red differential interference contrast on SliceScope Pro 6000 system (Scientifica). CA1 pyramidal neuron identity was verified by measurement of capacitance, input resistance and membrane potential. Neurons in the medial septum were patched blind to genotype. Slices were continually perfused with aCSF at 4–5 ml/min. Patch electrodes (4–7 MΩ resistance) were pulled from borosilicate glass capillaries (Harvard Apparatus)

using a PC-87 Micropipette puller (Sutter Instrument). Recording pipettes were filled with either voltage-clamp internal solution (in mM: 117 CsMeSO$_3$, 9 NaCl, 10 HEPES, 10 TEA, 2 MgATP, 0.3 NaGTP, 1 QX-314, 0.3 EGTA at pH 7.3 and 290 mOsm) or current-clamp internal solution (in mM: 135 K-Gluconate, 10 HEPES, 7 glucose, 8 NaCl, 2 MgATP, 0.3 NaGTP, 0.2 EGTA at pH 7.3 and 290 mOsm). In some experiments, GDP-β-S was included in the internal solution and allowed to diffuse for at least 15 min before application of carbachol to ensure equal concentration in all cellular compartments. Electrophysiological recordings were made with an Axoclamp 200B (Molecular Devices) filtered at 5 kHz and digitised at 10 kHz using a CED micro 1401 MKII board and Signal5 acquisition software (Cambridge Electronic Design). Series and input resistances were monitored by applying a 20 pA and 500 ms square pulse. Experiments where neurons displayed >25% change in series resistance were discarded from subsequent analysis. Membrane potentials were not corrected for junction potentials.

*Dual pathway (SC and TA) stimulation*. Bipolar stimulating electrodes were placed in CA3 to stimulate SC fibres and in the Stratum Lacunosum Moleculare (SLM) of subiculum to stimulate TA fibres. Synaptic responses were evoked alternately in either pathway at 15 s intervals. Monosynaptic EPSCs were recorded either at −65 mV membrane potential in the presence of GABA$_A$ receptor blocker picrotoxin (50 μM) or in control aCSF at the experimentally determined reversal potential for GABA$_A$ receptors (−60 mV). Disynaptic IPSCs were recorded in control aCSF at experimentally determining reversal potential for AMPA receptors (0 mV). NBQX (20 μM) was applied at the end of experiments to ascertain the contribution of direct stimulation of local interneurons to IPSCs and only responses which showed >70% reduction in IPSCs were used for analysis. In the experiments specified, SC and TA pathway EPSCs and IPSCs were recorded from the same CA1 pyramidal neuron to calculate the excitation–inhibition ratio (E–I ratio). EPSC and IPSC contributions were measured as charge transferred by calculating the area of each synaptic response in pC and the ratio of EPSC and IPSC charge for each response determined the E–I ratio. PPRs were calculated by normalising the amplitude of each response to the first response. TA over SC E–I ratio was calculated for each cell before averaging across cells.

*Current-clamp experiments* were performed at resting membrane voltage (−61.3 ± 3.5 mV). TA and SC pathways were stimulated at intervals of 20 s with trains of 10 stimuli at 10 Hz. Stimulation intensities were set to generate target spike probabilities between 30 and 70%. Spike probability was calculated as the number of spikes/number of stimuli. Time to first spike was measured from the first stimulus in the train. The post-synaptic potential (PSP) envelope was measured by calculating the area under the curve generated by joining the points of maximum hyperpolarisation in response to each stimulation as described previously[88]. Carbachol (CCh 10 μM)-induced depolarisations were neutralised by current injections to maintain a constant membrane voltage ($i \neq 0$). To investigate the impact of CCh-induced depolarisation, the injected current was removed ($i = 0$).

**Optogenetic stimulation**. Blue light from a 470 nm LED was targeted to slices via a 469 nm emission filter, a GFP dichroic mirror (Thorlabs) and the 4× (ChAT-Ai32) or 40× (PV-Ai32 or CCK-Ai32) microscope objective. 5 ms light pulses at 7–9 mW/mm$^2$ intensity were used for all stimuli. Optogenetically evoked IPSCs were recorded from pyramidal neurons at 0 mV membrane potential in the presence of the AMPA and NMDA receptor antagonists NBQX (10 μM) and DAPV (50 μM).

**Confocal imaging**. Recorded slices were permeabilized with 0.1% Triton X-100 (Sigma) and incubated with Alexa avidin (488 or 594 nm; ThermoFisher). CA1 pyramidal neurons from Chat-Ai32 mice were labelled with Alexa-594 and test proximity to cholinergic axons using Chat-Ai32 YFP fluorescence.

**Statistical analysis**. The experimental unit was defined as a cell for all conditions and only one cell was recorded from each slice. Cell and animal numbers are reported for all experiments. All data were plotted as the mean ± SEM. Where comparisons between two conditions were made paired or unpaired two-tailed Student's $t$-tests were applied as appropriate. For comparisons between more than two conditions one-way repeated measures ANOVA tests with Bonferroni post hoc correction were used. The level of significance was set to 0.05 and $p$-values are shown as follows: $*p < 0.05$; $**p < 0.01$; $***p < 0.001$. Experiments on CHRM3 WT and CHRM3 KO mice were performed blind to genotype.

**Reagents**. Carbachol (CCh), CGP55845, NBQX, DCG-IV, D-APV, picrotoxin, atropine, mecamylamine, nitrocaramiphen, DAU5884, AM251, GDP-β-S were purchased from Tocris (UK). GSK-5 was synthesised in-house at Eli Lilly and Co. Stock solutions of these compounds were made by dissolving in water. The selective muscarinic $M_1$ and $M_4$ receptor agonist Compound 1 was synthesised in-house at Sosei Heptares and dissolved in DMSO for stock solution. The purity of the final compounds was determined by HPLC or LC/MS analysis to be >95%. Additional experimental details relating to the synthesis of Compound 1 and associated structures are described in detail in WO2015/118342 which relates to the invention of agonists of the muscarinic $M_1$ receptor and/or $M_4$ receptor and which are useful in the treatment of muscarinic $M_1/M_4$ receptor-mediated diseases.

**Reporting summary**. Further information on research design is available in the Nature Research Reporting Summary linked to this article.

## Data availability
Further information and data that support the findings of this study are available upon reasonable request from the corresponding author Jack.Mellor@bristol.ac.uk. Source data are provided with this paper.

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

## Acknowledgements

We thank Paul Anastasiades and Paul Chadderton for critical input to previous versions of the manuscript and all members of the Mellor group for discussion. We also thank Dr Jürgen Wess (NIH, NIDDK) for providing the M₃ receptor KO mice and Sosei Heptares for providing Compound 1. This work was supported by Wellcome Trust (101029/Z/13/Z) and Biotechnology and Biological Sciences Research Council (BBSRC) (BB/N013956/1).

## Author contributions

Conceptualisation, J.P.-F. and J.R.M.; methodology, J.P.-F. and M.U.; investigation, J.P.-F. and M.U; provision of reagents, G.A.B., B.G.T., M.S.C., P.J.N. and A.J.H.B.; visualisation, J.P-F., M.U. and J.R.M.; writing—original draft, J.P.-F., M.U. and J.R.M.; writing—review and editing, J.P.-F., M.U., P.J.N., A.J.H.B. and J.R.M.; funding acquisition, J.R.M.; supervision, J.R.M.

## Competing interests

G.A.B., B.G.T., M.S.C., P.J.N. and A.J.H.B. are/were employees of Sosei Heptares and are/maybe shareholders of Sosei Group, the parent company of Heptares. Sosei Heptares is a drug discovery and development company working in the field of G-protein-coupled receptor structure-based drug design. J.P.-F., M.U. and J.R.M. declare no competing interests.
