## [Peer Review File · Nature Communications]

Reviewers' comments:

Reviewer #1 (Remarks to the Author):

Acetylcholine is one of the most well-studied, yet not really well-understood neuromodulators. In the hippocampus, acetylcholine is involved in numerous processes controlling network activity including the integration of new sensory information into pre-existing engrams (meant here in the context of Richard Semon's original proposal). In the present manuscript, Palacios-Filardo and co-workers aim to gain better insights into the circuit mechanisms regulated by acetylcholine. Their main hypothesis is based on the influential theory formulated by Hasselmo and colleagues postulating that acetylcholine prioritize the flow of sensory information from the entorhinal cortex to the hippocampus to update memory representations. More specifically, the authors reasoned that acetylcholine facilitates the temporoammonic pathway that presumably bring novel sensory representations to the CA1 over the Schaffer collateral pathway that is supposed to deliver pre-existing internal representations from the CA3 to the CA1. To test this hypothesis, the authors used various pharmacological and optogenetic approaches in acute hippocampal slice preparations. The manuscript reports three main findings. 1) Acetylcholine equally depresses both temporoammonic and Schaffer collateral excitatory synapses. 2) Feedforward inhibition recruited by the temporoammonic pathway is more strongly depressed by acetylcholine than feedforward inhibition triggered by Schaffer collaterals. 3) Involvement of different interneuron populations and distinct subtypes of muscarinic acetylcholine receptors underlie the larger CA1 response upon the stimulation of the temporoammonic pathway.

While the data are interesting, I have major concerns with the interpretation of all three main findings and with the main conclusion as well.

Issues with the main conclusion: The last sentence in the Abstract states that "These mechanisms enable acetylcholine to prioritize novel information inputs to CA1 during memory formation and suggest selective muscarinic targets for therapeutic intervention.". First of all, the current study used acute slice preparation. While in vivo manipulation approaches and behavioral paradigms are available to study sensory information flow and memory traces, I have serious doubts whether the in vitro slice technique is sufficient to draw such a general conclusion. In addition, the therapeutic significance is rather questionable considering the side-effect profile of drugs acting on the proposed M3 and M4 subtypes of acetylcholine receptors as the authors acknowledge themselves later in the manuscript.

Moreover, the study used either optogenetic stimulation of all hippocampal cholinergic afferents in the slice preparation or the slices were incubated with carbachol, a general cholinomimetic substance. These bulk approaches render circuit-component-specific interpretation very difficult, because basically all hippocampal cells (even glial cells) express their own specific repertoire of nicotinic and muscarinic acetylcholine receptors. Activation of these receptors substantially changes dendritic excitability, neurotransmitter release probability and potentially several other cell physiological phenomena. This enormous molecular and cellular complexity suggests that acetylcholine controls distinct circuit components (cells and synapses) in a different manner and at different time points in association with distinct behavioral components. The heterogeneity of basal forebrain cholinergic cells and their postsynaptic target cells suggest different timing for acetylcholine release at specific circuit locations under in vivo conditions (for review see Zaborszky et al 2018 *Journal of Neuroscience*, some details are described in Nyiri et al 2005 *European Journal of Neuroscience*; Takács et al 2018 *Nature Communications*). To further complicate the situation, the authors use a ChAT promoter-driven optogenetic approach that would also recruit local cholinergic interneurons that are non-GABAergic, but are likely to have a significant effect on excitatory synapses in cortical pyramidal neurons (von Engelhardt et al., 2007 *Journal of Neuroscience*). This is important, because the authors only used the GABA(A) receptor antagonist picrotoxin to exclude the contribution of local cholinergic interneurons. Finally, a similar conceptual problem exists with the compound stimulation of the temporoammonic afferents and the Schaffer collaterals. Under in vivo conditions this type of activity may more likely represent noise and the sparse coding ensembles would require the activity of much fewer relevant axons. It is very possible that physiologically released acetylcholine may differentially regulate synaptic activity of a select population of active and non-active cells and considering that almost all cholinergic afferents synaptically target certain postsynaptic neurons (Takács et al., 2018, *Nature Communications*). The authors use the term "overall enhancement of excitatory-inhibitory balance" in the Abstract and this statement is also odd in light of the very diverse circuit effects of acetylcholine. Taken

together, I am not convinced that the present in vitro and bulk stimulation methodological approaches would allow to investigate "the integrated effect of acetylcholine on the hippocampal network and its input-output function" as proposed by the authors (lines 102-104).

Issues with the first main finding:

I have difficulties to understand the meaning of "prioritize" already in the title. First of all, most of the electrophysiological data do not show robust differences in the effects of cholinomimetic manipulations. For example, acetylcholine depresses both synapse types at an equal magnitude that is indeed counter-intuitive as the authors note. Why their data deviate from prior work of Hasselmo and others and whether it has something to do with the bulk stimulation (i.e. selective relevant inputs could even be potentiated by acetylcholine)? Secondly, it is unclear why the temporoammonic input to the CA1 enjoys priority, because other subfields especially the CA3 (but also the dentate gyrus) also receive temporoammonic inputs. It remains elusive why a specific effect of acetylcholine would prioritize CA1 afferents over e.g. CA3 afferents at the network level in a behaving animal. Thinking further, the cholinergic impact on the temporoammonic-CA1 connection may not only have an effect on novel sensory information integration as solid data suggest a more broad function in other circuit computation associated with different aspects of memory formation and consolidation (see for example the Yamamoto and Tonegawa 2017 Neuron study about engram replay). The reader may also wonder if the comparison of the effect of acetylcholine on CA3 recurrent collaterals with the temporoammonic afferents would provide stronger differences in light of the clearly important role of the CA3 recurrent network in the rapid integration of novel information into engrams (see among the numerous studies for example Wagatsuma et al 2018 PNAS). Finally, CCK-positive interneurons are also likely to be recruited by temporoammonic afferents and thereby generate feedforward inhibition in the CA3 (see for example Lasztocki et al 2011 Journal of Neuroscience).

Issues with the second main finding:

The authors claim that CCK-positive interneurons mediate feedforward inhibition elicited by temporoammonic afferents, whereas Schaffer collaterals recruit parvalbumin-interneurons to trigger feedforward inhibition. This is wrong for various reasons. First of all, the presented experimental data do not show all or none differences just trends. Secondly, the slicing angle may impact differently the axonal and dendritic trees of the different interneuron types that may contribute to the trend-like small differences hence this type of interpretation from acute slice preparations is risky. Thirdly, most parvalbumin-positive basket and axo-axonic cells have extensive dendrites in the stratum lacunosum-moleculare that receive direct innervation from the entorhinal cortex (Halasy et al 1996, Hippocampus; Kiss et al 1996, Hippocampus). Fourthly, Glickfield and Scanziani 2006 Nature Neuroscience demonstrated that the two major interneuron populations are recruited by the same afferent inputs and both contribute to feedforward inhibition (the major difference lies in the temporal integration of their inputs). Fifthly, the DAMGO experiments showing a 51% versus 69% difference clearly suggest the involvement of parvalbumin interneurons in both forms of feedforward inhibition given the fairly selective expression of mu-opiate receptors.

Importantly, the other distinction suggested by the authors between the role of perisomatically- and dendritically-targeting interneurons is also wrong. In both the CA1 and CA3 subfields, the CCK-interneurons - regardless of the specific subtype - project substantial dendritic arbors into the stratum lacunosum-moleculare (see Table 1 in Lasztocki et al 2011 Journal of Neuroscience and Klausberger et al 2005 Journal of Neuroscience).

In addition, and most importantly, the authors used CCK promoter driven manipulations to dissect the two interneuron types. Unfortunately, the used mouse line turned out to be not reliable and various lab reported at interneuron conferences that PV-positive cells are also affected in these mice. Therefore, the authors should directly investigate by using immunostaining whether ChR2 is exclusively expressed in PV-positive cells in the first line and if ChR2 is restricted to CCK interneurons, but it is completely absent in PV neurons in the second mouse line. In case these lines are not specific enough (and I have serious doubts) then conclusion about the specific contribution of the different interneuron types cannot be drawn.

Finally, a general conceptual problem arises from the fact that the major interneuron type that controls the temporoammonic pathway is the O-LM interneuron. As shown by the Scanziani lab and others, these cells are readily activated by cholinergic activity via their specific alpha-2 nicotinic cholinergic receptors. Thus, optogenetic activation may trigger their feedback control in most experiments (except the case when the nicotinic receptor antagonists was applied. This further complicates the picture how distinct interneuron types and their control by acetylcholine

would differentiate between efficacy of the temporoammonic and the Schaffer collateral inputs to recruit feedforward inhibition.

Issues with the third main finding:

The authors propose that presynaptic M3 muscarinic receptors control the temporoammonic pathway, whereas M4 receptors regulate the Schaffer collaterals. The role of M4 has already been published and Sanchez et al 2009 Journal of Neuroscience Research should be cited in this regard. Moreover, I am not convinced that M3 is presynaptically located on the entorhinal afferents. According to the Allen Brain Atlas, the expression of M3 receptor is very high in CA1 pyramidal neurons, but very modest in the entorhinal cortex indicating a postsynaptic function for M3 in pyramidal neuron dendrites. M3 activity is coupled to Gq/11 signaling that is usually a somatodendritic signaling mechanism. In contrast, M4 activity is coupled to Gi/o signaling that is indeed ideal for a presynaptic function i.e. the inhibition of neurotransmitter release. The authors based their arguments of paired-pulse ratio measurement to draw conclusion. However, they failed to consider the possibility that postsynaptic M3 muscarinic receptors may readily trigger an indirect retrograde messenger that could act presynaptically. In accordance, Fukudome et al (2004, European Journal of Neuroscience) showed that endocannabinoids are released by somatodendritic M3 activation and remove inhibition of pyramidal neurons via presynaptic type I cannabinoid receptors located on GABAergic inputs. This indirect local mechanism could fully explain the authors findings on the increased CA1 output without any consideration of differential recruitment of feedforward inhibition (which is doubtful see above). Moreover, under in vivo conditions, endocannabinoids will likely be released by those pyramidal neurons that are very active (i.e. part of the engram) thus cholinergic effects via somatodendritic M3 receptors and retrograde endocannabinoid-mediated suppression of inhibition could readily facilitate the integration of novel external information with the pre-existing internal representations. Taken together, in light of these major concerns about all three major findings, I suggest the authors seriously reconsidering the interpretation of their findings, performing additional control experiments and applying more specific manipulation approaches before publishing their data in any journal.

Additional minor comments:

In contrast to the statement in lines 80-81, there is no data about the differential sensitivity of M3 and M4 receptors.

There is a typo in line 86 (weights).

In Figure 1B, it is not possible to see the colocalization of blue (neurobiotin) and yellow (ChR2) channels. Please add single channel image panels.

Differences in IPSC rise times would be much more informative about the subcellular location of the afferent synapses than IPSC decays that are more likely be explained by distinct subunit compositions, synaptic receptor density, or opening time (e.g. due to different phosphorylation states).

In line 312, the word "confirm" is an overstatement, the knockout mice cannot be used to demonstrate presynaptic localization in pharmacological experiments.

Reviewer #2 (Remarks to the Author):

The article from Palacios-Filardo and other address the effect of 'physiological' ACh release in 2 different hippocampal pathways. The authors explore paradigms widely accepted that have little experimental evidence (e.g. dichotomy in septal glutamatergic and cholinergic neurons in theta or memory). The article is very interesting and is timely and in my opinion should be published in 'prestige' journals in order to reach a broader audience. I have however, a few comments that are not 'major' but are necessary for my acceptance:

Lines 99-101:

the same distal dendritic regions as100 the TA pathway, are directly excited by acetylcholine (Leao et al., 2012; Pouille and Scanziani, 2004). This indicates that cholinergic modulation This indicates that cholinergic modulation of inhibition within the hippocampal circuit strongly dictates excitatory input integration and CA1 output

please remove the 'strongly'. The nicotinic response in OLM neurons is only strong if there is a very strong coordination of MS cholinergic neurons. Compared to local glutamatergic excitation, cholinergic EPSCs are small.

Line 512:

Again, at least in Leao et al 2012, the hypothesis was that if there would be enough cholinergic excitation through nicotinic receptors, ACh would then excite the OLM population and therefore modulate gating. No muscarinic effect was explored in that article. Nevertheless, a recent in vivo study does suggest that septal ACh excite OLM cells (Mikulovic et al, Nat Comm 2018).

Whole cell studies involving metabotropic receptors are still not widely accepted (dialysis, channel runout etc). I personally believe that these studies are valid and the g protein runout can be manageable if small pipettes are used with low ATP concentration in the internal solution (Fiszman et al 1991, Verrecchia et al 1999). Nevertheless, the authors should discuss the caveats of their methods in studying ACh effects in pathways using whole cell.

In my opinion, the only weak point in the article is the lack of experiments involving HCN pharmacology. ACh strongly modulate HCN channels (Zhao et al. Front Cell Neurosci. 2016). These channels have an obvious role in short term plasticity and internal ATP can differentially affect HCN subunits Pian et al J Gen Physiol 2006). Ideally, the authors could add a voltage clamp experiment using hyperpolarising steps (to trigger Ih) concomitant to light stimulation of Chat cells to show if cholinergic modulation of Ih could play any role in modulating synaptic currents. Or, test TA vs SC inputs while applying ZD7288 and optical stimulation. Or at least discuss whether or they believe Ih activation/inactivation could play a role in their interpretation.

Reviewer #3 (Remarks to the Author):

This is a very nice manuscript by Palacios-Filardo and colleagues that shows convincingly a role for differential cholinergic regulation of feedforward excitatory and inhibitory drive within the CA1 hippocampus. Using a combination of optogenetics and electrophysiology they clearly demonstrate that endogenous release of Ach or exogenous application of Ach ligands alters the nature of both excitatory and inhibitory drive onto the CA1 circuit. Using a number of clever manipulations they show that while Ach modulates excitatory inputs onto CA1 pyramidal cells from the entorhinal and CA3 subfield in a similar manner, disynaptic inhibition arising through the temporoammonic pathway is preferentially reduced compared to inhibition arising through Schaffer collateral inputs. The consequence of this is to prioritize entorhinal drive onto CA1 pyramids, suggesting an underlying mechanism for the shift in afferent drive observed during novel memory formation. As stated above this is a very nice piece of work however it suffers somewhat from being quite dense and difficult to follow. I work in this field and I found it hard to navigate and keep all of the parts in my head while trying to understand the manuscript. The authors are urged to either present the data in a slightly more straightforward manner and also provide a cartoon/schematic that highlights the main findings of the manuscript.

1. My biggest issue with the manuscript is the "bait and switch" approach used in Figure 1. In figure 1 The authors present a beautiful set of data obtained from a mouse line that has ChR2-encoded in CHAT-positive cells. They convincingly demonstrate that they can monitor released Ach and that it successfully modulates excitatory synaptic transmission in a slice preparation. This is a significant achievement in itself. However the rest of the manuscript resorts to using bath applied cholinergic agents. Although the data shown in Figure 2 are compelling one cant help but wonder why the authors had gone to the trouble of setting up the previous experiments illustrated in Figure 1 using intrinsic Ach release via optogenetics and then for some reason they resort to bath application of carbachol. Given the data in figure 1 showing that the authors have reliably established that they can release endogenous Ach in their system, was there a compelling reason

to resort back to bath applied CCh? I would urge the authors to repeat this (and other significant) expt using the experimental set up used in figure 1 since bath application of the agonist removes all of the temporal dependence of Ach released onto different targets, which will undoubtedly play into the overall response pattern observed. The ability to successfully release Ach optogenetically and show it can modulate the circuit is outstanding but I feel that the rest of the manuscript undercuts the true potential of this observaiton

2. Figure 3 and related text. The authors should provide clarification that both CCK+ and PV+ cells follow the 5x 10Hz light stimuli faithfully used to evoke the train of IPSCs studied in this expt.

3. Line 254 it is not correct to assume that the difference in decay kinetics reflect simply the somatic versus dendritic location of the inhibitory inputs. There are as many CCK-basket cells located in s. radiatum as there are dendritic targeting CCK cells. The kinetics may simply reflect the kinetics of the underlying GABA receptors at the different synapses.

4. Figure 3 and related text. How do the authors interpret the lack of change of release probability caused by CCh bath application in the CCK cell cohort? Is this a consequence of muscarinic receptor activation on pyramidal cells shunting the IPSC? How do the authors rule out an indirect effect of CCh on the pyramidal cell in this dataset?

5. In both figure 5 and Figure 7 the initial spike probability appears to be set higher for the SC input in control conditions than the TA input. This may have a potential impact in the outcome of the response seen after light driven Ach response or the CCh rersponse. In fact this would have a major influence on the time to first spike dataset since the SC input has a higher probability of firing in the first few spikes compared to the TA input dataset. Why were the initial spike probabilities not set to be equivalent in both inputs?

Minor comments

Lines 30-32 "Underpinning the prioritization..." This sentence is a little clumsy and should be reworded

Line 71-73 please provide a reference for this prediction

Figure 1 panel E. please clarify the nature of the electrically evoked events seen in the panel. Their identity is left ambiguous to the reader.

Response to reviewers' comments (comments in black, response in blue)

Reviewer 1:

Issues with main conclusion.

1) The last sentence in the Abstract states that “These mechanisms enable acetylcholine to prioritize novel information inputs to CA1 during memory formation and suggest selective muscarinic targets for therapeutic intervention.”. First of all, the current study used acute slice preparation. While in vivo manipulation approaches and behavioral paradigms are available to study sensory information flow and memory traces, I have serious doubts whether the in vitro slice technique is sufficient to draw such a general conclusion.

We feel this final sentence has been taken out of context in this instance. The opening sentences of the abstract set out the background for the hypothesis that we test: “Acetylcholine release in the hippocampus plays a central role in the formation of new memory representations by facilitating synaptic plasticity. An influential but largely untested theory proposes that memory formation requires acetylcholine to enhance responses in CA1 to new sensory information from entorhinal cortex whilst depressing inputs from previously encoded representations in CA3.”. The most direct way to test this hypothesis is to measure the effect of acetylcholine release on the synaptic strength at these two input pathways. Acute slice electrophysiology is the best and most direct approach to measure the relative strength of synaptic inputs within neuronal networks and the mechanisms underlying their modulation. Therefore, we believe our approach is the most appropriate and might lead in future to further complementary testing of the hypothesis by as yet unspecified measurement of sensory information flow and memory traces as suggested by the reviewer. However, we note that any results from in vivo experiments would almost certainly rely on our data for mechanistic understanding.

Given this context and that it is well established that entorhinal inputs to CA1 carry novel sensory information and that acetylcholine release occurs during memory formation (see manuscript introduction), we did not feel it was unreasonable to conclude that we provide a mechanism that enables acetylcholine to prioritize novel information inputs to CA1 during memory formation. However, to remove any doubt we have now substantially revised the abstract including the concluding sentence to say “Thus, our data support a role for acetylcholine to prioritise novel information inputs to CA1 during memory formation and provide mechanisms indicating selective muscarinic targets for therapeutic intervention.”.

2) In addition, the therapeutic significance is rather questionable considering the side-effect profile of drugs acting on the proposed M3 and M4 subtypes of acetylcholine receptors as the authors acknowledge themselves later in the manuscript.

We appreciate that the pathway from target identification to drug therapy is often tortuous but we disagree with the dismissal of an approach based on muscarinic receptors. Muscarinic receptor targeting compounds represent very rare instances of success (cited in the manuscript) in an otherwise moribund landscape. Furthermore, pharmaceutical companies (including the SoseiHeptares co-authors of this study, Cerevel and a collaboration between

Acadia and Vanderbilt who have recently announced clinical programs for selective muscarinic compounds) have invested substantially in developing selective muscarinic agents for the treatment of psychiatric and neurological disorders. The side effect profiles of current non-selective muscarinic agents are precisely why more selective compounds are required coupled with a better understanding of the role of muscarinic receptor subtypes within the brain.

3) Moreover, the study used either optogenetic stimulation of all hippocampal cholinergic afferents in the slice preparation or the slices were incubated with carbachol, a general cholinomimetic substance. These bulk approaches render circuit-component-specific interpretation very difficult, because basically all hippocampal cells (even glial cells) express their own specific repertoire of nicotinic and muscarinic acetylcholine receptors. Activation of these receptors substantially changes dendritic excitability, neurotransmitter release probability and potentially several other cell physiological phenomena. This enormous molecular and cellular complexity suggests that acetylcholine controls distinct circuit components (cells and synapses) in a different manner and at different time points in association with distinct behavioral components. The heterogeneity of basal forebrain cholinergic cells and their postsynaptic target cells suggest different timing for acetylcholine release at specific circuit locations under in vivo conditions (for review see Zaborszky et al 2018 *Journal of Neuroscience*, some details are described in Nyiri et al 2005 *European Journal of Neuroscience*; Takács et al 2018 *Nature Communications*).

The view that acetylcholine release specifically targets distinct cellular processes at different behavioural timepoints is attractive, but also speculative, and the data to support such a conclusion is suggestive at best, as outlined by the reviewer. Perhaps the best evidence to support such a scenario was recently published from the group of Balazs Hangya (*Nat Neurosci* August 2020 and now cited in the revised manuscript) showing dissociable subgroups of cholinergic neuron firing patterns related to behaviour. However, the limited information on the targets for these identified subtypes suggest a dissociation *between*, rather than *within*, brain regions. So, whilst it is formally plausible that acetylcholine release is targeted selectively within a brain region such as the hippocampus, to our knowledge there is no substantial published evidence to support this. Indeed, the Takacs paper cited specifically states that although acetylcholine is likely released from synaptic locations there is no evidence that it is targeted to specific cells or synapses. Conversely, there is a substantial body of data supporting a more generalised release. Published evidence from our group and others suggests that functionally there is remarkable synchronicity between acetylcholine release within and between different brain regions on both fast and slow timescales (eg Lovett-Baron et al 2014 *Science*, Eggerman et al 2014 *Cell Reports*, Hangya et al 2015 *Cell*, Teles-Grilo Ruivo et al 2017 *Cell Reports*, Crouse et al 2020 *eLife*). Therefore, the anatomical heterogeneity of cholinergic innervation cited by the reviewer does not necessarily translate into functional heterogeneity particularly within a brain region and we believe further investigation is required before we have concrete evidence to support one or the other conclusion.

In summary, given the lack of strong evidence for cholinergic heterogeneity within the hippocampus we believe it is reasonable to use optogenetic stimulation of multiple cholinergic fibers in the slice at frequencies deduced from in vivo recordings as the most

physiological and best approach to understanding the modulation of hippocampal networks by acetylcholine. Furthermore, this approach demonstrates fundamental principles for cholinergic modulation that would still apply even if it is found that cholinergic modulation is more targeted than the data currently suggests, and we now include discussion of these ideas in the revised manuscript (lines 536-548).

4) To further complicate the situation, the authors use a ChAT promoter-driven optogenetic approach that would also recruit local cholinergic interneurons that are non-GABAergic, but are likely to have a significant effect on excitatory synapses in cortical pyramidal neurons (von Engelhardt et al., 2007 *Journal of Neuroscience*). This is important, because the authors only used the GABA(A) receptor antagonist picrotoxin to exclude the contribution of local cholinergic interneurons.

We are fully aware of the potential recruitment of GABAergic and glutamatergic transmission from long-range and local cholinergic neurons and we tested this in recordings from both CA1 pyramidal cells and OLM interneurons. These controls are described in the manuscript and those specifically for glutamatergic transmission are shown in Figure 1D,E. In neither CA1 pyramidal cells nor OLM interneurons did we ever find evidence for GABAergic or glutamatergic release in response to optogenetic stimulation of ChAT+ cells.

The data cited by the reviewer refers specifically to interneurons expressing ChAT in the neocortex but there is no evidence to our knowledge that similar neurons are found in the hippocampus. Furthermore, the increase in spontaneous excitatory transmission reported in the cited paper is stated as modest (average of 0.64 Hz or ~10%) in response to stimulation frequencies much higher than those that we use in our study. We did not observe any increase in spontaneous EPSC frequency in response to optogenetic acetylcholine release. Finally, an increase in release probability caused by presynaptic nicotinic receptors at excitatory synapses suggested in the von Engelhardt paper would be expected to cause an increase in evoked transmission when in fact we see the reverse at both SC and TA synapses. For all these reasons we do not consider this issue to have any major impact on our results and stated in the discussion (lines 545-548).

5) Finally, a similar conceptual problem exists with the compound stimulation of the temporoammonic afferents and the Schaffer collaterals. Under in vivo conditions this type of activity may more likely represent noise and the sparse coding ensembles would require the activity of much fewer relevant axons. It is very possible that physiologically released acetylcholine may differentially regulate synaptic activity of a select population of active and non-active cells and considering that almost all cholinergic afferents synaptically target certain postsynaptic neurons (Takács et al., 2018, *Nature Communications*). The authors use the term “overall enhancement of excitatory-inhibitory balance” in the Abstract and this statement is also odd in light of the very diverse circuit effects of acetylcholine. Taken together, I am not convinced that the present in vitro and bulk stimulation methodological approaches would allow to investigate “the integrated effect of acetylcholine on the hippocampal network and its input-output function” as proposed by the authors (lines 102-104).

This comment is related to point 3 and articulates an interesting idea that somehow cholinergic inputs selectively target behaviourally relevant subsets of hippocampal neurons. Again, this is an attractive idea but to our knowledge there is no data to support this viewpoint. The cited Takacs et al paper shows acetylcholine is released at synaptic locations in the hippocampus but states that this does not mean it is targeted to specific cellular or synaptic processes. Therefore, given that we have no strong hypothesis for which cells or synapses might be targeted specifically by acetylcholine in a behaviourally-relevant manner, it is reasonable to investigate the general impact of cholinergic inputs on separate populations of TA or SC inputs to CA1. Again, we include additional discussion of these ideas in the revised manuscript highlighting how our approach provides fundamental principles that may apply to many anatomical arrangements (lines 536-548)

Issues with the first main finding:

6) I have difficulties to understand the meaning of “prioritize” already in the title. First of all, most of the electrophysiological data do not show robust differences in the effects of cholinomimetic manipulations. For example, acetylcholine depresses both synapse types at an equal magnitude that is indeed counter-intuitive as the authors note. Why their data deviate from prior work of Hasselmo and others and whether it has something to do with the bulk stimulation (i.e. selective relevant inputs could even be potentiated by acetylcholine)?

It is important to re-iterate the key mechanism underpinning the differential regulation of SC and TA inputs - namely the critical importance of feedforward inhibition. The result is robust and we now demonstrate it occurs when cholinergic receptors are activated by endogenous acetylcholine or exogenous carbachol (Figure 2). We show robust statistically significant effects of acetylcholine that depress the efficacy of combined excitatory and inhibitory SC inputs to a greater extent than TA inputs (Figure 2). We show that this causes CA1 pyramidal neurons to increase responsiveness to TA inputs whilst decreasing responsiveness to SC inputs thereby prioritizing TA inputs from entorhinal cortex (Figure 6). We hope that this additional clarification is helpful.

We do not know why our data deviate from those of Hasselmo and Schnell although we note that bulk stimulation is unlikely to be the reason since both groups used electrical stimulation of afferents in acute hippocampal slices. It is possible that the use of field potential recordings by Hasselmo and Schnell that necessarily incorporate elements of excitatory and inhibitory synaptic transmission may account for the discrepancies with our whole cell recordings that dissect effects on excitatory or inhibitory transmission.

7) Secondly, it is unclear why the temporoammonic input to the CA1 enjoys priority, because other subfields especially the CA3 (but also the dentate gyrus) also receive temporoammonic inputs. It remains elusive why a specific effect of acetylcholine would prioritize CA1 afferents over e.g. CA3 afferents at the network level in a behaving animal. Thinking further, the cholinergic impact on the temporoammonic-CA1 connection may not only have an effect on novel sensory information integration as solid data suggest a more broad function in other circuit computation associated with different aspects of memory formation and consolidation (see for example the Yamamoto and Tonegawa 2017 Neuron study about

engram replay). The reader may also wonder if the comparison of the effect of acetylcholine on CA3 recurrent collaterals with the temporoammonic afferents would provide stronger differences in light of the clearly important role of the CA3 recurrent network in the rapid integration of novel information into engrams (see among the numerous studies for example Wagatsuma et al 2018 PNAS). Finally, CCK-positive interneurons are also likely to be recruited by temporoammonic afferents and thereby generate feedforward inhibition in the CA3 (see for example Lasztocki et al 2011 Journal of Neuroscience).

The impact of acetylcholine more broadly within the hippocampal network is undoubtedly interesting and we have separate studies that directly address the role of acetylcholine within the CA3 network (see Prince et al 2018 BioRxiv). However, in this study we are focussed on addressing a specific hypothesis about how acetylcholine regulates the balance of TA and SC inputs to CA1 and we do not attempt to draw conclusions from the current data about the effects of acetylcholine in other areas of the hippocampus such as CA3 and dentate gyrus. To be clear, we are not stating that acetylcholine prioritises TA inputs to CA1 over TA inputs to CA3 or dentate gyrus.

We agree that the TA input provides more than just sensory input information. The recent paper from the Behrens and Burgess groups (Whittington et al 2020 Cell) outlines an interesting hypothesis for the interaction between entorhinal and hippocampal circuits and their functions and we now include additional discussion to reflect this (lines 561-568). We also thank the reviewer for pointing out the Lasztocki paper which we now include as further evidence that TA inputs can generate feedforward inhibition via CCK interneurons.

Issues with the second main finding:

8) The authors claim that CCK-positive interneurons mediate feedforward inhibition elicited by temporoammonic afferents, whereas Schaffer collaterals recruit parvalbumin-interneurons to trigger feedforward inhibition. This is wrong for various reasons. First of all, the presented experimental data do not show all or none differences just trends. Secondly, the slicing angle may impact differently the axonal and dendritic trees of the different interneuron types that may contribute to the trend-like small differences hence this type of interpretation from acute slice preparations is risky. Thirdly, most parvalbumin-positive basket and axo-axonic cells have extensive dendrites in the stratum lacunosum-moleculare that receive direct innervation from the entorhinal cortex (Halasy et al 1996, Hippocampus; Kiss et al 1996, Hippocampus). Fourthly, Glickfield and Scanziani 2006 Nature Neuroscience demonstrated that the two major interneuron populations are recruited by the same afferent inputs and both contribute to feedforward inhibition (the major difference lies in the temporal integration of their inputs). Fifthly, the DAMGO experiments showing a 51% versus 69% difference clearly suggest the involvement of parvalbumin interneurons in both forms of feedforward inhibition given the fairly selective expression of mu-opiate receptors.

We are unclear where the reviewer has formed this impression from, but the comments are a misrepresentation of our data and conclusions and in fact the statements in our manuscript are in agreement with the reviewer's comments. We state clearly that we engage a heterogeneous population of interneurons in feedforward inhibition for both input pathways. At no point do we claim homogeneous engagement of interneuron populations by either input pathway. What we show is that the TA pathway engages a higher proportion of CCK+ interneurons

and the SC pathway engages a higher proportion of PV⁺ interneurons. This agrees closely with observations from many other groups as described in the manuscript and in the citations given here by the reviewer and is therefore unlikely to arise from a specific slice angle. We completely agree that TA inputs will recruit some PV⁺ interneurons and we state this in the manuscript (lines 235-238) along with appropriate citations “*There are also minor components from other interneuron subtypes, most likely CCK⁺ basket cells in the SC pathway and PV⁺ or NPY⁺ interneurons in the TA pathway (Basu et al., 2013; Freund and Katona, 2007; Glickfeld and Scanziani, 2006; Klausberger and Somogyi, 2008; Milstein et al., 2015).*”

9) Importantly, the other distinction suggested by the authors between the role of perisomatically- and dendritically-targeting interneurons is also wrong. In both the CA1 and CA3 subfields, the CCK-interneurons - regardless of the specific subtype - project substantial dendritic arbors into the stratum lacunosum-moleculare (see Table 1 in Lasztozci et al 2011 Journal of Neuroscience and Klausberger et al 2005 Journal of Neuroscience).

This comment appears to confuse the distinction between the locations of synaptic input which occur on the dendritic arbors with synaptic outputs on axonal arbors. When we describe perisomatically- or dendritically-targeting interneurons we are referring to their axonal outputs – ie the pyramidal cell compartments that the interneuron forms synapses with. Both CCK⁺ and PV⁺ basket cells have axonal arbors that lie perisomatic to CA1 pyramidal cells but other CCK⁺ interneurons extend axons to more distal locations as shown in the studies cited by the reviewer. As described by the reviewer, the dendritic arbors of these interneurons can extend into stratum lacunosum moleculare but this is not relevant to their outputs. We show clear differences in IPSC kinetics between the two interneuron subtypes suggestive of differential targeting of output synapses to proximal and distal locations (or at least to locations with different GABA receptor subunit complements, see reviewer 3 point 3).

10) In addition, and most importantly, the authors used CCK promoter driven manipulations to dissect the two interneuron types. Unfortunately, the used mouse line turned out to be not reliable and various lab reported at interneuron conferences that PV-positive cells are also affected in these mice. Therefore, the authors should directly investigate by using immunostaining whether ChR2 is exclusively expressed in PV-positive cells in the first line and if ChR2 is restricted to CCK interneurons, but it is completely absent in PV neurons in the second mouse line. In case these lines are not specific enough (and I have serious doubts) then conclusion about the specific contribution of the different interneuron types cannot be drawn.

Despite the fact that CCK⁺ and PV⁺ interneurons arise from separate populations of ganglionic precursor cells (Tricoire et al 2011 J Neurosci; Lim et al 2018 Neuron), the Allen Brain atlas reports some overlap in CCK and PV expression for some interneurons in visual cortex and there is a recent report of a subset of PV expressing cells in PFC that also express CCK (Nguyen et al 2020 J Neurosci). Evidence from the hippocampus suggests limited overlap in expression (Harris et al 2018 PLoS Biol) but highlights that for all interneuron subtypes there is to some degree a continuum of gene expression patterns across the classified subtypes that form the basis of cre-dependent manipulations. This is also highlighted in

recent transcriptomic analyses of interneuron subtypes (Gouwens et al 2020 Cell; Scala et al 2020 Nature). Despite this, cre expression linked to specific promoters has successfully separated distinct interneuron subtypes in many studies. Specific to the CCK-cre and PV-cre mouse lines relevant to the hippocampus, non-overlapping immunostaining for ChR2 in CCK-cre and PV-cre mice has been demonstrated (Basu et al 2013 Neuron) and this study also showed functional distinction between IPSCs elicited by optogenetic stimulation in CCK-cre and PV-cre mice. Similarly, we demonstrate that use of the PV-cre and CCK-cre mouse lines produced functionally distinct IPSCs in terms of IPSC kinetics and the effect of acetylcholine on short-term plasticity. It is of course always formally possible that some non-specific cre expression is present but, importantly, we are able to separate 2 functionally distinct inhibitory pathways using these mouse lines.

Interestingly, the major off-target cre expression in the CCK-cre mouse line is reported in pyramidal cells not in PV+ interneurons (Taniguchi et al 2011 Neuron; Nguyen et al 2020 J Neurosci). We specifically controlled for this in our experiments (see lines 243-246 and Figure S3).

11) Finally, a general conceptual problem arises from the fact that the major interneuron type that controls the temporoammonic pathway is the O-LM interneuron. As shown by the Scanziani lab and others, these cells are readily activated by cholinergic activity via their specific alpha-2 nicotinic cholinergic receptors. Thus, optogenetic activation may trigger their feedback control in most experiments (except the case when the nicotinic receptor antagonists was applied. This further complicates the picture how distinct interneuron types and their control by acetylcholine would differentiate between efficacy of the temporoammonic and the Schaffer collateral inputs to recruit feedforward inhibition.

We are fully aware of the role that OLM feedback interneurons play in the hippocampal network and have studied this extensively (see eg Udakis et al Nat Comms 2020) and we discuss this in the manuscript (line 488). The role of cholinergic activation of OLM interneurons is very interesting and in fact these can be directly activated by both nicotinic and muscarinic receptors (although muscarinic appear to be the most important in vivo (Lovett Barron et al Science 2014)). Since OLM interneurons target the very distal dendrites of CA1 pyramidal cells where the TA inputs occur excitation of OLM interneurons would be expected to decrease the effectiveness of TA inputs but not SC inputs. Our results show the opposite and furthermore we show that the effects of acetylcholine on both excitatory and inhibitory TA inputs are mediated by M3 receptors localised to presynaptic TA inputs. Both these pieces of evidence suggest that in our experiments feedback inhibition by OLM cells is not a major component of the inhibition we measure.

In addition to the point raised here, there are many other nodes within the hippocampal network where acetylcholine may act (eg outlined in lines 492-502 of the discussion) and therefore we do not presume to provide a complete picture for cholinergic modulation of the hippocampal circuit. But by carefully controlling our experiments using specific genetic and pharmacological tools we dissect a specific mechanism by which acetylcholine regulates synaptic inputs to CA1 pyramidal cells and their resulting output.

Issues with the third main finding:

12) The authors propose that presynaptic M3 muscarinic receptors control the temporoammonic pathway, whereas M4 receptors regulate the Schaffer collaterals. The role of M4 has already been published and Sanchez et al 2009 Journal of Neuroscience Research should be cited in this regard. Moreover, I am not convinced that M3 is presynaptically located on the entorhinal afferents. According to the Allen Brain Atlas, the expression of M3 receptor is very high in CA1 pyramidal neurons, but very modest in the entorhinal cortex indicating a postsynaptic function for M3 in pyramidal neuron dendrites. M3 activity is coupled to Gq/11 signaling that is usually a somatodendritic signaling mechanism. In contrast, M4 activity is coupled to Gi/o signaling that is indeed ideal for a presynaptic function i.e. the inhibition of neurotransmitter release. The authors based their arguments of paired-pulse ratio measurement to draw conclusion. However, they failed to consider the possibility that postsynaptic M3 muscarinic receptors may readily trigger an indirect retrograde messenger that could act presynaptically. In accordance, Fukudome et al (2004, European Journal of Neuroscience) showed that endocannabinoids are released by somatodendritic M3 activation and remove inhibition of pyramidal neurons via presynaptic type I cannabinoid receptors located on GABAergic inputs. This indirect local mechanism could fully explain the authors findings on the increased CA1 output without any consideration of differential recruitment of feedforward inhibition (which is doubtful see above). Moreover, under in vivo conditions, endocannabinoids will likely be released by those pyramidal neurons that are very active (i.e. part of the engram) thus cholinergic effects via somatodendritic M3 receptors and retrograde endocannabinoid-mediated suppression of inhibition could readily facilitate the integration of novel external information with the pre-existing internal representations.

This is an interesting point. Our data are the first demonstration of a role for M3 receptors in regulating excitatory synaptic transmission in the hippocampus and formally it is possible that this effect could be mediated by a retrograde messenger triggered by postsynaptic M3 receptor activation. We have now included additional experiments to test the locus of M3 receptors. The first of these is to include G-protein inhibitors in the patch pipette (Figure 5). By blocking the effects of Gq coupled receptors specifically in the postsynaptic neuron we show that the effects on synaptic transmission remain arguing in favour of a presynaptic locus of M3 receptors. The second and third experiments block potential retrograde or spillover messengers triggered by postsynaptic M3 receptors. For the effects on excitatory TA input the most likely transmitter is GABA acting on presynaptic GABA_B receptors. In this case acetylcholine might modulate GABA release by either directly exciting interneurons or indirectly regulating GABA release by generating a retrograde messenger such as endocannabinoids. Therefore, we performed experiments in the presence of a GABA_B receptor antagonist CGP55845. We found no effect of CGP on the cholinergic depression of excitatory TA transmission ruling out a role for GABA spillover. As the reviewer points out there is evidence that muscarinic receptor activation can result in the production of endocannabinoids which can act at presynaptic CB1 receptors to reduce release probability and increase PPR and these endocannabinoid actions are known to occur at inhibitory synapses from CCK interneurons to pyramidal cells (Wilson et al 2001 Neuron). However, this is unlikely to be the mechanism in our experiments since the reduction in CCK transmission did not change PPR (Figure 3 but see also response to reviewer 3 point 4). Nevertheless, we also performed experiments with the CB1 receptor antagonist AM251 to test a role for endocannabinoid production and retrograde transmission in TA pathway synaptic transmission. In the presence of AM251 carbachol still reduced synaptic

transmission and increased PPR substantially but the reduction in synaptic transmission was slightly less than in control conditions (Figure 5). We conclude that although there may be a minor role for endocannabinoids, the majority of the cholinergic effects we find are not mediated by this mechanism. Taken together these additional experiments confirm our original conclusions that M3 receptors are located presynaptically and reduce the probability of neurotransmitter release.

There are additional lines of evidence that suggest M3 receptors are presynaptically located on TA fibers. Firstly, M3 receptor mRNA is expressed in the entorhinal cortical cells that project to CA1 in the TA pathway (Allen Brain Atlas) and M3 receptor protein is specifically localised to stratum lacunosum moleculare in CA1 where TA inputs terminate but exhibits much lower expression in other layers (Levey et al 1995 J Neurosci). Secondly, we have data not currently included in the manuscript showing that M3 receptors depress TA synaptic transmission onto feedforward CCK+ interneurons. This shows that M3 receptor-mediated depression happens selectively in the TA pathway at both excitatory and feedforward inhibitory synapses in CA1. The common feature of these synapses are the presynaptic TA fibers. Thirdly, there is precedent for a role for Gq coupled receptors regulating presynaptic function, for example group 1 metabotropic receptors (eg. Vergassola et al 2018 Front Mol Neurosci, Gerachshenko et al Nat Neurosci 2005). In addition to the new data, we now include discussion of these mechanisms (lines 515-524), the Fukudome reference for muscarinic activation of endocannabinoid signalling and the Sanchez reference for M4 receptors suggested by the reviewer.

Reviewer 2:

The article from Palacios-Filardo and other address the effect of ‘physiological’ ACh release in 2 different hippocampal pathways. The authors explore paradigms widely accepted that have little experimental evidence (e.g. dichotomy in septal glutamatergic and cholinergic neurons in theta or memory). The article is very interesting and is timely and in my opinion should be published in ‘prestige’ journals in order to reach a broader audience. I have however, a few comments that are not ‘major’ but are necessary for my acceptance:

1) Lines 99-101:

the same distal dendritic regions as the TA pathway, are directly excited by acetylcholine (Leao et al., 2012; Pouille and Scanziani, 2004). This indicates that cholinergic modulation of inhibition within the hippocampal circuit strongly dictates excitatory input integration and CA1 output

please remove the ‘strongly’. The nicotinic response in OLM neurons is only strong if there is a very strong coordination of MS cholinergic neurons. Compared to local glutamatergic excitation, cholinergic EPSCs are small.

Line 512:

Again, at least in Leao et al 2012, the hypothesis was that if there would be enough cholinergic excitation through nicotinic receptors, ACh would then excite the OLM population and therefore modulate gating. No muscarinic effect was explored in that article.

Nevertheless, a recent in vivo study does suggest that septal ACh excite OLM cells (Mikulovic et al, Nat Comm 2018).

This is a good point about the relative efficacy of cholinergic vs glutamatergic excitation of OLM cells. The sentence is now revised and we also include reference to Mikulovic et al 2018.

Whole cell studies involving metabotropic receptors are still not widely accepted (dialysis, channel runout etc). I personally believe that these studies are valid and the g protein runout can be manageable if small pipettes are used with low ATP concentration in the internal solution (Fiszman et al 1991, Verrecchia et al 1999). Nevertheless, the authors should discuss the caveats of their methods in studying ACh effects in pathways using whole cell.

We do use low ATP concentrations in our whole cell solution. In addition, we observe robust G-protein coupled receptor effects in our experiments – most obviously the M1 receptor mediated depolarisation and increase in input resistance in our whole cell recorded CA1 pyramidal neurons. Our new experiments in Figure 5 show this most clearly where a G-protein inhibitor included in the patch pipette completely blocks the muscarinic effects on input resistance and membrane potential. This positive control and the fact that we have previously observed very similar results in experiments using perforated patch recordings (Isaac et al., 2009 J Neurosci) indicates that washout is not a major factor in our experiments. In addition, our key results are dependent on effects attributable to G-protein coupled receptors expressed in other neurons to the recorded neuron where washout is not an issue. Taken together, we do not believe washout is a factor in our study.

In my opinion, the only weak point in the article is the lack of experiments involving HCN pharmacology. ACh strongly modulate HCN channels (Zhao et al. Front Cell Neurosci. 2016). These channels have an obvious role in short term plasticity and internal ATP can differentially affect HCN subunits Pian et al J Gen Physiol 2006). Ideally, the authors could add a voltage clamp experiment using hyperpolarising steps (to trigger Ih) concomitant to light stimulation of Chat cells to show if cholinergic modulation of Ih could play any role in modulating synaptic currents. Or, test TA vs SC inputs while applying ZD7288 and optical stimulation. Or at least discuss whether or they believe Ih activation/inactivation could play a role in their interpretation.

This is an interesting point and in theory modulation of HCN could play an important role since HCN channels are differentially expressed across the dendritic arbour of CA1 pyramidal neurons and control dendritic integration (Magee 1999, Nat Neurosci). However, the modulation of HCN channels is via cAMP signalling and therefore Gi/o. Indeed, this is the mechanism in striatal interneurons reported in the cited paper (Zhao et al 2016 Front Cell Neurosci). CA1 pyramidal neurons express M1 and M3 receptors which couple to Gq G-proteins and do not express M2 and M4 receptors that couple to Gi/o so it is unlikely that HCN channels are modulated by muscarinic receptors in CA1 pyramidal cells. This conclusion is further supported by data showing an occlusion of muscarinic effects on synaptic integration by the SK channel blocker apamin indicating no role for cholinergic

regulation of HCN (Buchanan et al 2010 Neuron). We now include mention of these considerations for HCN channels in the discussion (lines 492-502)

Reviewer #3 (Remarks to the Author):

This is a very nice manuscript by Palacios-Filardo and colleagues that shows convincingly a role for differential cholinergic regulation of feedforward excitatory and inhibitory drive within the CA1 hippocampus. Using a combination of optogenetics and electrophysiology they clearly demonstrate that endogenous release of Ach or exogenous application of Ach ligands alters the nature of both excitatory and inhibitory drive onto the CA1 circuit. Using a number of clever manipulations they show that while Ach modulates excitatory inputs onto CA1 pyramidal cells from the entorhinal and CA3 subfield in a similar manner, disynaptic inhibition arising through the temporoammonic pathway is preferentially reduced compared to inhibition arising through Schaffer collateral inputs. The consequence of this is to prioritize entorhinal drive onto CA1 pyramids, suggesting an underlying mechanism for the shift in afferent drive observed during novel memory formation. As stated above this is a very nice piece of work however it suffers somewhat from being quite dense and difficult to follow. I work in this field and I found it hard to navigate and keep all of the parts in my head while trying to understand the manuscript. The authors are urged to either present the data in a slightly more straightforward manner and also provide a cartoon/schematic that highlights the main findings of the manuscript.

We appreciate that some of the experiments are complex due to the number of pathways and inclusion of both excitatory and inhibitory feedforward pathways. We have already included schematics in each figure and colour codes for the results to guide the reader and we have now sought to further simplify the presentation of results. We also now provide a graphical abstract to summarise and highlight the main findings as suggested.

1. My biggest issue with the manuscript is the “bait and switch” approach used in Figure 1. In figure 1 The authors present a beautiful set of data obtained from a mouse line that has Chr2-encoded in CHAT-positive cells. They convincingly demonstrate that they can monitor released Ach and that it successfully modulates excitatory synaptic transmission in a slice preparation. This is a significant achievement in itself. However the rest of the manuscript resorts to using bath applied cholinergic agents. Although the data shown in Figure 2 are compelling one cant help but wonder why the authors had gone to the trouble of setting up the previous experiments illustrated in Figure 1 using intrinsic Ach release via optogenetics and then for some reason they resort to bath application of carbachol. Given the data in figure 1 showing that the authors have reliably established that they can release endogenous Ach in their system, was there a compelling reason to resort back to bath applied CCh? I would urge the authors to repeat this (and other significant) expt using the experimental set up used in figure 1 since bath application of the agonist removes all of the temporal dependence of Ach released onto different targets, which will undoubtedly play into the overall response pattern observed. The ability to successfully release Ach optogenetically and show it can modulate the circuit is outstanding but I feel that the rest of the manuscript undercuts the true potential of this observation

There is a strong rationale for the use of carbachol for the mechanistic investigation shown in Figures 3 and 4 (and new Figure 5): (i) the dissection of interneuron subtypes (Figure 3) requires optogenetic stimulation of specific interneurons that are extremely difficult to combine with optogenetic stimulation of acetylcholine (there is too much overlap between opsin excitation spectra with the tools currently available); (ii) the determination of receptor subtypes (Figure 4) requires transgenic knockouts and specific agonists and antagonists, again largely incompatible with optogenetic stimulation of acetylcholine; (iii) new data in Figure 5 requires the use of bath applied carbachol to provide the positive control for postsynaptic effects of Gq coupled signalling. We now provide justification for the carbachol approach in the results section (lines 191-193). Since these experiments require the use of bath applied carbachol we felt it was necessary to provide evidence that validates the equivalence of optogenetic endogenous release and exogenous agonist application to activate cholinergic receptors. We appreciate that this evidence was insufficient in the first submission and we now include further data in Figures 2 and 7 using optogenetic stimulation of acetylcholine release to complement the data in Figures 1 and 6. Importantly, these new data replicate data obtained using carbachol application (now moved to supplementary data or juxtaposed in Figure 2) and therefore we are confident that the two approaches are largely equivalent in the context of these experiments. Furthermore, we have also repeated the key pharmacological experiments in Figure 4 identifying M3 receptors as the key receptor subtypes modulating the TA pathway using optogenetic acetylcholine release.

2. Figure 3 and related text. The authors should provide clarification that both CCK+ and PV+ cells follow the 5x 10Hz light stimuli faithfully used to evoke the train of IPSCs studied in this expt.

Data showing that PV+ and SST+ cells faithfully follow 10 Hz light stimulation is shown in supplementary Figure 1 of another manuscript from our group (Udakis et al Nature Communications 2020). In the current manuscript we also show that ChAT+ cells follow 10 Hz stimulation. We do not have data specifically showing that CCK+ cells follow 10 Hz stimulation but all data from previous publications (eg Nguyen et al 2020 J Neurosci; Basu et al 2013 Neuron) and the fact that all other Chr2 expressing cells that we have recorded follow this frequency faithfully indicates that they do. We now include text to this effect (lines 246-248).

3. Line 254 it is not correct to assume that the difference in decay kinetics reflect simply the somatic versus dendritic location of the inhibitory inputs. There are as many CCK-basket cells located in s. radiatum as there are dendritic targeting CCK cells. The kinetics may simply reflect the kinetics of the underlying GABA receptors at the different synapses.

The difference in decay kinetics shows we are targeting separate populations of GABAergic synapses and it is true that it could arise from different locations or GABA receptor subtypes. We felt it was fair to state that the slower kinetics supports a conclusion that they target more distal dendritic regions but we have now included the possibility that the kinetics may also or alternatively result from different GABAA receptor composition (lines 227-228).

4. Figure 3 and related text. How do the authors interpret the lack of change of release probability caused by CCh bath application in the CCK cell cohort? Is this a consequence of muscarinic receptor activation on pyramidal cells shunting the IPSC? How do the authors rule out an indirect effect of CCh on the pyramidal cell in this dataset?

This is an interesting point. We believe we can rule out an indirect effect of carbachol on the pyramidal cell with the new data presented in Figure 5 coupled with the fact that the reduction in feedforward inhibition is mediated by M3 receptors (shown in Figure 4). The data in Figure 5 indicate these receptors are presynaptic (see response to reviewer 1, point 13) and so since the reduction in feedforward inhibition is prevented by blocking presynaptic receptors, this removes the possibility that postsynaptic receptors are shunting IPSCs. In addition, muscarinic receptor activation increases input resistance which would be expected to reduce rather than increase shunting. Our best explanation for why there is no change in PPR in the presence of carbachol is an interplay between enhancing excitability of CCK interneurons and inhibiting release of GABA. The latter is expected to increase PPR (similar to PV synaptic transmission) whereas the former might be expected to decrease PPR (eg Alle and Geiger 2006 Science). M3 receptors are expressed by CCK+ interneurons where they enhance excitability (Cea-del Rio et al 2010 and 2011 J Neurosci) so this potentially provides a mechanism for M3 specific modulation of CCK mediated inhibition in addition to presynaptic effects. This combined action for M3 receptors is included in the discussion (lines 478-491).

5. In both figure 5 and figure 7 the initial spike probability appears to be set higher for the SC input in control conditions than the TA input. This may have a potential impact in the outcome of the response seen after light driven Ach response or the CCh response. In fact this would have a major influence on the time to first spike dataset since the SC input has a higher probability of firing in the first few spikes compared to the TA input dataset. Why were the initial spike probabilities not set to be equivalent in both inputs?

We have now performed further analysis to address this important point as supplementary Figure S7. We aim to set the spike probabilities equivalent in each pathway but because the TA inputs target more distal dendrites, they have a generally lower efficacy for spike generation. Therefore, on average they had a lower initial spike probability, although there is considerable overlap in the distributions. To test whether this was a factor in the observed results we have now analysed the distributions of spike probability changes and correlated these with the initial spike probabilities for each pathway in each experiment. This revealed that the amount of spike probability change was indeed correlated with initial spike probability, but the direction of change was not. The same was true for the time to first spike analysis. Therefore, the conclusion that acetylcholine reprioritises CA1 responses to TA inputs is still valid.

Minor comments

Lines 30-32 “Underpinning the prioritization....” This sentence is a little clumsy and should be reworded

The abstract is now substantially reworded for clarity.

Line 71-73 please provide a reference for this prediction

This is based on the work of Michael Hasselmo and we insert a reference to indicate this.

Figure 1 panel E. please clarify the nature of the electrically evoked events seen in the panel. Their identity is left ambiguous to the reader.

These events are now identified clearly in the figure legend.

REVIEWER COMMENTS

Reviewer #1 (Remarks to the Author):

The authors have addressed some of my comments by adding new sentences in the Discussion and performing an additional experiment. The manuscript has been improved in clarity.

However, I still have major concerns about the interpretation of some of the experimental data.

1. On the role of presynaptic M3 receptors. The authors report "TA evoked EPSCs recorded from CHRM3 KO slices were reduced by CCh with an associated increase in PPR (Figure 4I-K; EPSC, $68 \pm 4\%$, $n = 20$ from 11 mice" in the M3 KO mice. Notably, when they block retrograde endocannabinoid signaling they obtain similar result "(Figure 5F-H; AM251 EPSC $65 \pm 4\%$). Apparently, the absence of muscarinic M3 receptors have similar effect as the blockade of cannabinoid CB1 receptors. These data are consistent with the possibility that somatodendritic M3 receptor activation in pyramidal cells, and the downstream triggering of retrograde endocannabinoid signaling acting via presynaptic CB1 receptors located on TA axon terminal would be a plausible scenario to fully explain the electrophysiological data without the presence of presynaptic M3 receptors on TA axons.

To support their original conclusion about presynaptic M3 receptors on TA afferents, the authors performed new experiments and added new arguments. However, there are some caveats with these data and arguments. The intracellular administration of GDP- β -S via the patch pipette is appreciated. However, it is very difficult to determine if this compound could readily reach the distant and thin secondary or tertiary dendrites in the stratum lacunosum-moleculare. The change in depolarization and input resistance measured via a somatic electrode may not necessarily represent an alteration in these parameters at the distant dendritic tips. In addition, the expression of M3 receptor mRNA in the entorhinal cortex shown in the Allen Brain Atlas may also represent dendritic M3 receptors at the protein level in neurons of the entorhinal cortex. In agreement, CA1 pyramidal neurons exhibit the highest expression of M3 receptors at the mRNA levels in the forebrain hence these neurons could readily carry somatodendritic M3 receptors. In fact, the authors use the Levey et al 1995 Journal of Neuroscience paper to support their argument "M3 receptor protein is specifically localised to stratum lacunosum moleculare in CA1 where TA inputs terminate but exhibits much lower expression in other layers (Levey et al 1995 J Neurosci)." However, what Levey et al states is the following "m3 is enriched in pyramidal neurons, the neuropil in stratum lacunosum-moleculare". I kindly ask the authors to have a careful look at Figure 2 in Levey et al. This Figure shows that the M3 receptor protein is present in the cell bodies and proximal dendrites. Faint dendrites in the stratum radiatum run in parallel towards the stratum lacunosum-moleculare. Most importantly, if one zooms into the Figure, the profiles are more reminiscent of thin dendrites than individual axon terminals (that would look like individual granules at the light microscopic level). This immunostaining pattern is more consistent with the possibility that these profiles are M3-immunostained dendrites extensively branching in the stratum lacunosum-moleculare. Moreover, I agree with Reviewer 2 that the involvement of HCN channels should also be considered and I would like to remind the authors that somatodendritic HCN1 immunostaining (see Lorincz et al 2002 Nature Neuroscience) has an almost identical distribution to the M3-immunostaining shown in Levey et al 1995. In conclusion, I am not convinced that there are presynaptic M3 receptors on the TA afferents, and the presented experiments together with available literature do not exclude that M3 receptors are located on the distal dendrites of CA1 pyramidal neurons.

2. On the usefulness of the CCK-Cre mice. Based on the experience of our lab and the observations from other labs, I again would like to emphasize that the authors should be very careful with the interpretation of the cell-type-selectivity of any CCK promoter-driven approach. In addition, I checked again the Basu et al 2013 Neuron paper that the authors refer to. In contrast to the statement of the authors, this study did not provide direct experimental evidence that the CCK promoter is selective for the so-called CCK interneurons. In Figure 4, Basu et al presented a low-magnification image about the general EYFP distribution. However, Basu et al did not do double labeling experiments and magnifying their image in Figure 4, the widespread distribution of numerous cell bodies is apparent together with the pyramidal neuron "background". I again recommend the authors to provide a direct quantitative experimental evidence that Chr2-EYFP expression is restricted to the CCK interneurons, whereas PV cells or other interneuron types do not express their optogenetic tool. While there may be distinct kinetics upon their optogenetic stimulation, this does not necessarily mean that their strong conclusion (PV versus CCK cells) is

true. Other interneuron types or in worst case even optogenetic activation of the dendritically-targeting PV-positive bistratified neurons could have contributed to their measurements. Verification of their animal model is required if the authors would like to keep their statement as take-home message "The differential effect of acetylcholine at PV+ and CCK+ synapses provides a mechanism for the enhancement of TA pathway excitatory-inhibitory ratio in comparison to SC pathway."

3. In light of these issues above, I had the impression that the cellular and subcellular specificity of the perturbation approaches and the data evaluation may require further attention to exclude potential mistakes. Please note that for example in Figure 1B, it is described in the legend that blue cells are the neurobiotin-filled cholinergic neurons, whereas ChR2-YFP proteins are shown in yellow. If ChR2 is expressed exclusively by cholinergic neurons, then one would expect that all blue cells should contain the yellow signal. However, there is an apparent mismatch in the image between yellow and blue, these cells are located adjacent to each other. This does not make sense. Please provide quantitative anatomical evidence that Cre-mediated expression in the medial septum is restricted to ChAT neurons, but it is not present in adjacent GABAergic neurons.

4. Similarly, in Figure 1C, the depicted neuron in light blue is a large multipolar interneuron based on its morphology. CA1 pyramidal cells have one thick apical dendrite with secondary dendrites first branching about 200 micron away from the cell body (see for example Figure 1 in Papp et al 2001 Neuroscience), basal dendrites emerge as a tuft. If the authors misidentified the neuron they use for presentation, how the reader can trust in their data when they report cell-type-specific parameters obtained from patch-clamp recordings? I suggest the authors adding further morphological or neurochemical verification of the specific cell-type they recorded from.

5. Finally, I refer again to the original review point 8 and 9, because the authors misunderstood my comments. In the revised manuscript, the authors state "Feedforward interneurons in the SC pathway are primarily perisomatic targeting basket cells expressing parvalbumin (PV+) or cholecystinin (CCK+) whereas the mediators of feedforward inhibition in the TA pathway are likely dendritically targeting CCK+ or neuropeptide Y (NPY+) expressing interneurons." If both perisomatically- and dendritically-targeting CCK interneurons have dendrites in the stratum lacunosum-moleculare (Klausberger et al 2005) then why the authors expect their different recruitment by the temporo-ammonic pathway? In addition, why most parvalbumin-positive basket and axo-axonic cells have extensive dendrites in the stratum lacunosum-moleculare that receive direct innervation from the entorhinal cortex (Halasy et al 1996, Hippocampus; Kiss et al 1996, Hippocampus). My overall fear is that the slicing angle used in the authors' lab differentially affects the dendritic trees of the different interneuron types and this leads to a major and perhaps wrong conclusion of the authors about the different recruitment and involvement of the distinct cell types ("interneuron subpopulations") by the different pathways. Could this "higher proportion" (the toning down term used in the rebuttal letter, but not in the paper) of contribution of the distinct cell types be due to a methodical issue? Would it be possible to exclude this possibility by using another slicing angle?

Overall, I hope I could convince the authors that they should reconsider that two major conclusions described in the Abstract as "Entorhinal and CA3 pathways engage distinct feedforward interneuron subpopulations and cholinergic modulation is mediated differentially by presynaptic muscarinic M3 and M4 receptors respectively" may not necessarily be correct based on the presented data and the existing literature.

Reviewer #2 (Remarks to the Author):

The authors substantially reviewed the paper and the additional experiments strengthen their conclusions. I am particularly happy with new fig 5. The authors have addressed my concerns and I believe the article is suitable for publication.

Reviewer #3 (Remarks to the Author):

I am happy with the revisions made by the authors they represent a conscientious attempt to address many of the difficult issues raised by the reviewers. I have no further comments

Response to reviewer's comments (comments in black, responses in blue)

Reviewer #1 (Remarks to the Author):

The authors have addressed some of my comments by adding new sentences in the Discussion and performing an additional experiment. The manuscript has been improved in clarity.

However, I still have major concerns about the interpretation of some of the experimental data.

1. On the role of presynaptic M3 receptors. The authors report "TA evoked EPSCs recorded from CHRM3 KO slices were reduced by CCh with an associated increase in PPR (Figure 4I-K; EPSC, $68 \pm 4\%$, $n = 20$ from 11 mice" in the M3 KO mice. Notably, when they block retrograde endocannabinoid signaling they obtain similar result "(Figure 5F-H; AM251 EPSC $65 \pm 4\%$). Apparently, the absence of muscarinic M3 receptors have similar effect as the blockade of cannabinoid CB1 receptors. These data are consistent with the possibility that somatodendritic M3 receptor activation in pyramidal cells, and the downstream triggering of retrograde endocannabinoid signaling acting via presynaptic CB1 receptors located on TA axon terminal would be a plausible scenario to fully explain the electrophysiological data without the presence of presynaptic M3 receptors on TA axons.

If one considers all the data we do not believe this is an appropriate conclusion or the correct comparison to make. We show in the manuscript that the CB1 receptor antagonist only reduces the effect of carbachol on TA pathway synaptic transmission amplitude by a small amount (carbachol $43 \pm 5\%$ vs carbachol + AM251 $65 \pm 4\%$) and does not prevent the increase in PPR. This is not commensurate with a conclusion that activation of presynaptic CB1 receptors can fully explain the electrophysiological data. Instead, the conclusion is that presynaptic CB1 receptors may explain a minor component of the effect of carbachol. In contrast, the data showing a complete block of the carbachol effect with the M3 receptor antagonist DAU 5884 (Figure 4) indicate that blockade of M3 receptors completely prevents the effects of carbachol and acetylcholine on TA pathway transmission. The partial blockade of the carbachol effect in the M3 KO mice is because there is compensation by M1 receptors (Supplementary figure S5). Therefore, taken together these data indicate the major component of the carbachol effect on TA pathway synaptic transmission is best explained by the presence of presynaptic M3 receptors.

To support their original conclusion about presynaptic M3 receptors on TA afferents, the authors performed new experiments and added new arguments. However, there are some caveats with these data and arguments. The intracellular administration of GDP- β -S via the patch pipette is appreciated. However, it is very difficult to determine if this compound could readily reach the distant and thin secondary or tertiary dendrites in the stratum lacunosum-moleculare. The change in depolarization and input resistance measured via a somatic electrode may not necessarily represent an alteration in these parameters at the distant dendritic tips.

This is a good point that we can address with additional data demonstrating that in our experiments small molecules diffuse to the distal dendrites within the timeframe of our experiments. The data below shows a CA1 pyramidal neuron that has been patched and filled with Alexa594, a small molecule fluorescent dye. Using 2photon microscopy we can measure the fluorescence intensity over time at the distal dendrites indicated. This shows that a stable plateau is reached after ~10 minutes indicating full diffusion of the molecule from soma to distal dendrite in this timeframe. We expect the diffusion of GDP- β -S to be faster since it has a lower molecular weight than Alexa594 (459 vs 820). In our experiments shown in Figure 5 we always waited at least 15 minutes after whole cell break in before applying carbachol and therefore it is reasonable to assume that the GDP- β -S was present at the most distal dendrites and effective at blocking G-protein mediated signalling. We now state the length of time after break in before application of carbachol that allows the diffusion in the methods (line 645).

In addition, the expression of M3 receptor mRNA in the entorhinal cortex shown in the Allen Brain Atlas may also represent dendritic M3 receptors at the protein level in neurons of the entorhinal cortex. In agreement, CA1 pyramidal neurons exhibit the highest expression of M3 receptors at the mRNA levels in the forebrain hence these neurons could readily carry somatodendritic M3 receptors. In fact, the authors use the Levey et al 1995 Journal of Neuroscience paper to support their argument “M3 receptor protein is specifically localised to stratum lacunosum moleculare in CA1 where TA inputs terminate but exhibits much lower expression in other layers (Levey et al 1995 J Neurosci).” However, what Levey et al states is the following “m3 is enriched in pyramidal neurons, the neuropil in stratum lacunosum-moleculare”. I kindly ask the authors to have a careful look at Figure 2 in Levey et al. This Figure shows that the M3 receptor protein is present in the cell bodies and proximal dendrites. Faint dendrites in the stratum radiatum run in parallel towards the stratum lacunosum-moleculare. Most importantly, if one zooms into the Figure, the profiles are more reminiscent of thin dendrites than individual axon terminals (that would look like individual granules at the light microscopic level). This immunostaining pattern is more consistent with

the possibility that these profiles are M3-immunostained dendrites extensively branching in the stratum lacunosum-moleculare. Moreover, I agree with Reviewer 2 that the involvement of HCN channels should also be considered and I would like to remind the authors that somatodendritic HCN1 immunostaining (see Lorincz et al 2002 Nature Neuroscience) has an almost identical distribution to the M3-immunostaining shown in Levey et al 1995. In conclusion, I am not convinced that there are presynaptic M3 receptors on the TA afferents, and the presented experiments together with available literature do not exclude that M3 receptors are located on the distal dendrites of CA1 pyramidal neurons.

We agree with the reviewer that M3 receptors are expressed on CA1 pyramidal neurons. The immuno staining shown in Levey et al 1995 indicates presence of M3 receptors on the soma and proximal dendrite regions with very low levels in stratum radiatum. But we have shown that these receptors are unlikely to underlie the effects on presynaptic release at TA pathway synapses that we describe. Furthermore, the higher resolution pictures shown in figure 4 of Levey et al indicate that the strong M3 staining in the stratum lacunosum moleculare is punctate (Levey's description) in contrast to the more string like expression pattern near the soma. This might suggest presence in presynaptic terminals of the TA pathway, but it is not possible to be certain. From the pictures, M3 receptors could be located on dendrites or presynaptic terminals or both. However, we note that the expression pattern is very similar to that seen for M4 receptors in stratum radiatum shown in Levey et al Figure 5 which are known to be located at presynaptic Schaffer collateral terminals. A proper answer to this question would require immunogold labelling and high-resolution microscopy which has only to our knowledge been attempted for the M1 receptors in the hippocampus (Yamasaki et al 2010 J Neurosci). It is also shown in Allen Brain Atlas that M3 receptors are expressed by the cells of entorhinal cortex that project to CA1 in the TA pathway but there is limited information on where those M3 receptors are located on the cell membrane. Again, it is likely that they are on somatic and dendritic membranes but they could also be present on axons and termini. Therefore, the immuno staining is entirely consistent with M3 localisation on presynaptic terminals of the TA pathway. When taken together with our electrophysiological data it is clear that presynaptic release probability is reduced by activation of M3 receptors and our data in Figure 5 support the conclusion that M3 receptors are located presynaptically rather than acting indirectly.

We do not believe HCN channels are relevant here for the reasons given in response to the comment made by reviewer 2 in the previous round of review.

2. On the usefulness of the CCK-Cre mice. Based on the experience of our lab and the observations from other labs, I again would like to emphasize that the authors should be very careful with the interpretation of the cell-type-selectivity of any CCK promoter-driven approach. In addition, I checked again the Basu et al 2013 Neuron paper that the authors refer to. In contrast to the statement of the authors, this study did not provide direct experimental evidence that the CCK promoter is selective for the so-called CCK interneurons. In Figure 4, Basu et al presented a low-magnification image about the general EYFP distribution. However, Basu et al did not do double labeling experiments and magnifying their image in Figure 4, the widespread distribution of numerous cell bodies is apparent together with the pyramidal neuron "background". I again recommend the authors to provide a direct quantitative experimental evidence that Chr2-EYFP expression is restricted to the CCK

interneurons, whereas PV cells or other interneuron types do not express their optogenetic tool. While there may be distinct kinetics upon their optogenetic stimulation, this does not necessarily mean that their strong conclusion (PV versus CCK cells) is true. Other interneuron types or in worst case even optogenetic activation of the dendritically-targeting PV-positive bistratified neurons could have contributed to their measurements. Verification of their animal model is required if the authors would like to keep their statement as take-home message “The differential effect of acetylcholine at PV+ and CCK+ synapses provides a mechanism for the enhancement of TA pathway excitatory-inhibitory ratio in comparison to SC pathway.”

We appreciate the concern surrounding the specificity of CCK expression and we are fully aware of the potential for confusion. In our study it is important to focus on the key observation that the use of PV-cre and CCK-cre mouse lines produced functionally distinct IPSCs that map onto the properties we observe for feedforward inhibition in the SC and TA pathways. The fact that we use both mouse lines *and make a comparison between them* reduces most of the problems highlighted by the reviewer. In contrast, if we were attempting to make strong statements about interneuron subtypes based on using just the CCK-cre line the reviewers’ comments would be fully justified. The comparison means that even if the interneuron populations have some overlap due to co-expression of PV and CCK in a subset of interneurons, the differential results we show are driven by the interneurons that do not co-express PV and CCK. The critical point is that we are making a comparison between the two subtypes and so the issue of whether CCK is expressed in some PV interneurons is less important, save to reduce the magnitude of the observed differences.

This means that quantification of PV and CCK co-expression is not critical for our conclusions. Furthermore, it is not expected that double labelling of ChR2-EYFP and PV in the CCK-ChR2 mice will reveal any meaningful data. This is because CCK is expressed in CA1 pyramidal cells (see Supplementary Figure 3; Basu et al 2013 (below); Tricoire et al 2011; Taniguchi et al 2011) and therefore EYFP staining will be extensive throughout CA1 and “co-localise” with multiple cell types. This may indeed be the reason why Basu et al did not attempt or show such data. However, it is important to note again that the images shown by Basu et al do indicate largely non-overlapping expression patterns for ChR2 in the CCK-cre and PV-cre mouse lines (see below) and that Taniguchi et al perform double staining and find no evidence for overlap of CCK and PV expression in CA1 interneurons.

Figure legend from Figure 4 Basu et al 2013: (B and C) ChR2-EYFP (green) expression in the hippocampus of CCK-Cre (B) and PV-Cre (C) mouse lines shown in a 203 confocal tiled image of the entire hippocampal section (top) and higher magnification views of CA1 (bottom left) overlaid with DAPI staining (blue). Uninfected CA1 PN (bottom right) filled during intracellular recording with neurobiotin (streptavidin-Alexa 555, white).

3. In light of these issues above, I had the impression that the cellular and subcellular specificity of the perturbation approaches and the data evaluation may require further attention to exclude potential mistakes. Please note that for example in Figure 1B, it is described in the legend that blue cells are the neurobiotin-filled cholinergic neurons, whereas ChR2-YFP proteins are shown in yellow. If ChR2 is expressed exclusively by cholinergic neurons, then one would expect that all blue cells should contain the yellow signal. However, there is an apparent mismatch in the image between yellow and blue, these cells are located adjacent to each other. This does not make sense. Please provide quantitative anatomical evidence that Cre-mediated expression in the medial septum is restricted to ChAT neurons, but it is not present in adjacent GABAergic neurons.

We thank the reviewer for alerting us to the lack of precision in the description of the data in Figure 1B. Cells in the medial septum were patched blind to genotype (our electrophysiology setup is not configured for sensitive fluorescence microscopy required to detect membrane bound ChR2-EYFP) and therefore multiple cells were often patched within a slice with only a fraction of these positive for ChAT. This explains why not all the blue neurobiotin filled neurons colocalise with EYFP. The electrophysiological recordings shown in Figure 1B were made from the neuron now indicated by the arrow. Higher resolution images shown below demonstrate the colocalization of ChR2-EYFP on the membrane of this neuron. We have reworded the figure legend to make this clear and provided more detail in the methods (lines 638 and 907).

We are confident that the optogenetic stimulation in hippocampal slices from ChAT-ChR2 is specific for cholinergic axons because the effects were blocked by cholinergic antagonists and we did not see any evidence for glutamate or GABA transmission, as discussed in the previous round of review.

Figure Legend. Image from Figure 1B shown with recorded ChAT+ neuron indicated by arrow. Higher magnification images on the right indicate neurobiotin in the soma with ChR2-EYFP on the somatic membrane.

4. Similarly, in Figure 1C, the depicted neuron in light blue is a large multipolar interneuron based on its morphology. CA1 pyramidal cells have one thick apical dendrite with secondary dendrites first branching about 200 micron away from the cell body (see for example Figure 1 in Papp et al 2001 Neuroscience), basal dendrites emerge as a tuft. If the authors misidentified the neuron they use for presentation, how the reader can trust in their data when they report cell-type-specific parameters obtained from patch-clamp recordings? I suggest the authors adding further morphological or neurochemical verification of the specific cell-type they recorded from.

We beg to differ with the reviewer on this point. The morphology of the neuron shown in Figure 1C is consistent with a CA1 pyramidal neuron. The apical dendrites of CA1 pyramidal neurons tend to emerge as a single primary dendrite but the branch point for secondary dendrites can occur at any point from the soma. Below we show a selection of filled pyramidal neurons that we have recorded from to illustrate the heterogeneity of morphology (see also the example pyramidal neuron from Basu et al 2013 above with apical dendrite that branches ~20 μm from the soma). Related to point 5 below, we also illustrate that the morphology is similar for both sagittal and transverse slices. We note that the example pyramidal neuron image cited

by the reviewer in Figure 1 of Papp et al 2001 has had dendrite branches removed so that it appears to be less branched than would be typical.

Figure legend. Top, 3 CA1 pyramidal neurons in sagittal slices filled with neurobiotin and imaged on a confocal microscope. Bottom, 4 CA1 pyramidal neurons in transverse slices filled with Alexa594 and imaged on a 2photon microscope. Red dashed lines indicate approximate strata boundaries. Note dendrites extending into lacunosum moleculare in all examples.

Importantly, we also electrophysiologically verify the identity of all recorded neurons using capacitance, input resistance and membrane potential. We now state this in the methods (line 637).

5. Finally, I refer again to the original review point 8 and 9, because the authors misunderstood my comments. In the revised manuscript, the authors state “Feedforward interneurons in the SC pathway are primarily perisomatic targeting basket cells expressing parvalbumin (PV+) or cholecystinin (CCK+) whereas the mediators of feedforward inhibition in the TA pathway are likely dendritically targeting CCK+ or neuropeptide Y (NPY+) expressing interneurons.” If both perisomatically- and dendritically-targeting CCK interneurons have dendrites in the stratum lacunosum-moleculare (Klausberger et al 2005) then why the authors expect their different recruitment by the temporo-ammonic pathway? In addition, why most parvalbumin-positive basket and axo-axonic cells have extensive dendrites in the stratum lacunosum-moleculare that receive direct innervation from the entorhinal cortex (Halasy et al 1996, Hippocampus; Kiss et al 1996, Hippocampus). My overall fear is that the slicing angle used in the authors’ lab differentially affects the dendritic trees of the different interneuron types and this leads to a major and perhaps wrong conclusion of the authors about the different recruitment and involvement of the distinct cell types (“interneuron subpopulations”) by the different pathways. Could this “higher proportion” (the toning down term used in the rebuttal letter, but not in the paper) of contribution of the distinct cell types be due to a methodical issue? Would it be possible to exclude this possibility by using another slicing angle?

We thank the reviewer for clarifying their point from the previous round of review. We completely agree that multiple subtypes of PV+ and CCK+ neurons receive direct glutamatergic input from the TA pathway but this does not mean that *functionally* they are all engaged in feedforward inhibition. The sentence quoted from our manuscript by the reviewer is our summary of the published evidence from several groups for the interneuron subtypes engaged functionally by the SC and TA pathways (see references 34-38 in the manuscript). The conclusions drawn in these published studies are based mainly on observations from slice experiments with a variety of slice orientations. For example, Milstein et al 2015 use longitudinal slices and demonstrate very limited TA input to PV interneurons, Basu et al 2013 use horizontal or transverse slices and demonstrate much greater CCK than PV involvement in feedforward inhibition in the TA pathway. In our study, we use sagittal slices to investigate the feedforward inhibition and see results that are in agreement with both Milstein et al and Basu et al. Therefore, it does not appear that slice orientation is a major factor in determining the relative engagement of interneuron subtypes in feedforward inhibition in the TA pathway. As the reviewer highlights, both PV and CCK interneurons have dendrites in stratum lacunosum moleculare and we show below that this is also true for PV and CCK expressing neurons in our sagittal slices. Given the similar dendrite locations for PV and CCK neurons it is perhaps surprising that CCK neurons receive considerably greater functional input from the entorhinal cortex than PV neurons but the available evidence indicates that this is the case.

Ideally, we would perform a further test using an alternative slicing angle but unfortunately this is not feasible. In our experiments we need to stimulate the excitatory and feedforward inhibitory inputs in both the SC and TA pathways without directly exciting inhibitory interneurons. To our knowledge, this is the first time any group has accomplished this and

highlights the technical difficulty of recording multiple input pathways simultaneously. The experiments require placement of stimulation electrodes at a lateral distance $>500\mu\text{m}$ from the recorded CA1 pyramidal neuron. We have found that only sagittal slice orientation maintains the necessary axon pathways sufficiently to allow this configuration of stimulation electrodes.

Figure legend. Top, PV⁺ interneuron in CA1 area of the hippocampus (left) and its reconstruction (right) highlighting its position in relation to strata boundaries. 5 additional examples of PV⁺ reconstructed interneurons are shown below. Bottom, CCK⁺ interneuron in CA1 (left) and its reconstruction (right). 5 additional examples of CCK⁺ reconstructed interneurons are shown below. Dendrites in bold, axons in faint. Neurons filled with neurobiotin and imaged with confocal microscope. PV and CCK expression confirmed with dual labelling.

Overall, I hope I could convince the authors that they should reconsider that two major conclusions described in the Abstract as “Entorhinal and CA3 pathways engage distinct

feedforward interneuron subpopulations and cholinergic modulation is mediated differentially by presynaptic muscarinic M3 and M4 receptors respectively” may not necessarily be correct based on the presented data and the existing literature.

For the reasons given above we stand by our central conclusions and believe the statements made in the abstract are a fair representation of the data.

REVIEWER COMMENTS

Reviewer #1 (Remarks to the Author):

I do hope that Palacios-Filardo and colleagues understand that I devote my time to write my suggestions with the aim to help them. After very careful reading of the responses and a look at the data again, I would like to alert the authors that some of their conclusions may not be fully justified by their data. It is certainly up to the authors if they wish to consider these collegial suggestions. Regardless of the prestige of the journal in which this study will be published, my impression is that expert colleagues will have mixed feelings to see this study in its present form.

1. One major caveat is the existence of presynaptic M3 receptor and its potential homeostatic compensation by presynaptic M1 receptors in M3 knockout mice on TA pathways. First, M1 and M3 are primarily Gq/11 coupled GPCRs and Gq/11-coupled GPCRs are almost exclusively dendritic (or glial). To convince the readers that their findings go against the prevailing view, the authors should strongly aim to present very solid evidence and not only circumstantial evidence for their conclusion as I emphasized in my previous reviews. Second, the authors still refer to the 1995 Levey et al paper that there is anatomical evidence for the presence of M3 on TA pathways. This is wrong. In lines 297-299, the authors state "The high density of muscarinic M3 receptors localised to Stratum Lacunosum Moleculare where TA inputs synapse in CA1 suggests a role for M3 receptors modulating the TA pathway." I kindly ask the Editor and the Reviewer colleagues to have a careful look at Figure 2 of Levey et al 1995 Journal of Neuroscience. The cell bodies in CA1 are clearly positive, faint parallel running dendrites are visible in stratum radiatum. Zooming into the stratum lacunosum-moleculare reveals M3-positive processes. These processes are thicker and more abundant compared to what would be expected based on a TA pathway dye labeling or a vGluT1-immunostaining of incoming axon terminals in the str. lacunosum-moleculare. In my opinion, these are apical dendrites. The HCN1-immunostaining does matter here in contrast to what the authors claim. I advise to have a look Lorincz et al 2002 Nature Neuroscience Figure 2 d. Using the same immunoperoxidase approach, HCN1 shows a very similar pattern to M3. Thus, it is entirely possible that a membrane protein has low copy numbers in the stratum radiatum and then concentrated in the distal dendrites in the str. lacunosum-moleculare. Third, I still find the similar effect sizes of the CB1 antagonist and the M3 knockout mice intriguing. PPR is indicative, but not unequivocal evidence for a presynaptic locus, because this parameter for short-term plasticity can be strongly influenced by postsynaptic receptor changes in abundance and desensitization (Trussel et al 1993 Neuron; Heine et al 2008 Science; Frischknecht et al 2009 Nature Neuroscience; Opazo et al 2010 Neuron). Fourth, the suggestion that dendritic M1 receptors (Yamasaki et al 2010 Journal of Neuroscience) can presynaptically appear on the TA pathway in the M3 knockout mice would suggest a remarkable capacity for molecular homeostasis. If this is the case, then the M1 antagonist nitrocaramiphen should be able to completely prevent the carbachol effect in the M3 knockout mice. This is a feasible experiment that the authors may consider. Carbachol and opto-stimulation of cholinergic fibers depresses EPCs in wild-type mice (Fig 4G-H). The M3 receptor antagonist DAU 5884 can completely block this response in wild-type mice (this is convincing). If M1 homeostatically replaces M3 in the knockout mice then nitrocaramiphen should have a strong effect in M3 knockout mice. Please also clarify if nitrocaramiphen was used in 1 μ M or 100 nM (there is a mismatch between the text and the figure legends).

2. I appreciate the authors effort to show that a fluorescent dye can diffuse to a distal dendrite. My comments again aim to highlight for them the difficulties of interpretation of circumstantial evidence. First, the multiphoton image in A visualizes only the major branches, but not the thin dendrites beyond the branch points. Second, GDP- β -S has different physicochemical properties and may readily be sequestered via the numerous GPCRs along the dendrites. For example please note the statement in Meis and Pape 2001 Journal of Physiology "One possible point of concern relates to distally located K⁺ channels, which, due to limited diffusion of GDP- β -S into dendritic compartments, might not have been reached." In this experiment, the critical issue again is the fact that the authors draw a major conclusion based on the absence of effect of a perturbation approach. The authors may be right, but may also

be wrong, it is difficult to determine. One would need a direct positive evidence that the perturbation approach is efficiently working. This is important to be able to unequivocally state that GDP- β -S had no effect because dendritic muscarinic GPCRs are not involved in the phenomenon (absence of evidence is not the evidence for absence). Moreover, the data distribution in Fig 5H may give a hint that GDP- β -S had some blocking effect at the trend level. Due to the robust variation in control (carbachol) experiments, I assume that the likelihood of having a false negative statistical error is very high. What is the level of β and the power of this analysis? In other words, with the current number of samples in the carbachol experiment, what would be the probability to be able to see a 25% effect size, assuming that PPR is linear and goes from 100% to 125% in case of carbachol treatment alone, and GDP- β -S would bring it back to 100%.

3. My comments are detailed above. But again, considering the lack of an available conclusive anatomical evidence, the statement of the authors in the response letter "Therefore, the immuno staining is entirely consistent with M3 localisation on presynaptic terminals of the TA pathway." is not justified. Regarding the M4-immunostaining, if the immunostaining pattern in Levey et al would visualize the Schaffer collaterals then the stratum oriens should have an increased density similarly to stratum radiatum. Moreover, the M4 antibody visualizes the nuclei of pyramidal cells indicating a strong antibody background. Finally, de Vin et al 2013 Brain Research draw the conclusion based on physiological approaches and using pharmacological tools that M3, but not M4 presynaptically regulates glutamate release from the Schaffer collaterals. This is apparently different from the conclusion of the present manuscript, but not mentioned and discussed in the manuscript. To be honest, I don't think the de Vin paper is right either but highlights that one needs to be careful about the strong statements until sufficient models are not available to provide causal evidence instead of circumstantial evidence. The appropriate model here would be the specific deletion of M3 receptors in the entorhinal cortex and then measurement of the effects of carbachol in the hippocampus. I certainly do not require that the authors would invest into this model at this stage, but I would like to emphasize that the authors should be very careful and substantially tone down their conclusions about their data about the selective M3 versus M4 regulation of TA versus SC pathways, respectively.

4. I thank the authors for their clarification. In their reply, they state that "if we were attempting to make strong statements about interneuron subtypes based on using just the CCK-cre line the reviewers' comments would be fully justified."

Well, in my opinion, the following statement in the revised Abstract is very strong and none of the two major conclusions are unequivocally justified. "Entorhinal and CA3 pathways engage distinct feedforward interneuron subpopulations and cholinergic modulation is mediated differentially by presynaptic muscarinic M3 and M4 receptors respectively."

It is an important issue of why the authors DO NOT attempt to validate their own experimental model. Considering their microscopic images, they have the capacity to run a double-labeling experiment to establish how the ratio of PV and CCK cells that are opto-stimulated in the distinct models. I need to call the attention of the authors to the recent paper by Dudok et al 2021 Neuron. Dudok et al used the same mouse line as the authors of the current manuscript (Stock number 012706). Please have a look at Supplementary Figure 3. Unfortunately, the CCK-Cre approach using this mouse line affects substantially more PV interneuron than CCK interneuron. Only 12% of CCK interneurons express Chr2 when using the CCK-Cre approach. While the reason for this false positive and false negative artifact is unknown, this problem is well-known in the interneuron community, several labs experienced the same problem. When these colleagues read the present study, they will wonder whether the observed difference could purely be related e.g. to the distinct ratio of PV interneurons that were opto-stimulated. I think this also explains why there are overlapping effects in the present experiments and instead of clear-cut differences, several data are more at the trend levels. The authors may be lucky, and in their mice the promoter may work better. Therefore, I would either do a simple quantitative experiment to establish the ration of PV and CCK interneurons in the different mouse models or remove all statements about interneuron classes, types, populations, subpopulations etc.

5. The new Fig 1b in the rebuttal letter is convincing and the fact that the authors made blind patch explains the lack of 100% colocalization. The authors kept the old Fig 1b in the resubmitted manuscript. Please include the new, much more convincing version, because the readers may have the same issues as the reviewer with the previous version. Please also explain why there are many

blue cells that are not yellow, and due to blind patch these cells may not be cholinergic (I saw in the Methods, but Figure legend would be more appropriate). Only 1 out of 6 blind patched cells turned out to be cholinergic? This is rather surprising as cholinergic cells are packed in the medial septum.

6. I agree with the authors that all cells presented in the rebuttal letter are pyramidal neurons. They have one clear apical dendrite and a characteristic basal tuft. On the other hand, I still maintain that the cell presented in Fig 1 C is more likely a multipolar interneuron. It does not have one clear apical dendrite but at least three (or perhaps four) apical dendrites run into multipolar directions in the radiatum. The basal tuft is also absent, instead other equally thick multipolar dendrites run in the oriens. The reviewer filled and analyzed several hundreds of pyramidal cells and interneurons. If the authors wish to convince the readership about the strength of their data, they may consider using a much more typical pyramidal neuron as an example. Regarding the physiological criteria, please note that the resting membrane potential and the input resistance are very similar for PV interneurons and CA1 pyramidal cells. I advise the authors to include the firing pattern that would be an unequivocal feature to distinguish CA1 pyramidal neurons from fast-spiking pyramidal cells.

7. I agree with the authors that it would be impossible and useless to repeat the entire project with different slicing angles. I suggest presenting the facts that both PV and CCK interneurons receive direct inputs from the TA pathways and include a statement in the Discussion that slicing may impair interneuron types in a different manner that may contribute to some of the observed physiological differences upon TA pathway stimulation. The filled cell images in the rebuttal letter exhibit convincing interneuron morphology. Please include these cells in a Supplementary Figure and please provide the direct evidence that these cells are PV- or CCK-positive. As stated in the rebuttal letter, this experiment has already been done by the authors. It is reassuring that the authors are capable to perform PV- and CCK-immunostaining, because it means that they can verify the neurochemical phenotype of the interneurons in their CCK-Cre-based mouse model to provide direct experimental evidence for their conclusions in the Abstract and throughout the manuscript.

8. For the reasons described above, I maintain that the experimental data in the manuscript do not necessarily support the two major conclusions of the Abstract: "Entorhinal and CA3 pathways engage distinct feedforward interneuron subpopulations and cholinergic modulation is mediated differentially by presynaptic muscarinic M3 and M4 receptors, respectively".

Response to reviewer's comments (comments in black, responses in blue)

Reviewer #1 (Remarks to the Author):

I do hope that Palacios-Filardo and colleagues understand that I devote my time to write my suggestions with the aim to help them. After very careful reading of the responses and a look at the data again, I would like to alert the authors that some of their conclusions may not be fully justified by their data. It is certainly up to the authors if they wish to consider these collegial suggestions. Regardless of the prestige of the journal in which this study will be published, my impression is that expert colleagues will have mixed feelings to see this study in its present form.

We very much appreciate the time and effort put in by the reviewer into assessing the data and our conclusions. We understand the concerns and here address them with additional data and explanations. We also make adjustments to the text in the revised manuscript to better describe the findings, our conclusions and their context.

1. One major caveat is the existence of presynaptic M3 receptor and its potential homeostatic compensation by presynaptic M1 receptors in M3 knockout mice on TA pathways. First, M1 and M3 are primarily Gq/11 coupled GPCRs and Gq/11-coupled GPCRs are almost exclusively dendritic (or glial). To convince the readers that their findings go against the prevailing view, the authors should strongly aim to present very solid evidence and not only circumstantial evidence for their conclusion as I emphasized in my previous reviews.

We agree that a role for M3 receptors presynaptically is surprising but this is what our data indicate. We have also performed additional experiments suggested by the reviewer that support these conclusions (Supplementary Fig S5).

Second, the authors still refer to the 1995 Levey et al paper that there is anatomical evidence for the presence of M3 on TA pathways. This is wrong. In lines 297-299, the authors state "The high density of muscarinic M3 receptors localised to Stratum Lacunosum Moleculare where TA inputs synapse in CA1 suggests a role for M3 receptors modulating the TA pathway." I kindly ask the Editor and the Reviewer colleagues to have a careful look at Figure 2 of Levey et al 1995 Journal of Neuroscience. The cell bodies in CA1 are clearly positive, faint parallel running dendrites are visible in stratum radiatum. Zooming into the stratum lacunosum-moleculare reveals M3-positive processes. These processes are thicker and more abundant compared to what would be expected based on a TA pathway dye labeling or a vGluT1-immunostaining of incoming axon terminals in the str. lacunosum-moleculare. In my opinion, these are apical dendrites. The HCN1-immunostaining does matter here in contrast to what the authors claim. I advise to have a look Lorincz et al 2002 Nature Neuroscience Figure 2 d. Using the same immunoperoxidase approach, HCN1 shows a very similar pattern to M3. Thus, it is entirely possible that a membrane protein has low copy numbers in the stratum radiatum and then concentrated in the distal dendrites in the str. lacunosum-moleculare.

We agree that the immunostaining for M3 receptors in Levey et al is consistent with a postsynaptic location and there are undoubtedly M3 receptors located on CA1 pyramidal neurons. The interesting observation in Levey et al is that the M3 receptors are preferentially localised to Stratum Lacunosum Moleculare where TA inputs synapse in CA1 suggesting a role for M3 receptors modulating the TA pathway. This is what we state at the start of this section in the results where we refer to the Levey data (quoted by the reviewer) without concluding whether the receptors are located pre- or post-synaptically. We now include text to make this clear (line 345). However, in our opinion it is not possible to make a definitive determination on the pre- or post-synaptic location of M3 receptors based on the Levey data. One can make a comparison to the HCN1 staining in Lorincz (postsynaptic localisation) or to the M4 staining in Levey (presynaptic localisation) and find little to distinguish. The pre- or post-synaptic location of M3 receptors is tested explicitly by experiments in Figure 5 where we show that the postsynaptic M3 receptors are unlikely to mediate the effects on presynaptic release at the TA pathway that we describe. We have now adjusted the discussion to reflect our scientific rationale and the possibility of postsynaptic location (lines 521-538). We also adjust the wording of the abstract to reflect the selective roles of M3 and M4 in each pathway without defining their location.

Third, I still find the similar effect sizes of the CB1 antagonist and the M3 knockout mice intriguing. PPR is indicative, but not unequivocal evidence for a presynaptic locus, because this parameter for short-term plasticity can be strongly influenced by postsynaptic receptor changes in abundance and desensitization (Trussel et al 1993 Neuron; Heine et al 2008 Science; Frischknecht et al 2009 Nature Neuroscience; Opazo et al 2010 Neuron).

We apologise for any ambiguity here. We referred to PPR in our previous response as an additional line of evidence showing that the CB1 receptor antagonist did not prevent the actions of carbachol on the TA pathway. The important comparison is between PPR change in carbachol and PPR change in carbachol/AM251. In both conditions PPR is increased indicating AM251 did not prevent the change in synaptic function.

Fourth, the suggestion that dendritic M1 receptors (Yamasaki et al 2010 Journal of Neuroscience) can presynaptically appear on the TA pathway in the M3 knockout mice would suggest a remarkable capacity for molecular homeostasis. If this is the case, then the M1 antagonist nitrocaramiphen should be able to completely prevent the carbachol effect in the M3 knockout mice. This is a feasible experiment that the authors may consider. Carbachol and opto-stimulation of cholinergic fibers depresses EPCs in wild-type mice (Fig 4G-H). The M3 receptor antagonist DAU 5884 can completely block this response in wild-type mice (this is convincing). If M1 homeostatically replaces M3 in the knockout mice then nitrocaramiphen should have a strong effect in M3 knockout mice. Please also clarify if nitrocaramiphen was used in 1 μ M or 100 nM (there is a mismatch between the text and the figure legends).

We have now performed the suggested experiments and show that the M1 receptor antagonist nitrocaramiphen completely blocked the effects of carbachol on the TA pathway in the M3 knockout mice (Supplementary Fig S5). This is entirely consistent with the data showing that the M1 agonist GSK-5 only depresses TA pathway transmission in the M3 knockout mice.

Thus, we now provide two lines of evidence that M1 receptors are able to partially compensate for the loss of M3 receptors.

Nitrocaramiphen was used at 1 μ M and this is now made clear in the text and figure legend.

2. I appreciate the authors effort to show that a fluorescent dye can diffuse to a distal dendrite. My comments again aim to highlight for them the difficulties of interpretation of circumstantial evidence. First, the multiphoton image in A visualizes only the major branches, but not the thin dendrites beyond the branch points. Second, GDP- β -S has different physicochemical properties and may readily be sequestered via the numerous GPCRs along the dendrites. For example please note the statement in Meis and Pape 2001 Journal of Physiology “One possible point of concern relates to distally located K⁺ channels, which, due to limited diffusion of GDP- β -S into dendritic compartments, might not have been reached.” In this experiment, the critical issue again is the fact that the authors draw a major conclusion based on the absence of effect of a perturbation approach. The authors may be right, but may also be wrong, it is difficult to determine. One would need a direct positive evidence that the perturbation approach is efficiently working. This is important to be able to unequivocally state that GDP- β -S had no effect because dendritic muscarinic GPCRs are not involved in the phenomenon (absence of evidence is not the evidence for absence). Moreover, the data distribution in Fig 5H may give a hint that GDP- β -S had some blocking effect at the trend level. Due to the robust variation in control (carbachol) experiments, I assume that the likelihood of having a false negative statistical error is very high. What is the level of β and the power of this analysis? In other words, with the current number of samples in the carbachol experiment, what would be the probability to be able to see a 25% effect size, assuming that PPR is linear and goes from 100% to 125% in case of carbachol treatment alone, and GDP- β -S would bring it back to 100%.

One can never be 100% certain, but we can be highly confident that GDP- β -S diffuses into all the very furthest and thinnest dendrites in the time after break in before we apply carbachol. By measuring the positive effect of GDP- β -S on somatic potassium currents and the likely time to diffuse to the most distal dendrites we have performed the best controls available to us. It is likely that GDP- β -S diffuses faster than Alexa594 because it is a smaller molecule and this may be offset by the slightly further travel into the thinnest dendrites in comparison to the data shown in our previous response. To clarify, the quantification shown in the figure is from a dendrite visible from break in and therefore not the thinnest, however, we have also quantified fluorescence in the thinnest dendrites and find that they reach saturation within 10-15 minutes after break in. Therefore, we expect that the total diffusion time is less than 15 minutes even to the furthest and thinnest dendrites which is the time we wait between break in and carbachol application.

There is perhaps a hint of an effect of GDP- β -S in the PPR data (that is necessarily more variable than the EPSC amplitude data) but when we consider the PPR data together with the EPSC amplitude data there is very little evidence of any difference between the control (carbachol) and GDP- β -S (carbachol + GDP- β -S) conditions. A power analysis indicates that the probability of a false negative (β) in the EPSC amplitude data is <0.001 and for the PPR data is 0.13. This suggests two false negative results is highly unlikely.

3. My comments are detailed above. But again, considering the lack of an available conclusive anatomical evidence, the statement of the authors in the response letter “Therefore, the immuno staining is entirely consistent with M3 localisation on presynaptic terminals of the TA pathway.” is not justified. Regarding the M4-immunostaining, if the immunostaining pattern in Levey et al would visualize the Schaffer collaterals then the stratum oriens should have an increased density similarly to stratum radiatum. Moreover, the M4 antibody visualizes the nuclei of pyramidal cells indicating a strong antibody background. Finally, de Vin et al 2013 Brain Research draw the conclusion based on physiological approaches and using pharmacological tools that M3, but not M4 presynaptically regulates glutamate release from the Schaffer collaterals. This is apparently different from the conclusion of the present manuscript, but not mentioned and discussed in the manuscript. To be honest, I don’t think the de Vin paper is right either but highlights that one needs to be careful about the strong statements until sufficient models are not available to provide causal evidence instead of circumstantial evidence. The appropriate model here would be the specific deletion of M3 receptors in the entorhinal cortex and then measurement of the effects of carbachol in the hippocampus. I certainly do not require that the authors would invest into this model at this stage, but I would like to emphasize that the authors should be very careful and substantially tone down their conclusions about their data about the selective M3 versus M4 regulation of TA versus SC pathways, respectively.

Conditional knockouts of M3 receptors in entorhinal cortex neurons and M4 receptors in CA3 pyramidal cells would be good tools to ascertain the pre- or post-synaptic location and pathway specificity of M3 and M4 receptors, preferably with temporal precision to avoid compensation. However, these tools do not exist and it is not feasible to develop them. In their absence it is well established that M4 receptors act selectively on the Schaffer collateral pathway to depress inputs to CA1. This is shown by Dasari and Gullledge (2011) using genetic tools, by Thorn et al (2017) using pharmacological tools and replicated by us in this manuscript (Figure 4). Some groups have reported alternative conclusions using less-selective pharmacological tools but like the reviewer we do not believe this evidence is as strong as the evidence cited here. For the role of M3 receptors in the TA pathway we again used a dual genetic and pharmacological approach to show causal evidence of selectivity for the effects of M3 in this pathway. As discussed above in point 2, we specifically address the issue of the location of the M3 receptors in the manuscript (Figure 5) and our results support the conclusion that the majority of the effects are mediated by presynaptically located receptors. We note the concerns of the reviewer that we cannot unequivocally rule out a role for M3 receptors located postsynaptically and we now make this explicitly clear in the text as described above.

4. I thank the authors for their clarification. In their reply, they state that “if we were attempting to make strong statements about interneuron subtypes based on using just the CCK-cre line the reviewers’ comments would be fully justified.” Well, in my opinion, the following statement in the revised Abstract is very strong and none of the two major conclusions are unequivocally justified. “Entorhinal and CA3 pathways engage distinct feedforward interneuron subpopulations and cholinergic modulation is mediated differentially

by presynaptic muscarinic M3 and M4 receptors respectively.” It is an important issue of why the authors DO NOT attempt to validate their own experimental model. Considering their microscopic images, they have the capacity to run a double-labeling experiment to establish how the ratio of PV and CCK cells that are opto-stimulated in the distinct models. I need to call the attention of the authors to the recent paper by Dudok et al 2021 Neuron. Dudok et al used the same mouse line as the authors of the current manuscript (Stock number 012706). Please have a look at Supplementary Figure 3. Unfortunately, the CCK-Cre approach using this mouse line affects substantially more PV interneuron than CCK interneuron. Only 12% of CCK interneurons express Chr2 when using the CCK-Cre approach. While the reason for this false positive and false negative artifact is unknown, this problem is well-known in the interneuron community, several labs experienced the same problem. When these colleagues read the present study, they will wonder whether the observed difference could purely be related e.g. to the distinct ratio of PV interneurons that were opto-stimulated. I think this also explains why there are overlapping effects in the present experiments and instead of clear-cut differences, several data are more at the trend levels. The authors may be lucky, and in their mice the promoter may work better. Therefore, I would either do a simple quantitative experiment to establish the ration of PV and CCK interneurons in the different mouse models or remove all statements about interneuron classes, types, populations, subpopulations etc.

Here we show double labelling for PV and CCK expression in the hippocampus requested by the reviewer. We crossed the cre-dependent TdTomato reporter mouse line with PV-cre mice and immunostained for CCK. To quantify the percentage of PV cells that express CCK we counted any cells that had significant CCK signal anywhere in the cell body. This is a highly conservative measure since the CCK signal was widespread (see image bottom right) and often not well spatially associated with the cell body outline. For example, the 2 cells highlighted with arrowheads were counted as co-expressing but it is more likely that the CCK signal is from background expression in pyramidal cells than co-expression in PV interneurons. Therefore, the measured 25% of PV cells that co-express CCK is almost certainly an overestimate. If we restrict definition of co-expression to cells with good spatial overlap of CCK and PV signal the rates of co-expression drop below 5%. This is in agreement with the estimated overlap in expression by Taniguchi et al and Whissell et al that use intersectional genetics to restrict coexpression to interneurons (see below).

These data show how difficult it is to accurately discern co-expression when CCK expression is widespread in pyramidal neurons and why we prefer not to include these data in the manuscript since they do not significantly add to the data already published by other groups using better intersectional genetic approaches. The difficulties increase if we perform the colocalization the other way around by crossing the TdTomato reporter mice with CCK-cre mice. In the image below (left) we see widespread TdTomato expression reflecting the expression of cre in pyramidal cells (and dentate gyrus granule cells) as well as CCK expressing interneurons. On the right we show a very similar expression pattern described by Taniguchi et al (2011, Fig 6D) using the same CCK-cre mice crossed with a GFP reporter mouse line. The widespread expression of CCK is also evident from the expression of ChR-EYFP (shown in Basu et al in previous response letter).

Reprinted from Neuron, VOLUME 71, ISSUE 6, Taniguchi, Hiroki et al., A Resource of Cre Driver Lines for Genetic Targeting of GABAergic Neurons in Cerebral Cortex, P995-1013., Copyright 2011, with permission from Elsevier.

The widespread expression is presumably why very few if any labs attempt to colocalise expression of other markers of interneuron subtypes with CCK-cre mice. To get around this issue other groups have used an intersectional genetic strategy (not immediately available to us) to visualise cre expression only in GABAergic interneurons. This was first used by Taniguchi et al to reveal very little if any overlap in CCK and PV expression colocalization in interneurons both in hippocampus and cortex (data from Figure 6 shown below with figure legend).

(G) In hippocampal dentate gyrus, CCK axon terminals (green) segregated from PV axon terminals (red). (H) CCK GABAergic neurons (star) and axons (green) in neocortex. PV axons were labeled with an antibody (red), and all cell somata were labeled with TOTO-3 (blue). Inset: CCK axon terminals (white arrowheads) segregated from PV axon terminals (yellow arrowheads) around the same cell soma. Scale bars: 25 μ m in (G) and (H), 5 μ m in inset of (H).

Subsequently, other groups have used similar intersectional genetic strategies with similar results. Whissell et al (2015) show different expression profiles for CCK-cre and PV-cre in hippocampus CA1 interneurons. Data taken from Figure 2 of this paper (below) shows ~40% of CCK expressing interneurons are in stratum radiatum of CA1 whereas very few PV expressing interneurons are found in this layer which closely agrees with other published data. A subsequent study (Whissell et al 2019) showed that a high proportion (~75%) of CCK-cre interneurons in these mice also stained positive for CCK. The Dudok et al paper cited by the reviewer uses a viral intersectional approach for GCaMP6f expression that is not so directly comparable (since our experiments use cross-bred reporter mouse lines) and will be subject to the additional complication of variable viral expression in different spatial locations and neuronal subtypes. This study reports levels of co-expression for PV in the CCK-cre mice of ~25% cells CCK-cre cells expressing PV. Confusingly, Dudok et al also report low numbers of GCaMP6f expressing cells in the CCK-cre mice that colocalise with CCK expression. Indeed, the percentage of CCK expressing cells (~14%) is lower than that previously reported for all GABAergic interneurons (~22%, eg Whissell et al 2019) which implies that Dudok et al are also labelling some pyramidal neurons with GCaMP6f. This is speculation but it highlights the difficulty in drawing conclusions from these data and making comparisons to our experimental model.

(D,E) Distribution of CCK- and PV- GABA cells by hippocampal layer. CCK-GABA cells were most common in the sr layer but were also found in the sp and so layers. PV- GABA cells were more concentrated in the sp layer, but were also found in the so layer. Abbreviations by layer: hi, hilus, sg, stratum granulosum, sl, stratum lucidum, slm, stratum lacunosum-moleculare, sm, stratum moleculare, so, stratum oriens, sp, stratum pyramidale and sr, stratum radiatum. Scalebar = 500 μ M.

What can we conclude from these studies and what are the implications for our study?

1. The data discussed above show that cre expression in CCK-cre mice is widespread in pyramidal cells as well as CCK expressing interneurons. CCK expression in pyramidal neurons is not an issue for our study since our experiments that use these mice are performed in the presence of glutamate receptor antagonists.
2. The data also indicate there is some overlap in expression of CCK and PV in interneurons but the degree of overlap is small (<25% and likely ~5%). In our study there are clear functional differences observed between the CCK-cre and PV-cre mice which supports the conclusion that the overlap in optogenetically targeted interneuron populations is small.
3. The conclusion that different populations of interneurons mediate feedforward transmission in the Schaffer collateral and temporoammonic pathways is not controversial nor is the fact that these populations may overlap. The specific identity of these different interneuron populations is less clear but based on previous evidence from multiple groups the major components of feedforward inhibition are PV and CCK interneurons in the Schaffer collateral and temporoammonic pathways respectively. Our data support these conclusions but do not provide definitive evidence since we are correlating observations using the CCK-cre and PV-cre mice with feedforward inhibitory transmission.

On reflection, this last point could be better articulated in the manuscript and we have now altered the text in multiple places to address this. The sentence from the abstract now reads “Entorhinal and CA3 pathways engage different feedforward interneuron subpopulations and cholinergic modulation of presynaptic function is mediated differentially by muscarinic M₃ and

M₄ receptors respectively.” We have altered text at lines 227, 239, 252, 265, and include a brief summary of the data presented in this response letter in the discussion (lines 487-489).

5. The new Fig 1b in the rebuttal letter is convincing and the fact that the authors made blind patch explains the lack of 100% colocalization. The authors kept the old Fig 1b in the resubmitted manuscript. Please include the new, much more convincing version, because the readers may have the same issues as the reviewer with the previous version. Please also explain why there are many blue cells that are not yellow, and due to blind patch these cells may not be cholinergic (I saw in the Methods, but Figure legend would be more appropriate). Only 1 out of 6 blind patched cells turned out to be cholinergic? This is rather surprising as cholinergic cells are packed in the medial septum.

We now include the new data in Figure 1B along with a description in the figure legend.

Cholinergic cells may be packed in the medial septum but they form a minority of the total number of cells in this region, as shown in multiple studies (eg. Jarzebowski et al eLife 2021).

6. I agree with the authors that all cells presented in the rebuttal letter are pyramidal neurons. They have one clear apical dendrite and a characteristic basal tuft. On the other hand, I still maintain that the cell presented in Fig 1 C is more likely a multipolar interneuron. It does not have one clear apical dendrite but at least three (or perhaps four) apical dendrites run into multipolar directions in the radiatum. The basal tuft is also absent, instead other equally thick multipolar dendrites run in the oriens. The reviewer filled and analyzed several hundreds of pyramidal cells and interneurons. If the authors wish to convince the readership about the strength of their data, they may consider using a much more typical pyramidal neuron as an example. Regarding the physiological criteria, please note that the resting membrane potential and the input resistance are very similar for PV interneurons and CA1 pyramidal cells. I advise the authors to include the firing pattern that would be an unequivocal feature to distinguish CA1 pyramidal neurons from fast-spiking pyramidal cells.

We appreciate the reviewer’s experience in this matter and now include an alternative CA1 pyramidal neuron in Figure 1C that has a more classical appearance.

Including the firing pattern of neurons would be a useful addition but most of our experiments use a Cs⁺ based internal solution that precludes assessment of spike characteristics.

7. I agree with the authors that it would be impossible and useless to repeat the entire project with different slicing angles. I suggest presenting the facts that both PV and CCK interneurons receive direct inputs from the TA pathways and include a statement in the Discussion that slicing may impair interneuron types in a different manner that may contribute to some of the observed physiological differences upon TA pathway stimulation. The filled cell images in the rebuttal letter exhibit convincing interneuron morphology. Please include these cells in a Supplementary Figure and please provide the direct evidence that these cells are PV- or CCK-positive. As stated in the rebuttal letter, this experiment has already been done by the authors. It is reassuring that the authors are capable to perform PV- and CCK-immunostaining, because it means that they can verify the neurochemical

phenotype of the interneurons in their CCK-Cre-based mouse model to provide direct experimental evidence for their conclusions in the Abstract and throughout the manuscript.

Supplementary Figure S8 now shows the interneuron morphologies and is referred to in a section of the discussion on the relative interneuron engagement in feedforward inhibition (lines 481-482).

8. For the reasons described above, I maintain that the experimental data in the manuscript do not necessarily support the two major conclusions of the Abstract: “Entorhinal and CA3 pathways engage distinct feedforward interneuron subpopulations and cholinergic modulation is mediated differentially by presynaptic muscarinic M3 and M4 receptors, respectively”.

We have now adjusted this sentence to more accurately reflect our findings. It now reads “Entorhinal and CA3 pathways engage different feedforward interneuron subpopulations and cholinergic modulation of presynaptic function is mediated differentially by muscarinic M₃ and M₄ receptors respectively.”